# Effects of accelerometer-based sedentary time and physical activity on DEXA-measured fat mass in 6059 children

Andrew O. Agbaje [1,2] ✉, Wei Perng[3] & Tomi-Pekka Tuomainen[1]

Globally, childhood obesity is on the rise and the effect of objectively measured movement behaviour on body composition remains unclear. Longitudinal and causal mediation relationships of accelerometer-based sedentary time (ST), light physical activity (LPA), and moderate-to-vigorous physical activity (MVPA) with dual-energy X-ray absorptiometry-measured fat mass were examined in 6059 children aged 11 years followed-up until age 24 years from the Avon Longitudinal Study of Parents and Children (ALSPAC), UK birth cohort. Over 13-year follow-up, each minute/day of ST was associated with 1.3 g increase in fat mass. However, each minute/day of LPA was associated with 3.6 g decrease in fat mass and each minute/day of MVPA was associated with 1.3 g decrease in fat mass. Persistently accruing ≥60 min/day of MVPA was associated with 2.8 g decrease in fat mass per each minute/day of MVPA, partly mediated by decrease insulin and low-density lipoprotein cholesterol. LPA elicited similar and potentially stronger fat mass-lowering effect than MVPA and thus may be targeted in obesity and ST prevention in children and adolescents, who are unable or unwilling to exercise.

Childhood obesity remains a global epidemic and the latest World Health Organization (WHO) physical activity (PA) guideline recommended at least 60 min/day of moderate-to-vigorous PA (MVPA) on average in children and adolescents less than 18 years as a way to maintain a healthy weight[1–4]. Most of the objectively measured movement behaviour studies in paediatric populations on which the PA guidelines are based are cross-sectional or short-term longitudinal reports, and the rest are questionnaire-based evidence[1,5]. More than 140 randomized controlled trials have reported little or no effect of school-based PA interventions on children and adolescent's body mass index (BMI)[6,7]. Identified knowledge gaps in the paediatric population include the lack of long-term longitudinal evidence on the relationships of objectively measured sedentary time (ST), light PA (LPA), and MVPA on progressive changes in objectively measured body composition[1,3,5–8].

Moreover, whether movement behaviours exert their effect on body composition directly or via metabolic pathways is not fully known[5,9–12]. A few biological pathways on the associations between ST and adiposity identified in an experimental animal model and small sample-sized human studies include elevated inflammation, altered lipid and glucose metabolism, and muscular atrophy[8–12]. It remains unclear whether movement behaviours may causally influence body composition during growth and development or whether the association is due to other health determinants or reverse causation bias[5]. Clarifying potential causal temporal associations among movement behaviours and body composition is an important enquiry that has implications for mounting effective childhood obesity prevention programs[4]. Longitudinal evidence on the importance of timing for PA intervention effect in the growing paediatric population is limited[1,5]. It is known that males generally engage in PA more than females,

[1]Institute of Public Health and Clinical Nutrition, School of Medicine, Faculty of Health Sciences, University of Eastern Finland, Kuopio, Finland. [2]Children's Health and Exercise Research Centre, Department of Public Health and Sports Sciences, Faculty of Health and Life Sciences, University of Exeter, Exeter, UK. [3]Colorado School of Public Health, Lifecourse Epidemiology of Adiposity and Diabetes Center, University of Colorado Anschutz Medical Campus, Aurora, CO, USA. ✉e-mail: andrew.agbaje@uef.fi

potentially yielding better health among males[1,5]. However, it is important to clarify whether exposure to the amount of ST or PA typical of a particular sex, significantly influences body composition during growth from childhood through young adulthood.

In this work, we examine the longitudinal associations of cumulative accelerometer-measured ST, LPA, and MVPA, with repeated measures of BMI, waist circumference, total fat mass, trunk fat mass, and lean mass at ages 11, 15, and 24 years in the total cohort of 6059 children and according to sex. Next, we assess the extent to which the associations of movement behaviours with body composition are mediated by glucose, insulin, lipid indices, and inflammation. Furthermore, we examine potential causal and temporal interrelations (potential reverse causality) among activity levels and body composition across the follow-up using data from the Avon Longitudinal Study of Parents and Children (ALSPAC) birth cohort, England, UK. We identify that during growth from childhood to young adulthood, ST increased from 6 to 9 h/day, LPA decreased from 6 to 3 h per day, and MVPA had a U-shaped increase, averaging ~50 min/day. Over the 13-year follow-up, each minute/day of ST was associated with 1.3 g increase in fat mass. However, each minute/day of LPA was associated with 3.6 g decrease in fat mass and each minute/day of MVPA was associated with 1.3 g decrease in fat mass. Persistently accruing ≥60 min/day of MVPA was associated with 2.8 g decrease in fat mass per each minute/day of MVPA when compared with children who persistently had MVPA < 40 min/day. Higher fat mass at age 11 years may potentially lead to lower MVPA by mid-adolescence, whereas higher MVPA in mid-adolescence may lead to lower fat mass in young adulthood, thus ≥60 min/day of MVPA should be prioritized in early childhood. MVPA may potentially lower fat mass via a decrease in insulin and low-density lipoprotein cholesterol. These findings present promising evidence for updating future health guidelines. Decreasing ST by at least 3 h/day and increasing LPA by the same amount during growth from childhood through young adulthood should be strongly recommended with specific emphasis on mid-adolescence timing. LPA elicited a similar effect as MVPA and thus may be targeted in obesity prevention in children and adolescents, who are unable or unwilling to exercise.

## Results

Of 14,901 children who were alive at 1 year of age, 7159 children participated in the age 11-year clinic visit, 5509 adolescents participated in the age 15-year clinic visit and 4026 young adults participated in the age 24-year clinic visits (Supplementary Fig. 1). Altogether 6059 participants who had at least one timepoint measure of body composition and movement behaviour during ages 11–24 years were included in the primary analyses. Moreover, 2457 participants had at least one time-point ST, LPA, and MVPA measure and completed three time-point measures of body composition, whereas 917 participants had complete body composition measures at ages 11, 15, and 24 years with at least two time-point measures of ST, LPA, and MVPA. From age 11 through 24 years, females had higher total and trunk fat mass but lower lean mass than males (Table 1). ST increased from 6 to 9 h/day while LPA decreased 6 to 3 h/day from ages 11 through 24 years in both males and females (Table 1 and Fig. 1). MVPA in min/day had a J or U-shaped increase in both males and females, with males accruing more ≥60 min/day of MVPA across the 13-year follow-up period than females (Table 1 and Fig. 1). Other characteristics and results are presented in Table 1 and Supplementary Tables 1–14.

### Longitudinal associations of ST, LPA, and MVPA with body composition in 6059 participants

Cumulative ST from childhood through young adulthood was directly associated with increased BMI, total fat mass, trunk fat mass, and lean mass but decreased waist circumference cumulatively measured from ages 11–24 years, after full adjustments for cardiometabolic and

lifestyle factors including LPA and MVPA (Table 2). Cumulative LPA was associated with decreased BMI, waist circumference, total fat mass, trunk fat mass, and lean mass (Table 2). Cumulative MVPA was associated with decreased BMI, total fat mass, and trunk fat mass but with increased waist circumference and lean mass (Table 2). Persistently engaging in MVPA of ≥60 min/day from childhood through young adulthood was associated with decreased BMI, total fat mass, trunk fat mass, but with increased lean mass (Table 2). Cumulative ST, LPA, and MVPA were not associated with the lean mass-to-total fat mass ratio.

Among males, cumulatively ST from childhood was associated with increased BMI and lean mass but not total or trunk fat mass during growth from ages 11 through 24 years (Table 3). In females, cumulative ST was associated with increased BMI, total fat mass, trunk fat mass, and lean mass but decreased waist circumference (Table 3). In both males and females, cumulative LPA was associated with decreased BMI, waist circumference, total fat mass, trunk fat mass, and lean mass (Table 3). In both males and females, cumulative MVPA and persistently engaging in MVPA of ≥60 min/day was associated with decreased total fat mass, trunk fat mass during growth from childhood through young adulthood (Table 3).

### The reduced cohort of 2457 participants

The characteristics of the 2457 participants who had at least one valid ST and PA variable and complete total fat mass and lean mass are described in Supplementary Tables 10 and 11. The included 2457 participants were similar in characteristics to the 3905 participants excluded from the study who also had at least one valid ST and PA but incomplete dual-energy Xray absorptiometry (DEXA) measures of total fat mass and lean mass across all time points (Supplementary Table 10). The baseline characteristics of the 6059 participants were similar to the 2457 participants who had at least one ST and PA variables and complete total fat mass, lean mass, BMI, and waist circumference variables throughout the follow-up period and the 917 participants who had at least two ST and PA variables and complete total fat mass, lean mass, BMI, and waist circumference variables throughout the follow-up period (Table 1 and Supplementary Tables 1, 2, 10, and 11).

In the confounding-based analyses, cumulative ST was directly associated with increased BMI, waist circumference, total fat mass, trunk fat mass, and lean mass, after full adjustments for cardiometabolic and lifestyle factors including LPA and MVPA (Table 4). Cumulative LPA was inversely associated with increased total fat mass, trunk fat mass, and lean mass. Cumulative MVPA was inversely associated with increased BMI, waist circumference, total fat mass, trunk fat mass, and lean mass (Table 4). Persistent MVPA of 40 – <60 min/day and ≥60 min/day from ages 11 to 24 years were inversely associated with increased waist circumference, total fat mass, and trunk fat mass (Table 4). In both males and females, cumulative MVPA was inversely associated with increased BMI, waist circumference, total fat mass, and trunk fat mass (Table 5).

### The reduced cohort of 917 participants

Among 917 participants who had at least two time-point measures of movement behaviour and complete body composition at all three time-point measurements, cumulative ST was directly associated with increased BMI, total fat mass, and lean mass but decreased waist circumference cumulatively measured from ages 11–24 years, after full adjustments for cardiometabolic and lifestyle factors including LPA and MVPA (Supplementary Table 3). Cumulative ST was directly associated with increased trunk fat mass which significantly attenuated after adjusting for LPA and MVPA (Supplementary Table 3). Cumulative LPA was inversely associated with increased total and trunk fat mass, and lean mass but directly associated with increased waist circumference (Supplementary Table 3). Cumulative MVPA was inversely associated with increased lean mass but the inverse associations with

**Table 1 | Descriptive characteristics of 6059 participants who had at least one time-point measure of movement behaviour and dual-energy Xray absorptiometry-measured fat mass and lean mass at either 11, 15, or 24 years clinic visits**

| Age at clinic visits/follow-up | 11 years | | | 15 years | | | 24 years | | |
|---|---|---|---|---|---|---|---|---|---|
| Variables | Male (n = 2881) | Female (n = 3178) | P-value | Male (n = 2881) | Female (n = 3178) | P-value | Male (n = 2881) | Female (n = 3178) | P-value |
| **Anthropometry** | | | | | | | | | |
| Age at clinic visit (years), mean (SD) | 11.74 (0.23) | 11.74 (0.24) | 0.340 | 15.41 (0.27) | 15.44 (0.30) | 0.019 | 24.53 (0.78) | 24.44 (0.78) | 0.003 |
| Height (m), mean (SD) | 1.50 (0.07) | 1.51 (0.07) | <0.0001 | 1.74 (0.08) | 1.65 (0.06) | <0.0001 | 1.80 (0.07) | 1.66 (0.06) | <0.0001 |
| *Weight (kg) | 40.70 (9.8) | 42.0 (12.2) | <0.0001 | 62.45 (11.0) | 57.40 (10.6) | <0.0001 | 79.20 (16.02) | 64.50 (16.45) | <0.0001 |
| **Body composition** | | | | | | | | | |
| *Total fat mass (kg) | 8.05 (6.46) | 10.89 (7.35) | <0.0001 | 8.26 (5.99) | 17.07 (8.42) | <0.0001 | 18.42 (10.80) | 21.34 (11.34) | <0.0001 |
| *Trunk fat mass (kg) | 3.01 (2.82) | 4.44 (3.61) | <0.0001 | 3.68 (2.84) | 7.76 (4.29) | <0.0001 | 9.18 (6.40) | 9.51 (6.62) | <0.0001 |
| *Lean mass (kg) | 29.67 (5.09) | 28.82 (6.11) | <0.0001 | 50.24 (8.19) | 37.01 (4.97) | <0.0001 | 57.05 (10.16) | 41.02 (6.40) | <0.0001 |
| Lean mass/fat mass ratio, mean (SD) | 4.01 (2.25) | 2.80 (1.29) | <0.0001 | 6.33 (3.47) | 2.29 (0.93) | <0.0001 | 3.25 (1.33) | 1.89 (0.63) | <0.0001 |
| *Body mass index (kg/m$^2$) | 17.88 (3.10) | 18.32 (3.80) | <0.0001 | 20.27 (2.79) | 20.95 (3.69) | <0.0001 | 24.46 (4.40) | 23.23 (5.42) | 0.021 |
| *Waist circumference (cm) | 65.56 (9.5) | 64.90 (9.3) | 0.088 | 74.95 (8.1) | 75.05 (10.4) | 0.573 | 83.63 (11.48) | 74.6 (13.49) | <0.0001 |
| **Vascular measures** | | | | | | | | | |
| Heart rate (beat/min), mean (SD) | 74 (11) | 78 (11) | <0.0001 | 71 (12) | 77 (12) | <0.0001 | 65 (11) | 68 (10) | <0.0001 |
| Systolic blood pressure (mmHg), mean (SD) | 105 (9) | 106 (10) | <0.0001 | 126 (10) | 120 (11) | <0.0001 | 123 (11) | 112 (10) | <0.0001 |
| Diastolic blood pressure (mmHg), mean (SD) | 58 (6) | 59 (6) | <0.0001 | 68 (9) | 67 (8) | <0.0001 | 68 (8) | 66 (8) | <0.0001 |
| **Lifestyle and sociodemographic factors** | | | | | | | | | |
| Smoked in the last 30 days (n,%) | 34 (1.4)† | 85 (3.2)† | <0.0001 | 253 (12.7) | 457 (19.6) | <0.0001 | 316 (28.6) | 491 (26.9) | 0.174 |
| Family history of H-D-C-V (n,%) | 468 (29.2) | 653 (30.9) | 0.140 | NA | | | NA | | |
| Sedentary time (min/day), mean (SD) | 345 (76) | 361 (72) | <0.0001 | 460 (93) | 482 (81) | <0.0001 | 527 (83) | 523 (85) | 0.602 |
| Light physical activity (min/day), mean (SD) | 367 (62) | 363 (60) | 0.037 | 286 (71) | 270 (63) | <0.0001 | 147 (60) | 149 (54) | 0.578 |
| MVPA (min/day), mean (SD) | 68 (33) | 46 (22) | <0.0001 | 55 (30) | 41 (34) | <0.0001 | 54 (33) | 47 (28) | 0.001 |
| MVPA < 40 min/day (n,%) | 460 (16.5) | 1302 (42.6) | <0.0001 | 331 (33.2) | 695 (56.3) | <0.0001 | 100 (39.5) | 227 (47.9) | 0.011 |
| MVPA 40 – <60 min/day (n,%) | 773 (27.8) | 1040 (44.1) | <0.0001 | 276 (27.7) | 326 (26.4) | <0.0001 | 65 (25.7) | 122 (25.7) | 0.011 |
| MVPA ≥ 60 min/day (n,%) | 1551 (55.7) | 711 (23.3) | <0.0001 | 391 (39.2) | 213 (17.3) | <0.0001 | 88 (34.8) | 125 (26.4) | 0.011 |
| Ethnicity-White (n,%) | 2523 (96.1) | 2765 (96.5) | 0.262 | NA | | | NA | | |
| Maternal social economic status (n,%) | | | 0.887 | NA | | | NA | | |
| *Professional* | 79 (6.1) | 57 (4.1) | | | | | | | |
| *Managerial and technical* | 465 (35.9) | 510 (37.0) | | | | | | | |
| *Skilled non-manual* | 479 (37.0) | 532 (38.6) | | | | | | | |
| *Skilled manual* | 17 (1.3) | 28 (2.0) | | | | | | | |
| *Partly skilled* | 207 (16.0) | 205 (14.9) | | | | | | | |
| *Unskilled* | 47 (3.6) | 47 (3.4) | | | | | | | |
| **Fasting plasma metabolic indices** | **15 years** | | | **17 years** | | | **24 years** | | |
| High-density lipoprotein (mmol/L), mean (SD) | 1.21 (0.27) | 1.36 (0.30) | <0.0001 | 1.19 (0.26) | 1.35 (0.32) | <0.0001 | 1.40 (0.37) | 1.65 (0.43) | <0.0001 |
| Low-density lipoprotein (mmol/L), mean (SD) | 1.99 (0.52) | 2.17 (0.56) | <0.0001 | 2.00 (0.57) | 2.21 (0.63) | <0.0001 | 2.47 (0.77) | 2.43 (0.75) | 0.150 |
| *Triglyceride (mmol/L) | 0.73 (0.35) | 0.75 (0.37) | <0.0001 | 0.75 (0.37) | 0.74 (0.36) | 0.044 | 0.95 (0.56) | 0.80 (0.41) | <0.0001 |
| Glucose (mmol/L), mean (SD) | 5.30 (0.39) | 5.14 (0.38) | <0.0001 | 5.16 (0.68) | 4.91 (0.38) | <0.0001 | 5.48 (0.82) | 5.20 (0.52) | <0.0001 |
| *Insulin (mU/L) | 7.94 (4.71) | 9.67 (5.53) | <0.0001 | 5.73 (3.79) | 7.15 (4.31) | <0.0001 | 7.41 (4.77) | 7.71 (4.98) | <0.0001 |
| *High sensitivity C-reactive protein (mg/L) | 0.39 (0.62) | 0.32 (0.53) | 0.491 | 0.46 (0.62) | 0.65 (1.30) | <0.0001 | 0.63 (0.88) | 0.95 (1.86) | <0.0001 |

The values are means (standard deviations) and *median (interquartile range) except for lifestyle factors and ethnicity. Differences between sexes were tested using Student's t test for normally distributed continuous variables, Mann–Whitney U test for skewed continuous variables, Chi-square test for dichotomous variable, and analysis of covariance for multicategory variable. A 2-sided P-value < 0.05 is considered statistically significant.

†Smoking status at age 13 years since there was no smoking data at age 11 years clinic visit.

H-D-C-V hypertension/diabetes/high cholesterol/vascular disease, MVPA moderate-to-vigorous physical activity, NA not available/applicable, p-value for sex differences.

Temporal causal longitudinal associations between moderate-to-vigorous physical activity (MVPA) and total body fat mass

**Fig. 1 | Mean trajectories of movement behaviours from ages 11 through 24 years in 6059 participants and temporal causal longitudinal associations between moderate-to-vigorous physical activity and total fat mass with the mediating effect of cumulative fasting insulin.** Trajectory data are presented as mean values ± SD. (All participants, $n = 6059$; Male, $n = 2881$; Female, $n = 3178$). In the temporal analysis, the blue arrow denotes autoregressive associations while red arrow denotes cross-lagged associations. Mediation structural equation model estimating natural direct and indirect effects of the relationship between moderate-to-vigorous physical activity and total fat mass with the mediating effect of insulin in 917 participants was adjusted for sex, family history of hypertension/diabetes/high cholesterol/vascular disease, and socioeconomic status, in addition to time-varying covariates such as age, high sensitivity C-reactive protein, heart rate, systolic blood pressure, smoking status, sedentary time, light physical activity, high-density lipoprotein cholesterol, low-density. lipoprotein cholesterol, triglyceride, lean mass, and glucose. β is standardized regression co-efficient with 95% confidence interval. Two-sided $p$-value $< 0.05$ was considered statistically significant. MVPA, moderate-to-vigorous physical activity.

BMI, waist circumference, total fat mass, and trunk fat mass were not statistically significant (Supplementary Table 3). Cumulative ST, LPA, and MVPA were not associated with the lean mass-to-total fat mass ratio. Cumulative MVPA was associated with decreased total fat mass, trunk fat mass, and BMI in females but not in males (Supplementary Tables 3 and 4). The complete case analysis of the longitudinal associations of ST, LPA, and MVPA with total fat mass, trunk fat mass, and lean mass from ages 11 to 24 years were consistent in the unadjusted model (Supplementary Table 9).

**Mediating or suppressing effects of glucose, insulin, and lipids in the longitudinal associations of ST, LPA, and MVPA with total and trunk fat mass and lean mass**

In 917 participants with at least two time-point movement behaviour measures and three time-point body composition measures during growth from ages 11–24 years, cumulative glucose had no statistically significant mediating effect on the direct associations of cumulative ST with increased total fat mass, trunk fat mass and lean mass, while cumulative lean mass partially mediated (4.4%) the positive associations of ST with total fat mass (Supplementary Table 5 and Fig. 2). Cumulative increased insulin and low-density lipoprotein cholesterol partially suppressed (3.5–7.5%) the positive associations of increased ST with total and trunk fat mass (Supplementary Table 5 and Fig. 2). Cumulative glucose and lipid indices had no mediating effect on the inverse associations of cumulative LPA with increased total fat mass, trunk fat mass and lean mass (Supplementary Table 6 and Fig. 3). Cumulative insulin had a partial mediation effect (5.3–5.7%) on the associations of cumulative LPA with decreased total fat mass and trunk fat mass (Fig. 3). Cumulative high-sensitivity C-reactive protein had a partial mediation effect (7.6%) on the associations of cumulative LPA with decreased total fat mass (Supplementary Table 6). Cumulative glucose had a partial suppression effect (5%) on the associations of cumulative MVPA with decreased total fat mass and trunk fat mass and a partial mediating effect (19.5%) on the positive associations of MVPA with increased lean mass (Fig. 4). Cumulative insulin had a partial mediation effect (8.7%) on the associations of cumulative MVPA with decreased trunk fat mass (Fig. 4). Total fat mass partially suppressed (14.6%) the positive associations of cumulative MVPA with lean mass, while insulin partially mediated (8.5%) the associations of MVPA with decreased total fat mass (Fig. 1 and Supplementary Table 7). For sensitivity analyses, additional mediation analyses were conducted for available blood samples measured only at 15 and 24 years with MVPA and fat mass at 15 and 24 years. The results of the two follow-up time-point analyses were consistent with the above three time-point analyses, except that increased low-density lipoprotein cholesterol from 15 to 24 years mediated the associations between MVPA and decreased total fat mass from ages 15 to 24 years by 12.8% (Supplementary Table 8).

In the mediation analyses among 2457 participants with at least one time-point movememt behaviour measure and three time-point body composition measures during growth from ages 11–24 years, cumulative glucose, insulin, high-density lipoprotein cholesterol, low-density lipoprotein cholesterol, and triglyceride, had no mediating effect on the positive associations of cumulative ST with increased total fat mass (Supplementary Table 12). Cumulative lean mass and high-sensitivity C-reactive protein partially mediated (2.3–3.9%) the positive associations of ST with total fat mass (Supplementary Table 12). Cumulatively increased insulin, high-density lipoprotein cholesterol, low-density lipoprotein cholesterol, and triglyceride, had no mediating effect on the associations of cumulative LPA with decreased total fat mass. Cumulative increased high-sensitivity C-reactive protein partially mediated (3.7%) the associations of LPA with decreased total fat mass (Supplementary Table 12). Cumulative increased high-density lipoprotein cholesterol, lean

**Table 2 | Longitudinal associations of cumulative sedentary time and physical activity with body composition from ages 11 through 24 years (n = 6059)**

| N = 6059 | Body mass index (z-score) | | Waist circumference (z-score) | | Trunk fat mass (z-score) | |
|---|---|---|---|---|---|---|
| | β (95% CI) | p-value | β (95% CI) | p-value | β (95% CI) | p-value |
| *Continuous cumulative predictor variables from ages 11–24 years* | | | | | | |
| Sedentary Time (min/day) | | | | | | |
| Model 1 | 0.142 (0.135–0.150) | <0.0001 | 0.144 (0.134–0.153) | <0.0001 | 0.131 (0.123–0.138) | <0.0001 |
| Model 2 | 0.027 (0.021–0.032) | <0.0001 | −0.011 (−0.017−−0.006) | <0.001 | 0.026 (0.020–0.032) | <0.0001 |
| Model 3 | 0.020 (0.015–0.025) | <0.001 | −0.019 (−0.024−−0.014) | <0.001 | 0.020 (0.014–0.025) | <0.001 |
| Model 4 | 0.019 (0.014–0.024) | <0.001 | −0.017 (−0.022−−0.012) | <0.001 | 0.017 (0.012–0.023) | <0.001 |
| Light Physical Activity (min/day) | | | | | | |
| Model 1 | −0.162 (−0.170−−0.155) | <0.0001 | −0.215 (−0.225−−0.204) | <0.0001 | −0.149 (−0.156−−0.141) | <0.0001 |
| Model 2 | −0.035 (−0.040−−0.030) | <0.0001 | −0.031 (−0.037−−0.026) | <0.0001 | −0.033 (−0.039−−0.028) | <0.0001 |
| Model 3 | −0.031 (−0.036−−0.025) | <0.0001 | −0.035 (−0.041−−0.030) | <0.0001 | −0.029 (−0.035−−0.024) | <0.0001 |
| Model 4 | −0.030 (−0.035−−0.025) | <0.0001 | −0.036 (−0.042−−0.031) | <0.0001 | −0.028 (−0.034−−0.023) | <0.0001 |
| Moderate-to-Vigorous Physical Activity (min/day) | | | | | | |
| Model 1 | −0.048 (−0.057−−0.039) | <0.0001 | −0.009 (−0.019−0.001) | 0.076 | −0.061 (−0.070−−0.051) | <0.0001 |
| Model 2 | −0.014 (−0.020−−0.008) | <0.001 | 0.019 (0.014–0.024) | <0.001 | −0.027 (−0.034−−0.020) | <0.001 |
| Model 3 | −0.011 (−0.017−−0.005) | <0.001 | 0.018 (0.013–0.023) | <0.001 | −0.025 (−0.031−−0.018) | <0.001 |
| Model 4 | −0.010 (−0.015−−0.004) | <0.001 | 0.019 (0.014–0.024) | <0.001 | −0.024 (−0.030−−0.017) | <0.001 |
| *Categorical cumulative predictor variable from ages 11–24 years* | | | | | | |
| Moderate-to-Vigorous Physical Activity ( < 40 min/day as reference) | | | | | | |
| 40 – < 60 min/day | −0.011 (−0.026–0.003) | 0.117 | −0.007 (−0.019–0.006) | 0.288 | −0.053 (−0.068−−0.038) | <0.001 |
| ≥60 min/day | −0.020 (−0.036−−0.005) | 0.011 | 0.012 (−0.002–0.026) | 0.081 | −0.097 (−0.114−−0.080) | <0.0001 |

| N = 6059 | Total fat mass (z-score) | | Lean mass (z-score) | | Lean mass/fat mass ratio (z-score) | |
|---|---|---|---|---|---|---|
| | β (95% CI) | p-value | β (95% CI) | p-value | β (95% CI) | p-value |
| *Continuous cumulative predictor variables from ages 11–24 years* | | | | | | |
| Sedentary Time (min/day) | | | | | | |
| Model 1 | 0.120 (0.113–0.128) | <0.0001 | 0.225 (0.217–0.233) | <0.0001 | −0.064 (−0.593–0.465) | 0.813 |
| Model 2 | 0.029 (0.023–0.035) | <0.0001 | 0.040 (0.035–0.045) | <0.0001 | −0.071 (−0.735–0.593) | 0.833 |
| Model 3 | 0.022 (0.016–0.027) | <0.001 | 0.032 (0.027–0.037) | <0.0001 | −0.189 (−0.814–0.437) | 0.554 |
| Model 4 | 0.019 (0.013–0.024) | <0.001 | 0.033 (0.028–0.038) | <0.0001 | −0.174 (−0.776–0.429) | 0.572 |
| Light Physical Activity (min/day) | | | | | | |
| Model 1 | −0.140 (−0.148−−0.133) | <0.0001 | −0.237 (−0.245−−0.229) | <0.0001 | −0.330 (−0.933–0.273) | 0.283 |
| Model 2 | −0.038 (−0.044−−0.032) | <0.0001 | −0.042 (−0.048−−0.037) | <0.0001 | −0.490 (−1.083–0.103) | 0.105 |
| Model 3 | −0.033 (−0.039−−0.028) | <0.0001 | −0.035 (−0.040−−0.031) | <0.0001 | −0.531 (−1.078–0.015) | 0.057 |
| Model 4 | −0.032 (−0.038−−0.027) | <0.0001 | −0.036 (−0.041−−0.031) | <0.0001 | −0.537 (−1.072−−0.001) | 0.050 |
| Moderate-to-Vigorous Physical Activity (min/day) | | | | | | |
| Model 1 | −0.056 (−0.065−−0.046) | <0.0001 | −0.055 (−0.067−−0.044) | <0.0001 | 0.081 (−0.439–0.602) | 0.760 |
| Model 2 | −0.031 (−0.038−−0.024) | <0.001 | 0.000 (−0.006–0.005) | 0.902 | 0.125 (−0.428–0.677) | 0.659 |
| Model 3 | −0.028 (−0.035−−0.021) | <0.001 | 0.004 (−0.001–0.010) | 0.110 | 0.118 (−0.396–0.623) | 0.653 |
| Model 4 | −0.027 (−0.034−−0.020) | <0.001 | 0.006 (0.000–0.011) | 0.038 | 0.138 (−0.365–0.641) | 0.591 |
| *Categorical cumulative predictor variable from ages 11–24 years* | | | | | | |
| Moderate-to-Vigorous Physical Activity (<40 min/day as reference) | | | | | | |
| 40 – < 60 min/day | −0.048 (−0.063−−0.033) | <0.001 | 0.012 (−0.001–0.025) | 0.061 | 0.809 (−1.009–2.627) | 0.383 |
| ≥60 min/day | −0.090 (−0.107−−0.073) | <0.0001 | 0.017 (0.002–0.032) | 0.025 | 1.366 (−0.218–2.951) | 0.091 |

For continuous variable analyses, model 1 was unadjusted. Model 2 was adjusted for sex, family history of hypertension/diabetes/high cholesterol/vascular disease, socioeconomic status, and other time-varying covariates measured at both baseline and follow-up such as age, low-density lipoprotein cholesterol, triglyceride, high sensitivity C-reactive protein, high-density lipoprotein cholesterol, heart rate, systolic blood pressure, glucose, insulin, smoking status, and fat mass or lean mass, depending on the outcome. Model 3 was an additional adjustment for sedentary time (LPA and MVPA model) or light physical activity (Sedentary time model). Model 4 was an additional adjustment for light physical activity (MVPA model) or moderate-to-vigorous physical activity (Sedentary time and LPA model). For categorical predictor variable analyses, all the above-listed covariates were adjusted for in one model. Skewed covariates were logarithmically transformed. Standardized regression coefficients (**β**) were computed from the generalized linear mixed-effect model for repeated measures, direct effect estimates are presented; CI, confidence interval. A 2-sided *P*-value < 0.05 is considered statistically significant. Multiple testing was corrected with Sidak correction. Multiple imputations were used to account for missing variables. For continuous variable predictors (ST, LPA and MVPA), a 1-standard deviation change is associated with a 1-standard deviation change in the outcome. For categorical variable predictor (MVPA), time spent in a category in relation to the reference is associated with 1-standard deviation change in the outcome.

mass, and glucose partially suppressed (3–3.4%) the associations of MVPA with decreased total fat mass (Supplementary Table 12). Cumulatively increased insulin partially mediated (7.4%) the associations of MVPA with decreased total fat mass from ages 11 to 24 years (Supplementary Table 12). These results were consistent in the sensitivity mediational analyses conducted for available blood samples measured only at 15 and 24 years with MVPA and fat mass at 15 and 24 years (Supplementary Table 13).

**Table 3 | Sex-specific longitudinal associations of cumulative sedentary time and physical activity with body composition from ages 11 through 24 years of 6059 participants (Male; n = 2881; Female; n = 3178)**

| | Body mass index (z-score) | | Waist circumference (z-score) | | Trunk fat mass (z-score) | |
|---|---|---|---|---|---|---|
| | β (95% CI) | p-value | β (95% CI) | p-value | β (95% CI) | p-value |
| *Male (n = 2881)* | | | | | | |
| *Continuous cumulative predictor variables from ages 11–24 years* | | | | | | |
| ST (min/day) | 0.012 (0.005–0.020) | 0.001 | −0.013 (−0.021−−0.006) | <0.001 | −0.004 (−0.013–0.005) | 0.400 |
| LPA (min/day) | −0.026 (−0.033−−0.018) | <0.001 | −0.032 (−0.040−−0.023) | <0.001 | −0.012 (−0.021−−0.021) | 0.012 |
| MVPA (min/day) | −0.019 (−0.026−−0.011) | <0.001 | 0.002 (−0.005–0.009) | 0.628 | −0.026 (−0.035−−0.017) | <0.001 |
| *Categorical cumulative predictor variable from ages 11–24 years* | | | | | | |
| Moderate-to-Vigorous Physical Activity (< 40 min/day as reference) | | | | | | |
| 40 – < 60 min/day | −0.038 (−0.062–0.015) | 0.001 | −0.008 (−0.029–0.014) | 0.482 | −0.027 (−0.055–0.000) | 0.051 |
| ≥60 min/day | −0.071 (−0.095–0.046) | <0.001 | 0.000 (−0.022–0.022) | 0.980 | −0.037 (−0.066–0.008) | 0.012 |
| *Female (n = 3178)* | | | | | | |
| *Continuous cumulative predictor variables from ages 11–24 years* | | | | | | |
| ST (min/day) | 0.025 (0.018–0.032) | <0.001 | −0.020 (−0.027−−0.014) | <0.001 | 0.030 (0.024–0.036) | <0.0001 |
| LPA (min/day) | −0.028 (−0.035−−0.021) | <0.001 | −0.033 (−0.040−−0.026) | <0.0001 | −0.032 (−0.038−−0.026) | <0.0001 |
| MVPA (min/day) | −0.005 (−0.014–0.003) | 0.224 | 0.019 (0.012–0.012) | <0.001 | −0.016 (−0.023−−0.008) | <0.001 |
| *Categorical cumulative predictor variable from ages 11–24 years* | | | | | | |
| Moderate-to-Vigorous Physical Activity (< 40 min/day as reference) | | | | | | |
| 40 – < 60 min/day | −0.001 (−0.019–0.017) | 0.893 | −0.031 (−0.046–0.016) | <0.001 | −0.046 (−0.062−−0.031) | <0.001 |
| ≥60 min/day | −0.009 (−0.019–0.017) | 0.433 | −0.041 (−0.060–0.023) | <0.001 | −0.077 (−0.096–0.058) | <0.001 |
| | **Total fat mass (z-score)** | | **Lean mass (z-score)** | | **Lean mass/fat mass ratio (z-score)** | |
| *Male (n = 2881)* | | | | | | |
| *Continuous cumulative predictor variables from ages 11–24 years* | | | | | | |
| ST (min/day) | 0.001 (−0.009–0.010) | 0.907 | 0.049 (0.042–0.057) | <0.0001 | −0.052 (−1.405–1.300) | 0.940 |
| LPA (min/day) | −0.018 (−0.028−−0.009) | <0.001 | −0.057 (−0.065−−0.050) | <0.0001 | −1.472 (−2.704−−0.241) | 0.019 |
| MVPA (min/day) | −0.028 (−0.038−−0.019) | <0.001 | 0.006 (−0.002–0.013) | 0.160 | 0.337 (−0.634–1.309) | 0.496 |
| *Categorical cumulative predictor variable from ages 11–24 years* | | | | | | |
| Moderate-to-Vigorous Physical Activity (< 40 min/day as reference) | | | | | | |
| 40 – < 60 min/day | −0.071 (−0.099−−0.044) | <0.001 | 0.014 (−0.010–0.038) | 0.253 | 0.607 (−4.226–5.440) | 0.805 |
| ≥60 min/day | −0.141 (−0.170−−0.112) | <0.0001 | 0.046 (0.021–0.071) | <0.001 | 0.975 (−2.793–4.743) | 0.612 |
| *Female (n = 3178)* | | | | | | |
| *Continuous cumulative predictor variables from ages 11–24 years* | | | | | | |
| ST (min/day) | 0.026 (0.020–0.032) | <0.0001 | 0.029 (0.024–0.034) | <0.0001 | −0.247 (−0.534–0.040) | 0.091 |
| LPA (min/day) | −0.035 (−0.041−−0.029) | <0.0001 | −0.026 (−0.031−−0.021) | <0.0001 | −0.150 (−0.578–0.277) | 0.490 |
| MVPA (min/day) | −0.015 (−0.022−−0.007) | <0.001 | 0.009 (0.003–0.015) | 0.006 | 0.309 (−0.149–0.767) | 0.186 |
| *Categorical cumulative predictor variable from ages 11–24 years* | | | | | | |
| Moderate-to-Vigorous Physical Activity (< 40 min/day as reference) | | | | | | |
| 40 – < 60 min/day | −0.050 (−0.066−−0.035) | <0.001 | 0.028 (0.015–0.041) | <0.001 | 0.714 (−0.552–1.979) | 0.269 |
| ≥60 min/day | −0.090 (−0.108−−0.071) | <0.0001 | 0.045 (0.029–0.061) | <0.001 | 1.200 (−0.250–2.650) | 0.105 |

Model was adjusted for family history of hypertension/diabetes/high cholesterol/vascular disease and socioeconomic status, and time-varying covariates measured at both baseline and follow-up such as age, low-density lipoprotein cholesterol, triglyceride, high sensitivity C-reactive protein, high-density lipoprotein cholesterol, heart rate, systolic blood pressure, glucose, insulin, smoking status, and fat mass, lean mass, depending on outcome with additional adjustments for both sedentary time (ST) and light physical activity (LPA) or moderate-to-vigorous physical activity (MVPA) depending on the predictor. Skewed covariates were logarithmically transformed. Standardized regression coefficients (*β*) were computed from generalized linear mixed-effect model for repeated measures, direct effect estimates are presented; CI, confidence interval. A 2-sided P-value < 0.05 is considered statistically significant. Multiple testing was corrected with Sidak correction. Multiple imputations were used to account for missing variables. A 1-standard deviation change in ST, LPA, and MVPA is associated with a 1-standard deviation change in the outcome. For categorical variable predictor (MVPA), time spent in a category in relation to the reference is associated with 1-standard deviation change in the outcome.

**Temporal causal (cross-lagged) and inter-relational (auto-regressive) associations of ST, LPA, and MVPA with total fat mass and lean mass**

ST, LPA, MVPA, total fat mass, and lean mass at age 11 years were directly associated with their individual variables at age 15 years; however, only MVPA, total fat mass, and lean mass at age 15 years were directly associated with their individual variables at age 24 years (Table 6 and Fig. 1). There were no bidirectional relationships between movement behaviours and body composition either at 11–15 years or 15–24 years observation period (Table 6 and Fig. 1). Higher total fat mass at 11 years was associated with lower MVPA at 15 years, but higher

MVPA at 15 years was associated with lower total fat mass at 24 years (Table 6 and Fig. 1). Higher lean mass at 11 years was associated with higher ST at 15 years, but higher ST at 15 years was associated with lower lean mass at 24 years (Table 6). Higher MVPA at 15 years was associated with higher lean mass at 24 years. All other relationships were not statistically significant (Table 6).

**Compositional data analysis of the longitudinal associations of ST, LPA, and MVPA with body composition in 6059 participants**

Cumulative ST relative to LPA and MVPA was associated with increased total fat mass and trunk fat mass (Supplementary Table 14). Cumulative

**Table 4 | Longitudinal associations of cumulative sedentary time and physical activity with body composition from ages 11 through 24 years of 2457 participants with at least one time-point measure of movement behaviour and three time-point measures of body composition (n = 2457)**

| N = 2457 | Body mass index (z-score) | | Waist circumference (z-score) | | Trunk fat mass (z-score) | |
|---|---|---|---|---|---|---|
| | β (95% CI) | p-value | β (95% CI) | p-value | β (95% CI) | p-value |
| *Continuous cumulative predictor variables from ages 11–24 years* | | | | | | |
| Sedentary Time (min/day) | | | | | | |
| *Model 1* | 0.235 (0.224–0.247) | <0.0001 | 0.240 (0.228–0.253) | <0.0001 | 0.225 (0.213–0.237) | <0.0001 |
| *Model 2* | 0.020 (0.014–0.027) | <0.001 | 0.010 (0.002–0.018) | 0.020 | 0.018 (0.011–0.026) | <0.001 |
| Light Physical Activity (min/day) | | | | | | |
| *Model 1* | −0.269 (−0.280–−0.258) | <0.0001 | −0.258 (−0.270–−0.245) | <0.0001 | −0.261 (−0,273–−0.250) | <0.0001 |
| *Model 2* | −0.004 (−0.011–0.003) | 0.231 | −0.006 (−0.015–0.002) | 0.141 | −0.012 (−0.020–−0.004) | 0.003 |
| Moderate-to-Vigorous Physical Activity (min/day) | | | | | | |
| *Model 1* | −0.052 (−0.067–−0.038) | <0.001 | −0.081 (−0.096–−0.096) | <0.0001 | −0.063 (−0.079–−0.047) | <0.001 |
| *Model 2* | −0.010 (−0.018–−0.003) | 0.005 | −0.015 (−0.024–−0.006) | <0.001 | −0.022 (−0.032–−0.012) | <0.001 |
| *Categorical cumulative predictor variable from ages 11–24 years* | | | | | | |
| Moderate-to-Vigorous Physical Activity (<40 min/day as reference) | | | | | | |
| *40 – < 60 min/day* | −0.006 (−0.024–0.012) | 0.501 | −0.031 (−0.054–−0.008) | 0.008 | −0.029 (−0.050–−0.009) | 0.005 |
| *≥60 min/day* | −0.021 (−0.037–−0.004) | 0.018 | −0.034 (−0.055–−0.012) | 0.002 | −0.050 (−0.070–−0.031) | <0.001 |
| N = 2457 | Total fat mass (z-score) | | Lean mass (z-score) | | Lean mass/fat mass ratio (z-score) | |
| | β (95% CI) | p-value | β (95% CI) | p-value | β (95% CI) | p-value |
| *Continuous cumulative predictor variables from ages 11–24 years* | | | | | | |
| Sedentary Time (min/day) | | | | | | |
| *Model 1* | 0.210 (0.198–0.222) | <0.0001 | 0.357 (0.344–0.370) | <0.0001 | −0.057 (−0.068–−0.045) | <0.0001 |
| *Model 2* | 0.021 (0.013–0.029) | <0.001 | 0.047 (0.040–0.054) | <0.001 | 0.002 (−0.007–0.011) | 0.660 |
| Light Physical Activity (min/day) | | | | | | |
| *Model 1* | −0.251 (−0.262–−0.239) | <0.0001 | −0.374 (−0.386–−0.362) | <0.0001 | 0.092 (0.081–0.104) | <0.0001 |
| *Model 2* | −0.013 (−0.021–−0.005) | <0.001 | −0.018 (−0.025–−0.012) | <0.001 | 0.005 (−0.004–0.014) | 0.296 |
| Moderate-to-Vigorous Physical Activity (min/day) | | | | | | |
| *Model 1* | −0.053 (−0.068–−0.037) | <0.001 | −0.104 (−0.124–−0.084) | <0.0001 | 0.005 (−0.007–0.018) | 0.401 |
| *Model 2* | −0.021 (−0.031–−0.011) | <0.001 | −0.031 (−0.039–−0.024) | <0.001 | 0.006 (−0.005–0.017) | 0.291 |
| *Categorical cumulative predictor variable from ages 11–24 years* | | | | | | |
| Moderate-to-Vigorous Physical Activity ( < 40 min/day as reference) | | | | | | |
| *40 – < 60 min/day* | −0.029 (−0.049–−0.009) | 0.005 | −0.016 (−0.035–0.003) | 0.095 | 0.023 (−0.001–0.048) | 0.060 |
| *≥60 min/day* | −0.047 (−0.067–−0.028) | <0.001 | −0.067 (−0.085–−0.049) | <0.001 | 0.016 (−0.007–0.040) | 0.174 |

For continuous variable analyses, model 1 was unadjusted. Model 2 was adjusted for sex, family history of hypertension/diabetes/high cholesterol/vascular disease, socioeconomic status, and other time-varying covariates measured at both baseline and follow-up such as age, low-density lipoprotein cholesterol, triglyceride, high sensitivity C-reactive protein, high-density lipoprotein cholesterol, heart rate, systolic blood pressure, glucose, insulin, smoking status, and fat mass or lean mass, depending on the outcome and additional adjustment for both sedentary time and light physical activity or moderate-to-vigorous physical activity depending on the predictor. For categorical predictor variable analyses, all the above-listed covariates were adjusted for in one model. Skewed covariates were logarithmically transformed. Standardized regression coefficients (β) were computed from the generalized linear mixed-effect model for repeated measures, direct effect estimates are presented; CI, confidence interval. A 2-sided P-value < 0.05 is considered statistically significant. Multiple testing was corrected with Sidak correction. Multiple imputations were used to account for missing variables. For continuous variable predictors (ST, LPA and MVPA), a 1-standard deviation change is associated with a 1-standard deviation change in the outcome For categorical variable predictor (MVPA), time spent in a category in relation to the reference is associated with 1-standard deviation change in the outcome.

LPA relative to ST and MVPA was associated with decreased total fat mass and trunk fat mass. Cumulative MVPA relative to ST and LPA was associated with decreased total fat mass and trunk fat mass. Cumulative LPA or MVPA relative to ST was associated with decreased total fat mass and trunk fat mass (Supplementary Table 14).

## Discussion

In the largest and longest follow-up study of children with objectively measured movement behaviour and DEXA-measured body composition to date, we showed that movement behaviours may be an independent and potential causal risk factor for increased fat mass from childhood through young adulthood. We thus conclude because first, in the confounding-based longitudinal analyses, we observed that cumulative ST was independently and directly associated with increased BMI, total fat mass, and trunk fat mass, whereas cumulative LPA and persistent ≥60 min/day of MVPA were associated with decreased BMI, total fat mass, and trunk fat mass. Next, in the mediation analyses, we reported that cumulative glucose had no mediating effect on the longitudinal relationships of ST, LPA, and MVPA with fat mass, however, cumulative insulin and lipids had partial mediating roles. Lastly, using cross-lagged temporal causal path analyses, we observed that higher total fat mass in childhood temporally preceded lower MVPA in adolescence, but higher MVPA in adolescence temporally preceded lower total fat mass in young adulthood.

The consistency in the direct longitudinal associations of ST and increased adiposity irrespective of model strategy and independent of LPA and MVPA strongly suggests that ST is deleterious for paediatric population health[1,3,13]. Although we did not observe a potential temporal relationship between ST and fat mass it likely exerted its influence through different pathways such as a decrease in lipoprotein lipase activity, decreased oxidative stress regulation as experimented in mice, and an increase in insulin level[8,14–16]. The mediation analyses showed that a higher insulin concentration which could reflect hyperinsulinemia was associated with higher total body fat mass.

**Table 5 | Sex-specific longitudinal associations of cumulative sedentary time and physical activity with body composition from ages 11 through 24 years of 2457 participants with at least one time-point measure of movement behaviour and three time-point measures of body composition. (Male; *n* = 964; Female; *n* = 1493)**

| | Body mass index (z-score) | | Waist circumference (z-score) | | Trunk fat mass (z-score) | |
|---|---|---|---|---|---|---|
| | *β* (95% CI) | *p*-value | *β* (95% CI) | *p*-value | *β* (95% CI) | *p*-value |
| **Male (n = 964)** | | | | | | |
| ST (min/day) | 0.008 (−0.002–0.017) | 0.102 | 0.000 (−0.012–0.013) | 0.949 | 0.007 (−0.007–0.021) | 0.309 |
| LPA (min/day) | 0.000 (−0.011–0.010) | 0.961 | 0.003 (−0.010–0.016) | 0.639 | −0.004 (−0.019–0.011) | 0.621 |
| MVPA (min/day) | −0.013 (−0.023–0.002) | 0.020 | −0.015 (−0.027–−0.004) | 0.010 | −0.022 (−0.038–−0.006) | 0.008 |
| **Female (n = 1493)** | | | | | | |
| ST (min/day) | 0.026 (0.017–0.035) | <0.001 | 0.005 (−0.005–0.016) | 0.309 | 0.020 (0.013–0.028) | <0.001 |
| LPA (min/day) | 0.000 (−0.009–0.008) | 0.938 | −0.006 (−0.016–0.005) | 0.292 | −0.009 (−0.017–−0.001) | 0.024 |
| MVPA (min/day) | −0.014 (−0.024–−0.003) | 0.011 | −0.020 (−0.033–−0.007) | 0.003 | −0.024 (−0.034–−0.014) | <0.001 |
| | **Total fat mass (z-score)** | | **Lean mass (z-score)** | | **Lean mass/fat mass ratio (z-score)** | |
| **Male (n = 964)** | | | | | | |
| ST (min/day) | 0.008 (−0.006–0.022) | 0.247 | 0.066 (0.054–0.0) | <0.0001 | 0.043 (0.026–0.061) | <0.001 |
| LPA (min/day) | −0.003 (−0.018–0.013) | 0.736 | −0.029 (−0.040–−0.018) | <0.001 | −0.020 (−0.038–−0.002) | 0.026 |
| MVPA (min/day) | −0.025 (−0.042–−0.009) | 0.003 | −0.014 (−0.026–−0.001) | 0.031 | 0.021 (0.004–0.037) | 0.015 |
| **Female (n = 1493)** | | | | | | |
| ST (min/day) | 0.020 (0.012–0.027) | <0.001 | 0.033 (0.027–0.040) | <0.0001 | −0.019 (−0.027–−0.011) | <0.001 |
| LPA (min/day) | −0.008 (−0.016–−0.001) | 0.033 | −0.015 (−0.022–−0.009) | <0.001 | 0.008 (−0.001–0.016) | 0.066 |
| MVPA (min/day) | −0.023 (−0.032–−0.014) | <0.001 | 0.000 (−0.023–0.010) | 0.979 | 0.024 (0.015–0.034) | <0.001 |

Model was adjusted for family history of hypertension/diabetes/high cholesterol/vascular disease and socioeconomic status, and time-varying covariates measured at both baseline and follow-up such as age, low-density lipoprotein cholesterol, triglyceride, high sensitivity C-reactive protein, high-density lipoprotein cholesterol, heart rate, systolic blood pressure, glucose, insulin, smoking status, and fat mass, lean mass, depending on outcome with additional adjustments for both sedentary time (ST) and light physical activity (LPA) or moderate-to-vigorous physical activity (MVPA) depending on the predictor. Skewed covariates were logarithmically transformed. Standardized regression coefficients (*β*) were computed from generalized linear mixed-effect model for repeated measures, direct effect estimates are presented; CI, confidence interval. A 2-sided *P*-value < 0.05 is considered statistically significant. Multiple testing was corrected with Sidak correction. Multiple imputations were used to account for missing variables. A 1-standard deviation change in ST, LPA, and MVPA is associated with a 1-standard deviation change in the outcome.

However, higher ST was associated with decreased insulin concentration. Evidence suggests that a 6-day period of physical inactivity reduces insulin action, through the reduction in Glut-4 levels in endurance-trained runners' skeletal muscle[10,11]. ST has been associated with increased endoplasmic reticulum oxidative stress, inflammation, and mitochondrial dysfunction, which could result in beta cell insufficiency and decreased insulin secretion[8,11,17]. It is however plausible that optimal insulin levels in relatively healthy young populations may offer minimal protection against the deleterious effect of ST on body fat since the magnitude of the suppression effect of insulin on the relationship between ST and fat mass was 7.5%.

We observed that increased ST was associated with increased fat mass via an increase in muscle mass. While the pathophysiological mechanism may not be fully understood, it is known that skeletal muscle mass regulation of lipoprotein lipase activity is impaired during physical inactivity[8,9,11]. Lipoprotein lipase influences cholesterol metabolism, the partitioning of triglyceride-derived fatty acid uptake between different tissues, downstream intracellular effects related to lipid availability, alterations in muscle glucose, and fatty acid metabolism[8,9]. Glucose had no mediating effect while low-density lipoprotein cholesterol had a partial suppression role in the relationship of ST with fat mass. A study recently reported that accelerometer-based ST from ages 11 to 15 years was not associated with adiposity at age 17 years[18]. The contrast in our findings with this study[18] may be due to longer follow-up and accounting for the dynamic changes in fat mass during growth from childhood through young adulthood. In this study, ST steadily increased from ~6 h/day in childhood to ~9 h/day by young adulthood. We observed that every 1 min/day spent in ST was associated with a 1.3 g increase in total fat mass and that both male and female children gained approximately 10 kg of fat mass during growth from childhood until young adulthood. This implies that ST potentially contributed 700 g to 1 kg of fat mass (approximately 7–10%) of the total fat mass gained during growth from childhood until young

adulthood. Among adults, >10 h/day of ST has been associated with an increased risk of incident cardiovascular events[15,19]. These findings provide answers to the American Academy of Pediatrics, the European Childhood Obesity Group's and systematic reviews and meta-analysis identified gaps in knowledge on the independent impact of ST on adiposity from childhood through young adulthood[3,4,6,7]. Of note, cumulative ST was associated with increased total and trunk fat mass in females but not in males. This might be related to males accumulating fewer minutes in ST and physiologically had lesser fat mass.

There is a substantial lack of longitudinal evidence on the intensity of accelerometer-measured PA in relation to body composition outcomes in the paediatric population, especially, for those who lack motivation or are unable to achieve the recommended guideline of at least 60 min/day of MVPA[1,5,13,20]. A meta-analysis of cross-sectional studies concluded that reallocating ST with LPA was not significantly associated with any adiposity outcomes[20]. In the present longitudinal study, cumulative LPA was associated with decreased total fat mass and trunk fat mass from childhood through young adulthood. Although this association may be partially suppressed by increased insulin levels, the paediatric population especially those with chronic diseases or mobility challenges could benefit from LPA. Obesity has been strongly associated with metabolic syndrome and our finding may suggest that an LPA-induced decrease in total and truncal adiposity may lower the risk of metabolic syndrome in the paediatric population[21]. The mediation analyses revealed that LPA may not lower fat mass via glucose or lipid pathway but partly by decreasing low-grade inflammation[22] but further experimental and mechanistic studies are warranted. Among adults, LPA has been associated with a reduced risk of all-cause mortality[23,24]. In our cohort we observed that each 1 min/day spent in LPA was associated with a 3.6 gram reduction in total fat mass which may be clinically significant[1,3]. This implies that cumulative LPA decreased total body fat mass by 950 g to 1.5 kg during growth from childhood to young adulthood, approximately 9.5–15%

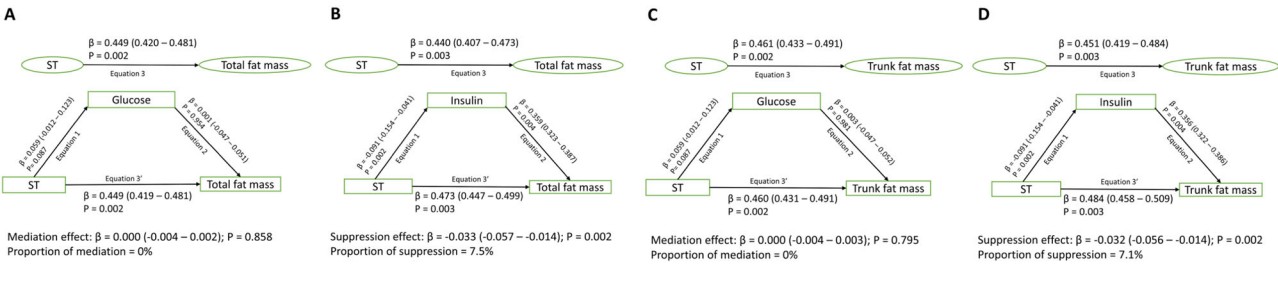

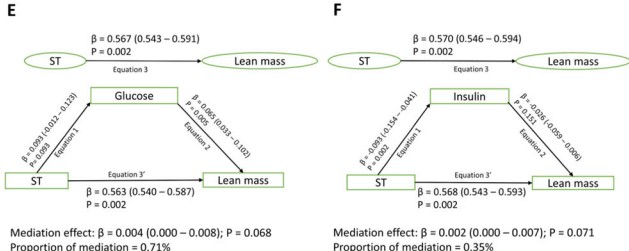

**Fig. 2 | Mediating or suppressing role of cumulative fasting plasma glucose and insulin on the longitudinal associations of cumulative sedentary time (ST) and total fat mass, trunk fat mass, and lean mass from ages 11 through 24 years of 917 participants. A** Sedentary time with total fat mass and glucose as a mediator. **B** Sedentary time with total fat mass and insulin as a mediator. **C** Sedentary time with trunk fat mass and glucose as a mediator. **D** Sedentary time with trunk fat mass and insulin as a mediator. **E** Sedentary time with lean mass and glucose as a mediator. **F** Sedentary time with lean mass and insulin as a mediator. When the magnitude of the longitudinal association between the predictor and outcome is increased upon inclusion of a third variable, a suppression is confirmed, However, when decreased it is mediation. Mediation structural equation model estimating natural direct and indirect effects was adjusted for sex, family history of hypertension/diabetes/high cholesterol/vascular disease, and socioeconomic status, in addition to time-varying covariates such as age, high-sensitivity C-reactive protein, heart rate, systolic blood pressure, smoking status, light physical activity, moderate-to-vigorous physical activity, high-density lipoprotein cholesterol, low-density. lipoprotein cholesterol, triglyceride, total fat mass, lean mass, glucose or insulin depending on the mediator and outcome. β is standardized regression coefficient. Two-sided $p$-value < 0.05 were considered statistically significant.

decrease in overall 10 kg gain in fat mass during the 13-year observation period. Exposure to LPA appears to have a higher fat mass-lowering effect than MVPA in consonance with a previous report on the longitudinal relationships between movement behaviour and inflammation but more studies across different ethnic and racial groups are needed[22]. We observed that while LPA decreased from 6 h/day in childhood to 3 h/day in young adulthood, ST increased from 6 h/day in childhood to 9 h/day in young adulthood. The significant loss in time spent in LPA was gained by ST, thus, these findings could contribute to updating future PA guidelines which currently largely focus on an average of 60 min/day of MVPA which 80% of adolescents do not meet[1,25]. A paradigm shift towards increasing and sustaining LPA of at least 3 h/day may yield several health benefits but more longitudinal evidence is warranted[22].

To accumulate at least 60 min/day of MVPA on average in children and adolescents has been recommended by the WHO, however, the guideline is largely based on iso-temporal accelerometer cross-sectional studies and a few short-term longitudinal studies[1,5]. Persistently accruing ≥60 min/day of MVPA throughout the 13-year follow-up period was associated with decreasing waist circumference, total fat mass, and trunk fat mass from childhood through young adulthood. We observed that each 1 min/day spent in MVPA was associated with a 1.3 gram reduction in total fat mass. When compared to children who persistently spent <40 min/day in MVPA, participants who persistently accumulated 40 to 59.9 min/day in MVPA had a 1.4 gram reduction in total fat mass per each minute/day of MVPA, but those who persistently spent ≥60 min/day of MVPA from childhood through young adulthood had a 2.8 gram reduction in total fat mass per each minute/day of MVPA. Time spent in MVPA including at least 60 min/day of MVPA during growth from childhood through young adulthood was associated with 70 to 170 g (approximately 0.7–1.7%)

reduction in 10 kg gain in total body fat mass from childhood through young adulthood. Thus, MVPA intensity of ≥60 min/day may be necessary to significantly attenuate total and truncal adiposity in the long term. MVPA may lower fat mass via decreased low-density lipoprotein cholesterol and insulin pathways. Earlier, in the same cohort, a 2-year change in MVPA as a continuous variable measured between ages 11.8 and 13.9 years was significantly and inversely associated with a 2-year change in total fat mass, without accounting for changes in plasma lipid, glucose, insulin, and inflammation[26].

Importantly, MVPA in mid-adolescence may potentially lower total fat mass in young adulthood but increase lean mass via an increased glucose pathway. There was a significant drop in MVPA in mid-adolescence compared to childhood and young adulthood resulting in a U or J-shaped curve. This drop in MVPA in mid-adolescence may be related to a high total fat mass in childhood which was associated with reduced MVPA by mid-adolescence in the temporal causal path, suggesting that obesity may decrease a child's interest in participating in MVPA[1,5,13]. Hence, at least 60 min of MVPA before age 11 years should be highly prioritized to prevent childhood obesity, since a higher BMI z-score in childhood has been associated with an increased risk of premature death by age 40 years[1,5,13,27]. Another explanation for the U or J-shaped curve may be due to different cutpoints used in adolescence (MVPA > 2296 counts per minute) and young adulthood (MVPA > 2020 counts per minute), since the average count per minute in the entire cohort steadily declined across the 13-year follow-up period[28–30]. While the results in 917 participants with at least two time-point valid accelerometer measures suggested that MVPA was associated with lower total fat mass in females but not in males, the analyses using a larger sample size of both 2457 and 6059 participants with less strict inclusion criteria of at least one time-point accelerometer measure indicates that increased MVPA was associated

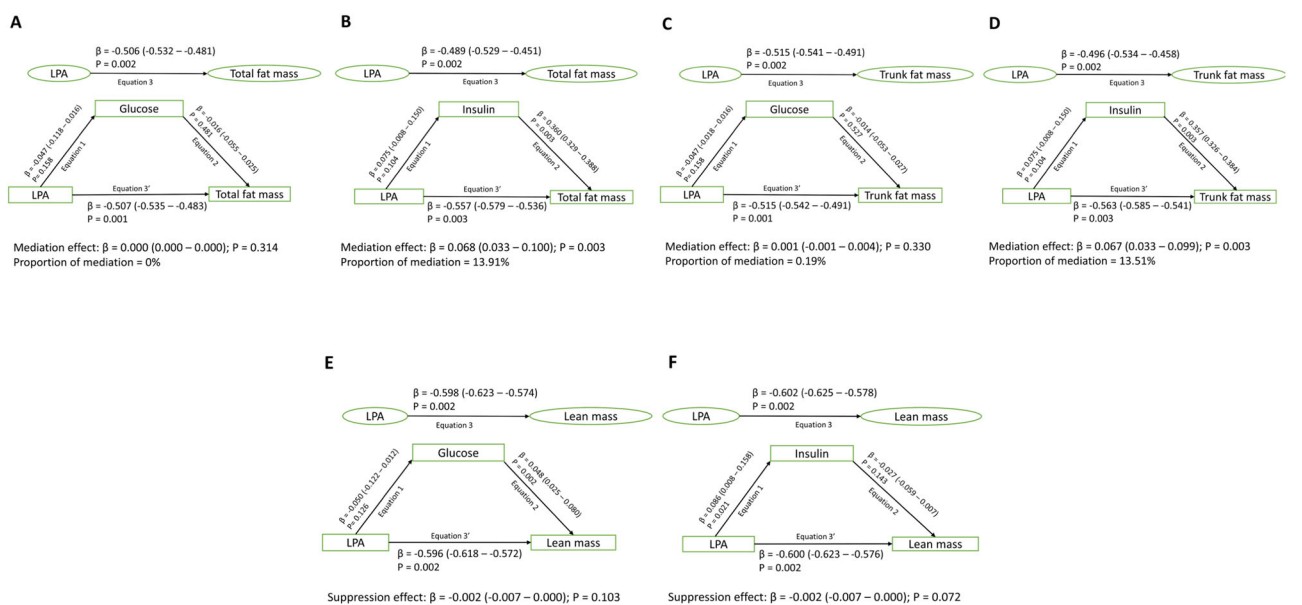

**Fig. 3 | Mediating or suppressing role of cumulative fasting plasma glucose and insulin on the longitudinal associations of cumulative light physical activity (LPA) with cumulative total fat mass, trunk fat mass, and lean mass from ages 11 through 24 years of 917 participants. A** Light physical activity with total fat mass and glucose as a mediator. **B** Light physical activity with total fat mass and insulin as a mediator. **C** Light physical activity with trunk fat mass and glucose as a mediator. **D** Light physical activity with trunk fat mass and insulin as a mediator. **E** Light physical activity with lean mass and glucose as a mediator. **F** Light physical activity with lean mass and insulin as a mediator. When the magnitude of the longitudinal association between the predictor and outcome is increased upon inclusion of a third variable, a suppression is confirmed, However, when decreased it is mediation. Mediation structural equation model estimating natural direct and indirect effects was adjusted for sex, family history of hypertension/diabetes/high cholesterol/vascular disease, and socioeconomic status, in addition to time-varying covariates such as age, high-sensitivity C-reactive protein, heart rate, systolic blood pressure, smoking status, sedentary time, moderate-to-vigorous physical activity, high-density lipoprotein cholesterol, low-density. lipoprotein cholesterol, trigly-ceride, total fat mass, lean mass, glucose or insulin depending on the mediator and outcome. β is standardized regression co-efficient. Two-sided *p*-value < 0.05 were considered statistically significant.

with lower total fat mass in both males and females. This disparity in findings is likely due to a lower sample size, i.e 334 males of 917 participants in comparison with 964 males of 2457 participants or 2881 males of 6059 participants. These consistent sex-specific findings suggest that PA is beneficial to both males and females irrespective of the exposure time to PA which might depend on inherent biological characteristics.

The ALSPAC dataset provides an extensive array of gold-standard and repeated measures of movement behaviours, body composition, and covariates throughout the follow-up period in a large paediatric population. Using advanced statistical models, we tested the likelihood of reverse causality, temporality, potential causal explanatory pathway, and consistency of the findings for the first time in a paediatric population. Our findings aptly fill several knowledge gaps recently identified in movement behaviour research and would be useful in updating future PA guidelines[1,3–7]. On the other hand, our study has some limitations. Our participants were mostly White; therefore, we are unable to generalize our findings to other racial and ethnic groups. Moreover, as with all observational studies, residual biases due to unmeasured confounders may distort observed associations such as the unavailability of quantitative sleep variables. Also, cohort attrition could lead to bias, however, participants who lacked certain movement behaviour and body composition variables had similar characteristics to those included in the analyses. The non-availability of fasting blood samples at baseline measures of ST, PA, and fat mass at age 11 years may limit the mediation analyses. Moreover, the misalignment between data collection of blood samples at the 17-year clinic visit with accelerometer and adiposity data at either 15 or 24-year clinic visit may bias the mediation analysis results. However, we conducted additional mediating analyses only with accelerometer and adiposity data and blood samples using only 2-time point measures at ages 15 and 24 years for predictor, mediator, and outcomes, and the results were substantially similar to the original analysis involving accelerometer and adiposity data at 11, 15 and 24 years with blood samples at 15, 17 and 24 years. Furthermore, we repeated the mediation analyses in the cohort of 2457 participants who had at least one PA valid measure and the results were consistent. Also, we repeated the 15 and 24-year specific mediation analyses including only blood samples at 15 and 24 years with PA and fat mass at 15 and 24 years in the expanded cohort of 2457 participants and the results were similar to previous findings. These consistencies in findings support the validity of the mediation analyses.

In conclusion, ST increased from 6 to 9 h/day, LPA decreased from 6 to 3 h/day, and MVPA had a U-shaped increase from childhood through young adulthood. ST increase was associated with increased total fat mass and truncal fat mass while cumulative LPA and MVPA were associated with lower total fat mass and truncal fat mass. Over a 13-year follow-up period, persistent exposure to ≥60 min/day of MVPA was associated with decreased total fat mass and truncal fat mass and MVPA may potentially lower total fat mass via a decrease in insulin and low-density lipoprotein cholesterol. Each minute/day of ST was associated with 1.3 g increase in total fat mass. However, each minute/day of LPA was associated with 3.6 g decrease in total fat mass and each minute/day of MVPA was associated with 1.3 g decrease in total fat mass. LPA elicited a similar effect as MVPA and thus may be targeted in obesity prevention in children and adolescents, who are unable or unwilling to exercise. Accumulating 3-4 h/day of LPA may lower total fat mass ten times more than accumulating 60 min/day of MVPA. Higher total fat mass at age 11 years may potentially lead to lower MVPA by mid-adolescence, thus ≥60 min/day of MVPA should be prioritized in early childhood. Further experimental and intervention studies are needed to clarify the mechanisms through which movement behaviours alter body composition in a growing asymptomatic paediatric population. Future guidelines could emphasize

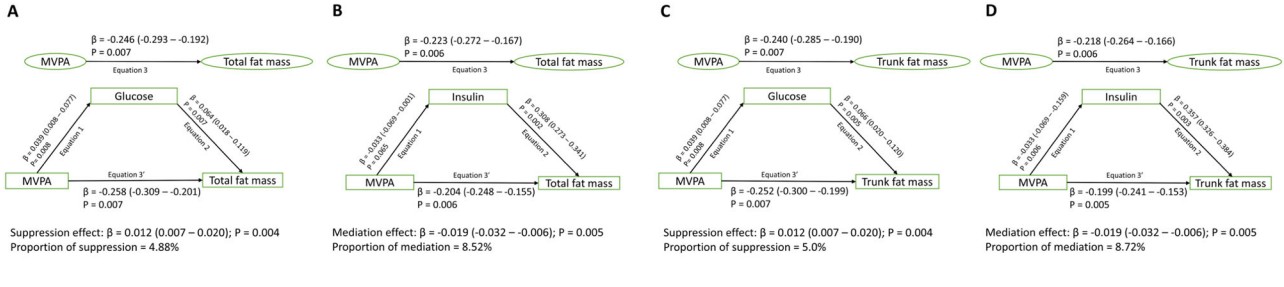

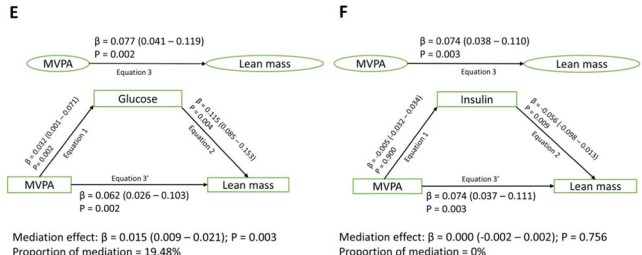

**Fig. 4 | Mediating or suppressing role of cumulative fasting plasma glucose and insulin on the longitudinal associations of cumulative moderate-to-vigorous physical activity (MVPA) with cumulative total fat mass, trunk fat mass, and lean mass from ages 11 through 24 years of 917 participants. A** Moderate-to-vigorous physical activity with total fat mass and glucose as a mediator. **B** Moderate-to-vigorous physical activity with total fat mass and insulin as a mediator. **C** Moderate-to-vigorous physical activity with trunk fat mass and glucose as a mediator. **D** Moderate-to-vigorous physical activity with trunk fat mass and insulin as a mediator. **E** Moderate-to-vigorous physical activity with lean mass and glucose as a mediator. **F** Moderate-to-vigorous physical activity with lean mass and insulin as a mediator. When the magnitude of the longitudinal

association between the predictor and outcome is increased upon inclusion of a third variable, a suppression is confirmed, However, when decreased it is mediation. Mediation structural equation model estimating natural direct and indirect effects was adjusted for sex, family history of hypertension/diabetes/high cholesterol/vascular disease, and socioeconomic status, in addition to time-varying covariates such as age, high-sensitivity C-reactive protein, heart rate, systolic blood pressure, smoking status, sedentary time, light physical activity, high-density lipoprotein cholesterol, low-density. lipoprotein cholesterol, tri-glyceride, total fat mass, lean mass, glucose or insulin depending on the mediator and outcome. β is standardized regression co-efficient. Two-sided *p*-value < 0.05 were considered statistically significant.

the importance of early LPA and MVPA intervention before mid-adolescence for optimal obesity prevention. Decreasing ST by at least 3 h/day and increasing LPA by the same amount during growth from childhood through young adulthood should be strongly recommended with specific emphasis on mid-adolescence timing.

## Methods

### Study cohort

Data were from the ALSPAC birth cohort, which investigates factors that influence childhood development and growth. Pregnant women resident in Avon, UK with expected dates of delivery between 1st April 1991 and 31st December 1992 were invited to take part in the study. Altogether 20,248 pregnancies were eligible, and 14,541 pregnancies were initially enroled. Of the initial pregnancies, there were 14,676 foetuses, 14,062 live births and 13,988 children who were alive at 1 year of age. An attempt was made to bolster the initial sample with eligible cases who had failed to join the study originally when the oldest children were approximately 7 years of age. As a result, when considering variables collected from the age of seven onwards (and potentially abstracted from obstetric notes) there are data available for more than the 14,541 pregnancies mentioned above. By the age of 24-year clinic visits, 906 new pregnancies not in the initial sample (known as Phase I enrolment) have been added, resulting in an additional 913 children being enroled (456, 262 and 195 recruited during Phases II, III and IV respectively). Thus, the total sample size for analyses using any data collected after the age of seven is therefore 15,447 pregnancies, resulting in 15,658 foetuses and 14,901 children who were alive at age 1 year. At age 7 years, regular clinic visits of the children were commenced and are still ongoing into adulthood. Using the REDCap electronic data capture

tools, study data at age 24 years clinic visit were collected and managed[31]. In this study, 6059 participants who had at least one time-point valid measure of ST and PA and one time-point DEXA body composition measure were included in the analyses. Moreover, 2457 participants had at least one time-point valid measure of ST and PA with complete DEXA measures at three time-points during the 13-year follow-up (Supplementary Fig. 1). However, 917 participants had complete measures of total fat mass, trunk fat mass and lean mass, BMI, and waist circumference at age 11, 15, and 24 years clinic visits, and at least two time-point valid ST, LPA, and MVPA measurements at either age 11, 15, or 24 years clinic visit (Supplementary Fig. 1). The excluded participants who had at least two timepoint measures of ST and PA and incomplete body composition measures during the 13-year-long follow-up study had similar characteristics to those included in the study (Supplementary Table 1). Further details regarding participant selection are presented in the supplementary information file. Ethical approval for the study was obtained from the ALSPAC Ethics and Law Committee and the Local Research Ethics Committees. Informed consent for the use of data collected via questionnaires and clinics was obtained from participants following the recommendations of the ALSPAC Ethics and Law Committee at the time[32–34]. Consent for biological samples has been collected in accordance with the Human Tissue Act (2004). All participants were asked to complete one clinic visit that lasted about 2–3 h. As compensation for participating in the study, participants received a £40 e-voucher valid at most online retailers or high street stores in the UK. Please note that the study website contains details of all the data that is available through a fully searchable data dictionary and variable search tool (http://www.bristol.ac.uk/alspac/researchers/our-data/).

**Table 6 | Auto-regressive cross-lagged temporal causal longitudinal analyses of sedentary time and physical activity in relation to total fat mass and lean mass at 11, 15, and 24 years of age in participants with at least two time-point measures of movement behaviour and three-time point measures of body composition (n = 917)**

**917 participants**

### Total fat mass (kg)

**Auto-regressive**

| | | | B | β | SE | p-value |
|---|---|---|---|---|---|---|
| ST T1 | ⇨ | ST T2 | 0.344 | 0.301 | 0.045 | <0.0001 |
| ST T2 | ⇨ | ST T3 | -0.271 | -0.240 | 0.172 | 0.114 |
| LPA T1 | ⇨ | LPA T2 | 0.311 | 0.270 | 0.044 | <0.0001 |
| LPA T2 | ⇨ | LPA T3 | 0.164 | 0.202 | 0.145 | 0.255 |
| MVPA T1 | ⇨ | MVPA T2 | 0.168 | 0.192 | 0.029 | <0.0001 |
| MVPA T2 | ⇨ | MVPA T3 | 0.825 | 0.714 | 0.146 | <0.0001 |
| Total FM T1 | ⇨ | Total FM T2 | 0.661 | 0.623 | 0.022 | <0.0001 |
| Total FM T2 | ⇨ | Total FM T3 | 0.486 | 0.675 | 0.021 | <0.0001 |

**Cross-lagged**

| | | | B | β | SE | p-value |
|---|---|---|---|---|---|---|
| ST T1 | ⇨ | Total FM T2 | 0.000 | 0.025 | 0.000 | 0.215 |
| Total FM T1 | ⇨ | ST T2 | 7.010 | 0.020 | 13.924 | 0.615 |
| ST T2 | ⇨ | Total FM T3 | 0.000 | -0.052 | 0.000 | 0.098 |
| Total FM T2 | ⇨ | ST T3 | -3.669 | -0.010 | 31.366 | 0.907 |
| LPA T1 | ⇨ | Total FM T2 | 0.000 | 0.032 | 0.000 | 0.112 |
| Total FM T1 | ⇨ | LPA T2 | 6.642 | 0.024 | 11.182 | 0.553 |
| LPA T2 | ⇨ | Total FM T3 | 0.000 | -0.009 | 0.000 | 0.782 |
| Total FM T2 | ⇨ | LPA T3 | 11.938 | 0.056 | 18.463 | 0.518 |
| MVPA T1 | ⇨ | Total FM T2 | 0.000 | 0.010 | 0.000 | 0.579 |
| Total FM T1 | ⇨ | MVPA T2 | -11.019 | -0.086 | 4.728 | 0.020 |
| MVPA T2 | ⇨ | Total FM T3 | 0.000 | -0.058 | 0.000 | 0.046 |
| Total FM T2 | ⇨ | MVPA T3 | -8.541 | -0.066 | 10.683 | 0.424 |

### Lean mass (kg)

**Auto-regressive**

| | | | B | β | SE | p-value |
|---|---|---|---|---|---|---|
| ST T1 | ⇨ | ST T2 | 0.335 | 0.294 | 0.045 | <0.0001 |
| ST T2 | ⇨ | ST T3 | -0.281 | -0.247 | 0.172 | 0.104 |
| LPA T1 | ⇨ | LPA T2 | 0.307 | 0.268 | 0.044 | <0.0001 |
| LPA T2 | ⇨ | LPA T3 | 0.123 | 0.199 | 0.149 | 0.408 |
| MVPA T1 | ⇨ | MVPA T2 | 0.170 | 0.194 | 0.029 | <0.0001 |
| MVPA T2 | ⇨ | MVPA T3 | 0.797 | 0.694 | 0.149 | <0.0001 |
| Lean mass T1 | ⇨ | Lean mass T2 | 0.477 | 0.396 | 0.155 | 0.002 |
| Lean mass T2 | ⇨ | Lean mass T3 | 0.948 | 0.873 | 0.019 | <0.0001 |

**Cross-lagged**

| | | | B | β | SE | p-value |
|---|---|---|---|---|---|---|
| ST T1 | ⇨ | Lean mass T2 | 0.000 | -0.021 | 0.000 | 0.243 |
| Lean mass T1 | ⇨ | ST T2 | 742.97 | 0.612 | 334.31 | 0.026 |
| ST T2 | ⇨ | Lean mass T3 | 0.000 | -0.040 | 0.000 | 0.026 |
| Lean mass T2 | ⇨ | ST T3 | 133.70 | 0.117 | 100.64 | 0.184 |
| LPA T1 | ⇨ | Lean mass T2 | 0.000 | 0.009 | 0.000 | 0.619 |
| Lean mass T1 | ⇨ | LPA T2 | -300.4 | -0.303 | 270.4 | 0.267 |
| LPA T2 | ⇨ | Lean mass T3 | 0.000 | -0.002 | 0.000 | 0.905 |
| Lean mass T2 | ⇨ | LPA T3 | -6.835 | -0.010 | 62.20 | 0.912 |
| MVPA T1 | ⇨ | Lean mass T2 | 0.000 | 0.022 | 0.000 | 0.169 |
| Lean mass T1 | ⇨ | MVPA T2 | 170.03 | 0.411 | 111.82 | 0.128 |
| MVPA T2 | ⇨ | Lean mass T3 | 0.000 | 0.058 | 0.000 | <0.0001 |
| Lean mass T2 | ⇨ | MVPA T3 | 7.606 | 0.019 | 34.963 | 0.828 |

Time T1, 11 years of age; Time T2, 15 years of age; Time T3, 24 years of age. B, unstandardized regression; β, standardized regression; SE, standard error; Model was adjusted for baseline age, sex, low-density lipoprotein cholesterol, insulin, triglyceride, high-sensitivity C-reactive protein, high-density lipoprotein cholesterol, heart rate, systolic blood pressure, smoking status, family history of hypertension/diabetes/high cholesterol/vascular disease, socioeconomic status, insulin, glucose with fat mass or lean mass, depending on outcome with additional adjustment for sedentary time (ST), light physical activity (LPA) or moderate-to-vigorous physical activity (MVPA) depending on the predictor. Skewed variables were logarithmically transformed before analyses. A 2-sided P-value < 0.05 is considered statistically significant. Auto-regressive cross-lagged longitudinal analyses were conducted using structural equation temporal causal path models.

## Exposures: Sedentary time and physical activity assessment

With an ActiGraph™ (LLC, Fort Walton Beach, FL, USA) accelerometer worn on the waist for 7 consecutive days, ST, LPA, and MVPA were assessed at 11- and 15-year clinic visits, whereas at 24 years movement behaviour was assessed using ActiGraph GT3X+ accelerometer device worn for four consecutive days[22,35]. Strong absolute agreement between the Actigraph™ models (intraclass correlation coefficient 0.99 (95% CI = 0.98–0.99) has been reported, thus it is acceptable to use different models within a study[29]. Providing data for at least 10 h per day (excluding sequences of 10 or more minutes with consecutive zero counts) was considered a valid day and children were only included in the analyses if they provided at least 3 valid days of recording. The devices capture movement in terms of acceleration as a combined function of frequency and intensity. Data are recorded as counts that result from summing postfiltered accelerometer values (raw data at 30 Hz) into 60-second epoch units. Data were processed using Kinesoft software, version 3.3.75 (Kinesoft), according to established protocol[28]. Activity counts per minute threshold validated in children and adolescents were used to calculate the amount of time spent; the Evenson cutpoint MVPA, >2296 counts per minute; for LPA, 100–2296 counts per minute; and for ST, 0 – <100 counts per minute was used at ages 11 and 15 years whereas, at the 24-year clinic assessment, the 2020 counts per minute Troiano cut point was used[28,30,36]. The Evenson cutpoint used in stratifying activity threshold has been suggested as the most appropriate cut point for youth having shown the best overall performance across all intensity levels[30,36]. Pearson bivariate correlations ($r^2$) between cumulative 11, 15, and 24 years ST and LPA are (−0.70), ST and MVPA (−0.33), LPA and MVPA (0.19), $p$-value < 0.001 for all. For each age clinic visit, at age 11 years, the correlations between ST and LPA are (−0.59), ST and MVPA (−0.27), LPA and MVPA (0.12), $p$-value < 0.001 for all. At age 15 years clinic visit, the correlations between ST and LPA are (−0.47), ST and MVPA (−0.35), LPA and MVPA (0.35), $p$-value < 0.001 for all. At age 24 years clinic visit, the correlations between ST and LPA are (−0.18), $p = 0.009$, ST and MVPA (−0.22), $p = 0.002$, LPA and MVPA (0.14), $p = 0.043$. These correlation matrices are similar to a meta-analysis of cross-sectional accelerometer report[37]. MVPA was classified as <40 min/day as low (reference), 40 – <60 min/day as moderate, and ≥60 min/day as high. The 40 – <60 min/day of MVPA was based on the lowest tertile cutpoint (39.95 min) for MVPA in the total population. The cut point between the middle and highest MVPA tertile was 59.8 min/day in line with the current PA guideline[1].

## Outcomes: anthropometry and body composition

Height was measured to the nearest 0.1 cm with Harpenden wall-mounted stadiometer (Holtain Ltd, Crosswell, Crymych, UK), weight to the nearest 0.1 kg was measured using (Tanita TBF-401 Model A, Tanita Corp., Tokyo, Japan electronic scale), waist circumference to the nearest 1 mm at the midpoint between the lower ribs and the pelvic bone with a flexible tape was assessed, and BMI was computed as weight in kilograms per height in metres squared[38,39]. Body composition (total fat mass, trunk fat mass, and lean mass) was assessed using DEXA scanner (GE Medical Systems, Madison, Wisconsin) at 11, 15, and 24-year clinic visits[38–41]. Repeated DEXA measurements for 122 children were performed on the same day, and the repeatability coefficient (twice the standard deviation of the difference between measurement occasions) for body fat mass was 0.5 kg[39,40,42]. Lean mass-to-total fat mass ratio was computed at all three-time points.

## Confounders and covariates: Cardiometabolic, socioeconomic, and lifestyle factors

Heart rate and systolic and diastolic blood pressure were measured with Dinamap 9301 Vital Signs Monitor (Morton Medical, London, United Kingdom) at ages 11 and 15, and Omron M6 monitor at age 24 years[38,39]. Fasting blood samples at ages 15, 17, and 24 years were collected, spun, and frozen at −80 °C, using standard protocols, and a detailed assessment of fasting glucose, insulin, high-sensitivity C-reactive protein, low-density lipoprotein cholesterol, high-density lipoprotein cholesterol, and triglycerides has been reported (coefficient of variation was <5%)[38–40]. Fasting insulin was measured using an ultrasensitive automated microparticle enzyme immunoassay (Mercodia), which does not cross-react with proinsulin and the sensitivity of the immunoassay was 0.07 mU/L[42,43]. Participants were briefly asked about their personal and family (mother, father, and siblings) medical history such as a history of hypertension, diabetes, high cholesterol, and vascular disease at the 17-year clinic visit. Biological sex variable was collected by ALSPAC staff at delivery, from medical records, or birth notifications. All participants had attained puberty at the 17-year clinic visit which was estimated from time in years to age at peak height velocity objective assessment derived from Superimposition by Translation And Rotation mixed-effects growth curve analysis[39,43,44]. The participant's mother's socioeconomic status was grouped according to the 1991 British Office of Population and Census Statistics classification[45]. Questionnaires to assess smoking behaviour were administered at the 13, 15, and 24-year clinic visits. A specific question regarding whether participants smoked in the last 30 days was used as an indicator of current smoking status.

## Handling of missing covariates and multiple imputations

Eligible sample size varied by covariates and exclusions via listwise deletion of missing values ranging from 27.6% to 50.1% for covariates at either 11, 15, or 24-year clinic visits. The study participants were those who had at least one time-point measure at ages either 11, 15, or 24 years ($N = 6059$) and repeated the analyses among participants who met with more strict inclusion criteria. A Little's missing completely at random (MCAR) test was conducted to ascertain data missingness[46]. With Little's MCAR test: Chi-Square = 1159.60, degree of freedom = 260, $p$-value < 0.0001, we concluded that the variables were not missing completely at random. Regression-modelled multiple imputations were conducted using SPSS version 27 (IBM Corp, Armonk, NY, USA). 20 cycles of imputation with 10 iterations generated 20 imputed data sets and the specified constraints for the imputation process were the observed minimum and maximum values. In line with previous evidence[38], the percentage of missing values would be sufficiently addressed with 20 imputations: the variable with the highest missing value (50.1%, maternal socioeconomic status) had an estimate that was 98% efficient after 20 imputations (computed using Rubin's formula)[46]. The distributions of imputed variables were similar to the observed data when checked with histogram normality plot. Presenting pooled imputed results after conducting multiple imputations is preferred to presenting non-imputed results[47].

## Statistical analysis

The descriptive characteristics of the participants were summarized as frequencies and percentages, means and standard deviation, or medians and interquartile ranges. We explored sex differences using independent $t$ tests, Mann–Whitney $U$ tests, or Chi-square tests for normally distributed, skewed, or dichotomous variables, respectively. Multicategory variables were analysed using a one-way analysis of variance. Normality was assessed by quantile-quantile plot, histogram curve, and Kolmogorov-Smirnov tests with $p$-value > 0.05. We conducted a logarithmic transformation of skewed model residuals and confirmed normality prior to further analysis (mediation and temporal analysis).

**Analyses of longitudinal associations.** We examined the separate longitudinal associations of each of the 13-year ST, LPA, and MVPA progression (11 through 24 years) with each of BMI, waist circumference, total fat mass, trunk fat mass, lean mass, and lean mass to

fat mass ratio measured at ages 11, 15, and 24 years using generalized linear mixed-effect models (GLMM) with an identity link. The GLMM is robust for handling highly correlated variables such as ST and LPA[48]. The optimal model with the lowest Bayesian Information Criteria was one with sex as a main effect, a random intercept modelled for the participants and family cluster to account for within-individual correlations. Whilst the GLMM with a full information maximum likelihood is robust for handling missing at random predictor and covariate data, we elected to additionally conduct 20 cycles of multiple imputations to account for missing data. The GLMM accounted for baseline ST, LPA, MVPA predictors, body composition outcomes, and covariates and their repeated measures. Model 1 was unadjusted. Model 2 was adjusted for sex, family history of hypertension/diabetes/high cholesterol/vascular disease, socioeconomic status, and other time-varying covariates measured at three time-points such as age, low-density lipoprotein cholesterol, triglyceride, high-sensitivity C-reactive protein, high-density lipoprotein cholesterol, heart rate, systolic blood pressure, glucose, insulin, smoking status, and total fat mass or lean mass depending on the outcome. Model 3 was an additional adjustment for ST or LPA, depending on the predictor. Model 4 was an additional adjustment for LPA or MVPA, depending on the predictor to mutually adjust for activity levels. A different GLMM was conducted for each sex and sex-specific analyses were not adjusted for sex. BMI and lean mass-to-fat mass ratio outcomes were not adjusted for total fat mass and lean mass.

**Mediation path analyses.** Mediating path analyses using structural equation models separately examined the mediating role of cumulative glucose and insulin on the longitudinal associations of cumulative ST, LPA, or MVPA with each of cumulative total fat mass, trunk fat mass, and lean mass. The mediation analysis was conducted in line with the Guideline for Reporting Mediation Analyses of Randomized Trials and Observational Studies (AGReMA)[49]. The examined mediation mechanism between PA and body composition is partly based on previous studies in which higher PA was associated with a better metabolic profile with the latter, in turn, associated with a reduced risk of obesity[1,5,11,50]. The natural direct and indirect effects were estimated and presented[51,52]. Analyses were adjusted for age, sex, high-density lipoprotein cholesterol, low-density lipoprotein cholesterol, triglyceride, high-sensitivity C-reactive protein, family history of hypertension and cardiovascular diseases, smoking status, heart rate, glucose, insulin, ST, LPA, MVPA, total fat mass, or lean mass depending on the mediator, predictor, or outcome. The relationships between exposures, outcomes, and confounders are portrayed in the directed acyclic graph in Supplementary Fig. 2. The path models had three equations per regression analysis: the longitudinal associations of cumulative ST, LPA, or MVPA with cumulative glucose or insulin (Equation 1); the longitudinal associations of cumulative glucose or insulin with total fat mass, trunk fat mass, or lean mass (Equation 2); and the longitudinal associations of cumulative ST, LPA, and MVPA with cumulative total fat mass, trunk fat mass or lean mass (Equation 3, total effect), and Equation 3' (direct effect) accounted for the mediating role of glucose or insulin on the longitudinal associations of cumulative ST, LPA, and MVPA with cumulative total fat mass, trunk fat mass or lean mass. The proportion of mediating or suppressing roles was estimated as the ratio of the difference between Equation 3 and Equation 3' or the multiplication of Equations 1 and 2 divided by Equation 3 and expressed in percentage. A mediating or indirect role is confirmed when there are statistically significant associations between (a) the predictor and mediator, (b) the predictor and outcome, (c) the mediator and outcome, and when (d) the longitudinal association between the predictor and outcome variable was attenuated upon inclusion of the mediator[53]. However, when the magnitude of the longitudinal association between the predictor and outcome is increased upon

inclusion of a third variable, a suppression is confirmed[53]. This means that suppression occurs when the mediational path has an opposite effect, i.e. instead of a decrease in the point estimate of the direct effect between an exposure and an outcome in relation to the total effect, there is rather an increase in the direct effect above the total effect's point estimate[53]. We additionally examined the mediating or suppressing role of lipids, an inflammatory marker, and either total fat mass or lean mass depending on the outcome variable on the longitudinal associations of cumulative ST, LPA, and MVPA with cumulative total fat mass and lean mass. We considered a statistically significant mediation or suppression of <1% as minimal, and ≥1% as partial. Path analyses were conducted with 1000 bootstrapped samples[54,55].

**Temporal causal path analyses.** Lastly, we used structural equation modelling with autoregressive cross-lagged design to examine the separate temporal associations of ST, LPA, and MVPA with each of total fat mass and lean mass. The cross-lagged models first tested the separate associations of ST, LPA, and MVPA at 11 years with each of total fat mass and lean mass at 15 years. Next, the separate associations of total fat mass and lean mass at 11 years with ST, LPA, and MVPA at 15 years were examined. Thereafter, we examined the separate associations of ST, LPA, and MVPA at 15 years with each of total fat mass and lean mass at 24 years. Lastly, the separate associations of total fat mass and lean mass at 15 years with ST, LPA, and MVPA at 24 years were examined. These models were adjusted for all the covariates measured at 11 and 15 years or 15 and 17 years for metabolic factors. In the cross-lagged design, the potential association could be; ST, LPA, and MVPA leading to obesity risks, obesity risks leading to ST, LPA, and MVPA, or bidirectional associations of ST, LPA, and MVPA with obesity risks. If a path from ST, LPA, and MVPA at time t-1 (11 years) to each of total fat mass and lean mass at time t-2 (15 years) reach significant ($p$-value < 0.05), changes in the earlier variables are considered to temporally precede changes in the later, and vice versa. Likewise, if a path from ST, LPA, and MVPA at time t-2 (15 years) to each of total fat mass and lean mass at time t-3 (24 years) reach significant ($p$-value < 0.05), changes in the earlier variables are considered to temporally precede changes in the later, and vice versa. A stronger predictive effect is determined by a larger standardized regression coefficient. Error terms were included in the cross-lagged model.

Collinearity diagnoses were performed and accepted results with a variance inflation factor <5, considered differences and associations with a 2-sided $p$-value < 0.05 as statistically significant, and made conclusions based on effect estimates and their confidence intervals (CI). Covariates were identified based on previous studies and are portrayed in the directed acyclic graph (Supplementary Fig. 2)[1,5,40,42,43,45,50,56–58]. We applied Sidak-correction for potential multiple comparisons. To ascertain the robustness of the results, we conducted additional analyses among 2457 participants who had at least a one-time point measure of ST and PA and complete DEXA measure of total fat mass and lean mass and among 917 participants who had at least a two-time-point measure of ST, PA, and complete three time-point measure of total fat mass and lean mass during the 13-year follow-up. We also presented compositional data analyses with the isometric log-ratio transformation of movement behaviour in relation to body composition outcomes in the supplementary information file[59]. Analyses involving 50% of a sample of 10,000 ALSPAC children at 0.8 statistical power, 0.05 alpha, and 2-sided p-value would show a minimum detectable effect size of 0.048 standard deviations if they had relevant exposure for a normally distributed quantitative variable[60]. All statistical analyses were performed using SPSS statistics software, Version 27.0 (IBM Corp, Armonk, NY, USA), whilst mediation analyses and auto-regressive cross-lagged temporal causal path structural equation modelling were conducted using AMOS version 27.0. Chicago: IBM SPSS.

**Reporting summary**

Further information on research design is available in the Nature Portfolio Reporting Summary linked to this article.

## Data availability

All individual-level data analysed in this study can be accessed via an approved application to ALSPAC Executive. The informed consent obtained from ALSPAC participants does not allow the data to be made freely available through any third-party maintained public repository - ALSPAC access policy (PDF, 614kB), Section 6.7.1.1b. Summary statistics of the data supporting the key findings of this study are available in the article and in its Supplementary Data files. The ALSPAC data management plan describes in detail the policy regarding data sharing, which is through a system of managed open access. Full instructions for applying for data access can be found here: http://www.bristol.ac.uk/alspac/researchers/access/. The ALSPAC study website contains details of all the data that are available (http://www.bristol.ac.uk/alspac/researchers/our-data/). To access data, please submit your research proposal (https://proposals.epi.bristol.ac.uk/) for consideration by the ALSPAC Executive Committee. You will receive a response within 10 working days to advise you whether your proposal has been approved. Once your research proposal has been approved, you will be assigned a data buddy who will help you at every stage of your project. They will send you the relevant paperwork and documentation for accessing the data and samples. Please read the ALSPAC access policy (PDF, 614kB) which describes the process of accessing the data and samples in detail, and outlines the costs associated with doing so. If you have any questions about accessing data, please email alspac-data@bristol.ac.uk. The data points which were used to plot the trajectories in Fig. 1 are provided in the Source Data file. Source data are provided with this paper.

## Code availability

Only publicly available tools (SPSS statistics software, Version 27.0 (IBM Corp, Armonk, NY, USA) and AMOS version 27.0. Chicago: IBM SPSS) were used in data analysis and the parameters have been described wherever relevant in Methods and Reporting Summary.

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

## Acknowledgements

We are extremely grateful to all the families who took part in this study, the midwives for their help in recruiting them, and the whole ALSPAC team, which includes interviewers, computer and laboratory technicians, clerical workers, research scientists, volunteers, managers, receptionists and nurses. The UK Medical Research Council and Wellcome (Grant ref: 217065/Z/19/Z) and the University of Bristol provide core support for ALSPAC. The British Heart Foundation grant (CS/15/6/31468) funded blood pressure, carotid-femoral pulse wave velocity, and Actigraph activity monitoring device measurement at 24 years. The Medical Research Council grant (MR/M006727/1) supported smoking

data collection. A comprehensive list of grant funding is available on the ALSPAC website (http://www.bristol.ac.uk/alspac/external/documents/grant-acknowledgements.pdf); A.O.A's research group (UndeRstanding FITness and Cardiometabolic Health In Little Darlings: *urFIT-child*) was funded by the Jenny and Antti Wihuri Foundation (Grant no: 00180006), the North Savo regional and central Finnish Cultural Foundation (Grants no: 65191835, 00200150 and 00230190), Orion Research Foundation sr, Aarne Koskelo Foundation, Antti and Tyyne Soininen Foundation, Paulo Foundation, Paavo Nurmi Foundation, Yrjö Jahnsson Foundation (Grant no: 20217390), Ida Montin Foundation (Grant no: 20230289), Eino Räsänen Fund, Matti and Vappu Maukonen Fund, Foundation for Pediatric Research, the Finnish Foundation for Cardiovascular Research (Grant no: 220021 and 230012) and the Alfred Kordelin Foundation (230082). W.P., and T-P.T., have no funding to disclose. The funders had no role in the design and conduct of the study; collection, management, analysis, and interpretation of the data; preparation, review, or approval of the manuscript; and decision to submit the manuscript for publication.

## Author contributions

A.O.A. is the principal investigator in this study who conceived the study, obtained financial support, and take responsibility for the integrity of the data and the accuracy of the data analysis and supervised the entire study. A.O.A. had full access to all the data in the study acquired from the Avon Longitudinal Study of Parents and Children (ALSPAC). A.O.A. is responsible for the study design, data curation, performed all statistical analyses, interpreted the results, and drafted initial manuscript. A.O.A., W.P. and T-P.T. critically revised the manuscript for important intellectual content: A.O.A. responded to all reviewers' concerns. All authors approved the final report for publication.

## Competing interests

The authors declare no competing interests.
