## [Peer Review File · Nature Communications]

Reviewers' Comments:

Reviewer #1:

Remarks to the Author:

I greatly enjoyed reading this manuscript and thinking about your findings and their implications. There are a number of moderately complexity statistical methods involved in combination in your work and it is challenging to clearly communicate results from these. It might help for me to note that many of my comments are specifically from my perspective as a biostatistician whose work often involves childhood and adult obesity in a public health context.

While I greatly appreciated the details the authors provided in the methods for their analyses, I wondered how well typical readers would follow some of these and wondered if some details (which I also think need adding to in places) could be included in a supplement with briefer descriptions in the manuscript body itself. At the same time, I will suggest that you could provide code in a supplement to help readers such as data scientists and (bio)statisticians follow your work. My expectation as a biostatistician is that if I had an article and the data used for it in hand, I should be able to use the methods to reproduce all values in the results (and any new values in the discussion) without too much, and ideally no, trial and error.

The use of PA measures is almost always complicated (what does the slope for ST mean if LPA and MVPA have to be held constant at the same time for interpretation, for example), particularly without data on sleep. If ST increases, doesn't this suggest a decrease in LPA, MVPA, or sleep (or at least non-wear)? Compositional data approaches can be useful/required in such cases and I wondered whether the authors had considered these. There is a large literature on this topic if the authors had not already worked through this.

I also have, what are for me, significant questions about the analytic data set (far more participants seem to have been excluded from analysis than strictly required given the MI strategy, but perhaps I'm misunderstanding some details here) and so have tried to avoid too many specific/detailed comments in case responding to these leads to substantial revisions. I hope that these comments are useful in thinking about possible modifications to your interesting manuscript.

I will now make some more specific comments, not always in order of appearance in the manuscript.

For the model diagnostics at the start of the statistical analysis section (Line 175 onward), I am assuming the examinations of normality (Lines 180 and 181) were of the model residuals (or the data within groups, so as to inform on the population error term) and that "skewed" on Line 178 means "skewed (including after a logarithmic transformation)". I couldn't recommend K-S tests (Line 180) but if used, the level of significance should be stated (c.f. Line 250). I was a little surprised that sometimes the assumptions for t-tests were found wanting but not for one-way ANOVAs (Line 179). This is entirely possible, it just caught my attention. I would have liked to see more model diagnostics (not just VIFs) for the regression models. I was unable to see why "Skewed covariates were logarithmically transformed" (table notes) unless this was due to non-linearities.

Are you sure you used "generalized linear mixed-effect models (GLMM)" as this would indicate a family other than Normal/Gaussian and suggest the possibility of a non-identity link function. Perhaps you mean a general linear mixed model? It wasn't clear to me what the "unstructured correlation matrix" (Line 190) was for. With a single random effect (for participants, and I suggest avoiding "subject", e.g., Line 189, as some people see this as dehumanising), it's unclear to me from the text what the error covariance matrix here would be accounting for. I think Lines 190-192 could be made clearer, or perhaps removed.

A mixed model assumes data is MCAR (missing completely at random) or MAR (missing at random) conditional on some variable on the RHS of the equation, rather than the unconditional statement on Line 193. It's certainly not robust to missing data (Line 193) without qualification as it cannot address informative missing data (or MAR data that would be NMAR without conditioning

on some variable(s)). I would also question its robustness to collinearity (Lines 187–188) being notable (particularly since the example given is for what are usually here exposures rather than outcomes). I wasn't sure what the "collinearity diagnostics" (Line 256) referred to beyond VIFs.

The reference to the optimal model (Lines 188–189) cannot be interpreted without knowing what models were investigated, and arguably whether BIC was reasonably improved compared to a more parsimonious model. The issues with using IC to select models are well known but I would accept this as a pragmatic approach here.

I wondered at this point about families with multiple participants (i.e., siblings). A family random effect would be indicated for these (in all inferential analyses) if they were included in data, which seems likely to me, but this did not seem to be explained in the methods. Can the authors comment on this? Tangentially related to this, the Pearson's correlations on Lines 145–146 would be interpretable even if these are based on 1, 2, or 3 rows of data per participant and/or multiple siblings per family, but the p-values (Line 146) would need to accommodate the clustering (at the individual and perhaps also family levels).

As mentioned above, I wondered about the (perhaps more demanding than strictly necessary) inclusion criteria "In this study, of the 917 participants with complete body composition measures at 11, 15, and 24 years clinic visits, with at least two time-points valid ST, LPA, and MVPA measurements at either age 11, 15, or 24 years clinic visit and were eligible for analyses" (Lines 119–122). Given the much higher sample sizes in Supplementary Figure 1 and explained in the text, I'm not able to understand why only 917 of the study participants were included. Surely you wish to also generalise to those who did not satisfy these data-based inclusion criteria? The use of ML, FIML, and MI would each allow you to include many more (MI and FIML addressing cases where exposure or covariate data and not just outcome data is missing). From Supp. Figure 1, it looks like as many as 6362 participants might potentially have valid exposure data (the wording in the boxes to the left seemed like it could be slightly misleading, if 5956 completed PA assessment at age 11, I'm guessing that 3948 with "at least 1 clinic visit [PA] measurement" might be referring to "exactly 1 clinic visit measurement", or am I confusing myself here?) and at least 7051 (perhaps more) had at least one outcome measurement, suggesting around, say, 6000 or more with at least some exposure and outcome data. In theory, even more could be included in the imputation models if they had covariate data for variables in the imputation model, but this is perhaps unlikely to be a significant number of participants. I understand that without MI, including only those with sufficient data would be reasonable, but here, you appear to have the opportunity (arguably the ethical obligation) to include many more participants. Supp. Table 1 along with Line 43 suggests that the 1123 participants excluded (and above I have speculated that even more than these could have included in the MI analyses) are more often male (sex was not included in the table for some reason) and perhaps differ just enough from those included to be interesting, and might lead one to wonder if the others who could also have been included might not be even more so. The worst case of including more participants, aside from the additional statistical work, is likely to be more precise estimates for a more clearly defined population.

This then leads into questions about informative missing data. I appreciate that there are already a considerable number of statistical analyses here, but pattern-mixture models (or another approach to NMAR) would seem prudent to see how robust the findings are to informative missing data mechanisms and could be useful in the discussion.

Note that when presenting results after using MI, it is conventional (at least in my circles) to show the MI-based results in the manuscript (as you do) but also to show CC-based results in a supplement to help readers understand how missing data could have (subject to assumptions) distorted results and sometimes to show the robustness of the findings to assumptions around missing data.

The description on Lines 193–195 doesn't, for me, explain exactly how MI was done so that a reader would know (at least roughly) what code they would have used in their preferred statistical software. If providing this level of detail becomes cumbersome in the text, providing code in a supplement might help? It's unclear precisely what "especially" (Line 195) means. If variables on both sides of the equation were imputed, which is my presumption, what does this mean? A little

more detail about the bootstrapped mediation models (around Line 232) would also be useful (Preacher-Hayes?), along with a reference or two for the reader wishing to understand exactly what was done here (I do appreciate the authors explaining more in the text before this point). Related to this, I'd avoid absolute phrases such as "no mediating effect" when the values listed include non-zero. The definition of what constitutes mediation and suppression (Lines 223–229) could be expanded to distinguish between non, partial, and total (e.g., Lines 307–325 might indicate an implicit rule that <1% = no [which I would avoid, perhaps "minimal" or similar], 1%+ = partial, but if so this should be made explicit in the methods), although there is plenty of debate about this, if you decided to move some of the more complicated methods to a supplement as I suggest.

I appreciate the use of Model 1 and Model 2, presumably to reduce the risk of overadjustment in Model 2, but Model 1 has several time-varying variables that could potentially be on the causal pathway between exposure and outcome (e.g., blood glucose and insulin for PA → body composition). Indeed, this is part of the research question addressed in the manuscript. Is there a reason for not including a Model 0 (with only baseline/unchanging characteristics included or reflecting "Equation 3" from the Figures)? Model 2 is challenging to interpret as, e.g., a decrease in either/both LPA or/and MVPA would presumably automatically increase ST if sleep was not affected (or increase non-wear times)? I also wonder if readers will not perhaps be confused by the table note "For continuous variable predictors (ST, LPA and MVPA), a 1-standard deviation change is associated with a 1-standard deviation change in the outcome" without this being related back to the reported coefficients.

The main models seem to have a single slope for the PA and biochemistry variables over all three ages. Assuming this is the case, are the authors confident that, e.g., the effect of ST on blood glucose might not vary over the three ages, i.e., did they consider and perhaps investigate effect modification by age? I appreciate that the presence of such interactions would complicate the presentation of the results, but I feel that such interactions would also be of interest and importance.

This leads to an important question about sleep for me. Sleep (both quantity and quality) seem especially plausible mediators between the PA and composition variables (as well as potentially independent predictors of the outcomes here). The lack of sleep data seems a major limitation of the present research and I was very surprised to be unable to find any instances of the word "sleep" (or its inflected forms) in the manuscript or the supplement.

I think readers will appreciate a clearer identification of the family for the multiplicity adjustment (Lines 259–260, and table notes). In theory, at least, the family could be at the regression model level, table level, or any of a number of other levels.

I appreciated the authors considering the adequacy of their sample size for their proposed analyses (Lines 260–262) but I'm not sure that these will assist readers, particularly in the context of the present research with its clustering and multiplicity adjustments. Could this be made more specific (including using your actual analytic sample size) and perhaps easier for readers to understand? Ultimately, the power of your analyses is now entirely communicated through your confidence interval limits, so perhaps explaining this, along with calculations prior to performing the analyses, would be useful.

Table 3 stratifies analyses by sex, but doesn't examine evidence for effect modification by sex (which would be "sex-specific" rather than "sex-based", c.f. title and Lines 204–205 and 303–304). This might be easier to include by rearranging the tables so that males and females are columns. With appropriate interactions, results for the two sexes and evidence for effect modification could be extracted from a single model in each case.

I appreciate the authors overall care around causality in places, but I feel that "causal effects" (Line 41) and similar overstate their ability to draw causal claims from observational data here. With an underlying causal model, they certainly can investigate consistency between this model and their data, but given the likelihood of unmodelled confounding (I felt Lines 427–430 was rather optimistic), I would carefully remove all causal language (including for me "confirms" on Line 372) aside from where it is clearly about potential mechanisms or speculative (e.g., Lines 81–

83 are clearly about mechanisms and Lines 85–89 are perfectly fine to me). This includes the conclusion about associations being “unlikely [to be] causal” (Line 54) and even “may be causally associated” (Line 55) unless stronger arguments (around confounding bias being sufficiently low to support these) can be added. Lines like 247–248’s “changes in the earlier variables are considered to lead to changes in the later, and vice versa.” in particular stood out to me as claiming much too much for observational data.

I appreciated you providing references to articles that informed your underlying causal model (Lines 258–259) but I wonder if a visual representation (e.g., a directed acyclic graph) here would help readers to understand the minimum adjustment set for each exposure:outcome pairing. This could also go in a supplement, with a more thorough explanation of the variables in the models, but I personally like to see a DAG in any observational study of moderate complexity or above as a matter of course.

It wasn’t clear to me why the goodness of fit indices described on Lines 252–255 seemed specific to the cross-lagged models and not the mediation models. Personally, I wouldn’t see these as necessary in general given the seeming lack of latent variables. Regression model diagnostics would suffice for me here if this is the case.

I was uncertain whether “cumulative” here (e.g., Lines 145 and 209) meant more than “at each age”, particularly for ST, LPA, and MVPA.

Reviewer #2:

Remarks to the Author:

General comments:

This is an interesting paper that uses data from the UK AVON birth cohort study to explore the overall and sex-specific longitudinal associations between accelerometer assessed movement behaviours and adiposity from age 11y to 24y. There is a lack of longitudinal research using quality objective measures, so this paper fills this gap. I do have specific comments below, however, about some of the analyses in terms of timing of data collection and claims about causality. A general comment is there is a lot in this paper, and I wonder for the sake of coherence and the sheer volume of analyses performed if the number of tests could be reduced (eg, the mediation and categorical analyses, and even the sex specific analyses which are not discussed at all or mentioned in conclusions).

Specific comments:

1. Paper Title: While I understand the analyses provide an estimation of “causal effects”, I have concerns about this term (described further in comment #2 below) and wonder if the term is appropriate here?
2. Abstract, line 32: The authors need to be careful in use of the term cause here as longitudinal data are still observational. The analytic technique applied in the paper can facilitate understanding of causal pathways (potential causality), but not prove them. I recommend toning down claims regarding cause-effect throughout the paper (including the title).
3. Abstract, lines 33-35: Please amend the aims here to include mention of sex specific analyses and also exploring potential 'reverse causality'.
4. Abstract, lines 39-40: Please mention who’s guidelines. The WHO?
5. Abstract, lines 49-50: “Cumulative insulin, glucose, lipids, and low-grade inflammation differently mediated the associations of movement behaviours with body composition”. The wording in this sentence is awkward, please amend.
6. Abstract, lines 51-52: I was surprised by the mention of fat mass predicting MVPA because this wasn’t mentioned in the aims (see comment #3).

7. Abstract, lines 53-54: "ST increases and LPA decreases fat mass from childhood through young adulthood, but the relationship is unlikely causal." I don't think this statement can be made based on the study design. Please amend this statement.
8. Abstract, lines 54-56: I have concerns about the mediating analysis because of the timing of the collection of bloods. This conclusion may need to change.
9. Keywords: I don't think Health Promotion is an appropriate key word for this paper.
10. Introduction, line 77: The updated WHO PA guidelines have removed the need for this to be every day. The updated version recommends at least 60 minutes/day of moderate-to-vigorous PA (MVPA) on average.
11. Introduction, lines 84-90: I understand that word limits in journals make it difficult to provide a comprehensive rationale leading to the study aims, but I think some mention of the biological mediating pathways is important in the Introduction. For example, what are some biological pathways that mediate associations between movement behaviours and adiposity so that there is no direct causal pathway (what is being argued in the conclusions I believe)?
12. Introduction: In relation to the above point, a brief rationale is needed for the sex specific analyses. While many studies report boys/men are more active than girls/women no study is cited to support this aim. Furthermore, what is the rationale for a sex-specific longitudinal association and/or mediating pathway?
13. Introduction, line 96: Aim 3 needs to mention also exploring the potential reverse associations between adiposity and PA/ST.
14. Methods, lines 133-135: It would be helpful to mention consistency in assessing movement behaviours between these two accelerometer models.
15. Methods, lines 143-144: Please report what published accelerometer algorithm (cutpoints) was used for the 11yr olds and 15yr olds. Reference 8 is suitable for the adult cutpoint, I don't believe the other references (9 and 10) are validity studies?
16. Methods, lines 145-146: Please report which timepoint the correlations between intensities are taken from.
17. Methods, lines 148-149: These categories of PA are not recommended by the WHO, this reference needs to be removed from this sentence.
18. Methods, lines 149-150: I have a few issues with the categories, both in interpreting the analyses (more on that later) and in understanding whether this was completed at each time point? And if tertiles were created at each time point, because of the declining & then increasing PA levels with age, I assume that at T2 and T3 the tertiles may not have aligned with the guidelines so well?
19. Methods, line 155: The BMI acronym was already defined in the Introduction
20. Methods, lines 162-174, and lines 200-202: The use of data collected at timepoints (13yrs and 17yrs) that differ from the exposure and outcome variables (11yrs, 15yrs, 24yrs) is I think problematic. While family medical history (collected at 17yrs) is probably ok to include as a covariate, I have concerns about including bloods from the data collection at 17yrs when no corresponding physical activity or adiposity data have been collected. This is a 2 yr gap from the 15yr old timepoint and 7 year gap to the 24 yr old timepoint. Given how much PA changes with age, having this time point included as a mediator assumes from a causal perspective steady state of PA/ST between 15yrs and 17yrs? Also smoking was assessed at 13yrs and 17yrs as well as 24yrs. Although this is just a covariate, only 1 of these time points lines up with the exposure and outcome data.

21. Methods, lines 187-188 and lines 203-204: I am concerned about adjustments for all intensities in Model 2. The high correlation between several intensities (especially ST and LPA) is a concern, especially with a longitudinal study where changes in one intensity will be related to changes in another. It is for this reason that more recently researchers have been using compositional analyses which is better equipped to handle inter-related (in this case time bound) data. Is it possible to provide a reference to support the statement that "The GLMM is robust for handling highly correlated variables such as ST and LPA"?

22. Methods, lines 208-210: As noted in comment #20, the collection of bloods at 17 yrs means there is a misalignment here between the PA/ST exposures and the adiposity outcomes with relation to the mediators. Given the declines in PA in youth (eg, from 15 yrs to 17yrs), does this model assume some stability in PA behaviour given the bloods were collected at 17 yrs? How were the bloods from the 2 time points included in the model? I don't think the mediating analysis is strong given this limitation and suggest it is removed.

23. Methods, line 256: "Collinearity diagnoses were performed and accepted results with a variance inflation factor <5". Please clarify which models had a VIF greater than 5.

24. Results, lines 284-285: What about the number of participants who provided data at 17 yrs? This should also be presented in Supplemental Figure 1.

25. Results, lines 290-291: The J or U-shaped increase in PA from 15 yrs to 24 yrs could be due to changes in cut-points from child to adult. One of the challenges with longitudinal studies from early childhood to childhood or from childhood to adulthood is the change in threshold counts means it is 'easier' for young adults to accumulate MVPA mins due to the lowering of the threshold. Did you also look at CPM? I would be surprised if PA actually increased from 15 yrs to 24 yrs.

26. Discussion: There is no discussion at all about how the findings differed by sex.

27. Discussion, lines 361-364 and lines 401-402: Please re-word these sentences for ease of reading.

28. Discussion, lines 413-414: As per comment #25, different count thresholds may also explain this apparent U/J-shaped curve.

29. Discussion, line 426: The mis-alignment between data collection of bloods at 17y with accelerometer and adiposity data is a major limitation which I think impacts credibility of the mediation analyses.

30. Discussion, line 434 and line 439: I disagree with these conclusions about causality, particularly if this is based on the mediation analysis which used bloods collected at a different time point from the PA/adiposity measures (eg 17yrs).

31. Table 1: The 'mean (SD)' at the top of each column in Table 1 don't align with all the variables underneath (eg, %s). I suggest removing these and instead put 'mean (SD)' alongside relevant rows for continuous variables. Also, the sub-heading 'Anthropometry' doesn't seem to describe the variables sitting under it. Perhaps change to Sociodemographic characteristics?

32. Table 1: It seemed from the Methods that sociodemographic information was collected at 17yrs of age, not at 11yrs?

33. Table 2: "Categorical cumulative predictor variable from ages 11 – 24 years". I had trouble understanding the categorical analysis and didn't see a clear explanation of it in the results. Because participants can move from one category to another at each time point, how are the changes in categories (or stability) over time handled & interpreted? Is this analysis needed given how much is already in this paper?

Reviewer #3:

Remarks to the Author:

This study examined the associations of cumulative sedentary time (ST), light physical activity (LPA), and moderate to vigorous physical activity (MVPA) from age 11 – 24 years with repeated measures of body mass index (BMI), and measures of body composition assessed by dual-energy X-ray absorptiometry. The authors found that over a 13-year follow-up, ST increase was associated with increased adiposity while cumulative LPA was associated with lower adiposity. Furthermore, persistent exposure to =60mins/day of MVPA was associated with decreasing total and truncal adiposity. Authors also report that MVPA may causally lower total fat mass via an increase in high-density lipoprotein cholesterol and lean mass. This is an important study providing novel information of significance that can inform physical activity guidelines and interventions. The manuscript is clearly written. However, I do have the following comments.

It would be informative to provide estimates of minimally adjusted or crude models in addition to models 1 and 2.

Model 1 is adjusted for the hypothesized mediators, glucose and insulin, which is not recommended.

In contrast, height adjustment should be considered. For example, waist circumference is also determined by height.

Mediation analyses are a major part of the manuscript, which adds to the strengths of the work. However, I do have a few comments that may help to improve the interpretation of the findings. It would be good to follow the consensus-based reporting guideline for studies reporting mediation analyses (A Guideline for Reporting Mediation Analyses; AGRReMA):

<https://jamanetwork.com/journals/jama/fullarticle/2784353>

In particular, it would be helpful to provide a directed acyclic graph (DAG) to represent the causal relationship between variables.

Also, the theory that underpins the proposed mechanisms of interest in the mediation analyses should be described and evidenced.

In Figure 1, it would be preferable to report 95% CI in addition to (or instead of) of p-values.

Discussion, line 361, it is not clear what the authors meant with “confounding longitudinal analyses”.

Interpretation of “suppressive roles” is unclear and not common in a mediation context. Could the authors add a brief explanation.

How do the authors explain the “suppressive role” of ST through insulin with total fat mass? One would expect a positive association between ST and insulin levels.

Could the authors explain a bit more how ST would increase fat mass via increased muscle mass?

Could the authors add a brief explanation in line 433-434 as to why the relationships are “...unlikely causal”.

Despite the longitudinal study design, causality in observed associations is uncertain, because unmeasured confounding or other biases may distort the observed associations. This should probably be emphasized more in the limitations and use of causal language tempered in the conclusions (abstract and main text). Furthermore, it is unclear how authors judge which of the relationships were deemed causal while others were not.

In the abstract, please provide the unit in the reported “effect estimate”.

Response to reviewers' comments

Many thanks for the detailed and incisive comments. The reviewers' time spent in providing such helpful suggestions and several compliments is well acknowledged. Kindly find our point-by-point response to the suggestions below. We look forward to further helpful suggestions in improving the presentation of the revised manuscript.

As suggested by reviewer #1, we have expanded the inclusion criteria and repeated the analyses in 2457 participants who had at least one timepoint valid accelerometer measure during the 13-year follow period and complete DEXA measure of fat mass at the three-time points 11, 15, and 24 years. Overall the new results were consistent with what was earlier reported and have been presented in the supplementary appendix.

Reviewer #1 (Remarks to the Author):

I greatly enjoyed reading this manuscript and thinking about your findings and their implications. There are a number of moderately complexity statistical methods involved in combination in your work and it is challenging to clearly communicate results from these. It might help for me to note that many of my comments are specifically from my perspective as a biostatistician whose work often involves childhood and adult obesity in a public health context.

While I greatly appreciated the details the authors provided in the methods for their analyses, I wondered how well typical readers would follow some of these and wondered if some details (which I also think need adding to in places) could be included in a supplement with briefer descriptions in the manuscript body itself. At the same time, I will suggest that you could provide code in a supplement to help readers such as data scientists and (bio)statisticians follow your work. My expectation as a biostatistician is that if I had an article and the data used for it in hand, I should be able to use the methods to reproduce all values in the results (and any new values in the discussion) without too much, and ideally no, trial and error.

Response: We thank the reviewer for the compliment and suggestion. It is noted in Nature communication authors guidelines that the method section is not included in the manuscript word count. *"The main text (not including abstract, Methods, References and figure legends) is limited to 5,000 words.... Methods are typically less than 3000 words."* Hence, we could retain some details in the method as the method section word count is currently 2848 words after revision.

Regarding data availability, *"The informed consent obtained from ALSPAC participants does not allow the data to be made freely available through any third-party maintained public repository. However, data used for this submission can be made available on request to the ALSPAC Executive. The ALSPAC data management plan describes in detail the policy regarding data sharing, which is through a system of managed open access. Full instructions for applying for data access can be found here: <http://www.bristol.ac.uk/alspac/researchers/access/>. The ALSPAC study website contains details of all the data that are available (<http://www.bristol.ac.uk/alspac/researchers/our-data/>)."*

Regarding code, we did not generate any new customizable code but applied the practical guidelines in generally accessible IBM Amos and SPSS manuals for the GLMM and Structural Equation Models.

<https://www.ibm.com/support/pages/ibm-spss-statistics-27-documentation#en>

https://www.ibm.com/docs/en/SSLVMB_27.0.0/pdf/amos/IBM_SPSS_Amos_User_Guide.pdf

The use of PA measures is almost always complicated (what does the slope for ST mean if LPA and MVPA have to be held constant at the same time for interpretation, for example), particularly without data on

sleep. If ST increases, doesn't this suggest a decrease in LPA, MVPA, or sleep (or at least non-wear)? Compositional data approaches can be useful/required in such cases and I wondered whether the authors had considered these. There is a large literature on this topic if the authors had not already worked through this.

Response: We have extended the analytic strategy of the GLMM to include 4 models which involve adjusting for each movement behaviour in Models 3 and 4. Our colleague has recently analysed PA data in the ALSPAC cohort with compositional analyses and compared with regression analyses and the results were similar, now published in a master's thesis.

Sansum KM. Associations between physical activity and sedentary time with endothelial function, arterial stiffness, arterial elasticity, and clustered cardiometabolic risk in children: The ALSPAC Study.

<http://hdl.handle.net/10871/130013>

Similar consistent results between compositional data and regression analyses have been shown in a Canadian pediatric cohort.

Bird et al. Associations of neighborhood walkability with moderate to vigorous physical activity: an application of compositional data analysis comparing compositional and non-compositional approaches. Int J Behav Nutr Phys Act. 2022 May 18;19(1):55. doi: 10.1186/s12966-022-01256-6.

I also have, what are for me, significant questions about the analytic data set (far more participants seem to have been excluded from analysis than strictly required given the MI strategy, but perhaps I'm misunderstanding some details here) and so have tried to avoid too many specific/detailed comments in case responding to these leads to substantial revisions. I hope that these comments are useful in thinking about possible modifications to your interesting manuscript. I will now make some more specific comments, not always in order of appearance in the manuscript.

For the model diagnostics at the start of the statistical analysis section (Line 175 onward), I am assuming the examinations of normality (Lines 180 and 181) were of the model residuals (or the data within groups, so as to inform on the population error term) and that "skewed" on Line 178 means "skewed (including after a logarithmic transformation)".

Response: The skewed data referred to before logarithmic transformation.

I couldn't recommend K-S tests (Line 180) but if used, the level of significance should be stated (c.f. Line 250).

Response: K-S tests with p-value >0.05 has been added to the text.

I was a little surprised that sometimes the assumptions for t-tests were found wanting but not for one-way ANOVAs (Line 179). This is entirely possible, it just caught my attention. I would have liked to see more model diagnostics (not just VIFs) for the regression models. I was unable to see why "Skewed covariates were logarithmically transformed" (table notes) unless this was due to non-linearities.

Response: Skewed variables were log-transformed due to non-linearity and particularly used in mediation and temporal analyses but not in GLMM as discussed below.

Are you sure you used "generalized linear mixed-effect models (GLMM)" as this would indicate a family other than Normal/Gaussian and suggest the possibility of a non-identity link function. Perhaps you mean a general linear mixed model?

Response: We used GLMM in the study for the non-linear data. The log-transformed data were used in the mediation and temporal analyses.

It wasn't clear to me what the "unstructured correlation matrix" (Line 190) was for. With a single random effect (for participants, and I suggest avoiding "subject", e.g., Line 189, as some people see this as dehumanising), it's unclear to me from the text what the error covariance matrix here would be accounting for. I think Lines 190–192 could be made clearer, or perhaps removed.

Response: Lines 190-192 has been removed.

A mixed model assumes data is MCAR (missing completely at random) or MAR (missing at random) conditional on some variable on the RHS of the equation, rather than the unconditional statement on Line 193. It's certainly not robust to missing data (Line 193) without qualification as it cannot address informative missing data (or MAR data that would be NMAR without conditioning on some variable(s)). I would also question its robustness to collinearity (Lines 187–188) being notable (particularly since the example given is for what are usually here exposures rather than outcomes). I wasn't sure what the "collinearity diagnostics" (Line 256) referred to beyond VIFs.

The reference to the optimal model (Lines 188–189) cannot be interpreted without knowing what models were investigated, and arguably whether BIC was reasonably improved compared to a more parsimonious model. The issues with using IC to select models are well known but I would accept this as a pragmatic approach here.

Response: We have qualified the sentence as follows "*the mixed-effect model is robust for handling missing at random (MAR) predictor and covariate data*".

I wondered at this point about families with multiple participants (i.e., siblings). A family random effect would be indicated for these (in all inferential analyses) if they were included in data, which seems likely to me, but this did not seem to be explained in the methods. Can the authors comment on this? Tangentially related to this, the Pearson's correlations on Lines 145–146 would be interpretable even if these are based on 1, 2, or 3 rows of data per participant and/or multiple siblings per family, but the p-values (Line 146) would need to accommodate the clustering (at the individual and perhaps also family levels).

Response: The ALSPAC cohort of almost 15,000 children were 98% singleton from over 15,000 separate mothers. Hence we do not consider that a family cluster is applicable.

Boyd et al. Cohort Profile: the 'children of the 90s'--the index offspring of the Avon Longitudinal Study of Parents and Children. *Int J Epidemiol.* 2013 Feb;42(1):111-27. doi: 10.1093/ije/dys064.

As mentioned above, I wondered about the (perhaps more demanding than strictly necessary) inclusion criteria "In this study, of the 917 participants with complete body composition measures at 11, 15, and 24 years clinic visits, with at least two time-points valid ST, LPA, and MVPA measurements at either age 11, 15, or 24 years clinic visit and were eligible for analyses" (Lines 119–122). Given the much higher sample sizes in Supplementary Figure 1 and explained in the text, I'm not able to understand why only 917 of the study participants were included. Surely you wish to also generalise to those who did not satisfy these data-based inclusion criteria? The used of ML, FIML, and MI would each allow you to include many more (MI and FIML addressing cases where exposure or covariate data and not just outcome data is missing). From Supp. Figure 1, it looks like as many as 6362 participants might potentially have valid exposure data (the wording in the boxes to the left seemed like it could be slightly misleading, if 5956 completed PA assessment at age 11, I'm guessing that 3948 with "at least 1 clinic visit [PA] measurement" might be referring to "exactly 1 clinic visit measurement", or am I confusing myself here?) and at least 7051 (perhaps more) had at least one outcome measurement, suggesting around, say, 6000 or more with at least some exposure and outcome data. In

theory, even more could be included in the imputation models if they had covariate data for variables in the imputation model, but this is perhaps unlikely to be a significant number of participants. I understand that without MI, including only those with sufficient data would be reasonable, but here, you appear to have the opportunity (arguably the ethical obligation) to include many more participants. Supp. Table 1 along with Line 43 suggests that the 1123 participants excluded (and above I have speculated that even more than these could have included in the MI analyses) are more often male (sex was not included in the table for some reason) and perhaps differ just enough from those included to be interesting, and might lead one to wonder if the others who could also have been included might not be even more so. The worst case of including more participants, aside from the additional statistical work, is likely to be more precise estimates for a more clearly defined population.

Response: We expanded the inclusion criteria to include all 6362 participants with at least one valid measure of accelerometer variables during the 13-year follow-up. However, only 2457 of these 6362 had complete DEXA measures of fat mass and lean mass only across T1, T2, and T3 clinic visits. In the earlier inclusion criteria, 917 participants had at least 2 valid accelerometer timepoint measures and complete DEXA measured total fat mass in addition to body mass index and waist circumference across T1, T2, and T3 clinic visits. We used multiple imputations to account for missing exposures and covariates in the 2457 participants' analyses. We presented the new findings in Tables 4 and 5 and Supplemental Tables 8 – 11. Overall, the results of the 2457 participants were consistent with the 917 participants.

This then leads into questions about informative missing data. I appreciate that there are already a considerable number of statistical analyses here, but pattern-mixture models (or another approach to NMAR) would seem prudent to see how robust the findings are to informative missing data mechanisms and could be useful in the discussion. Note that when presenting results after using MI, it is conventional (at least in my circles) to show the MI-based results in the manuscript (as you do) but also to show CC-based results in a supplement to help readers understand how missing data could have (subject to assumptions) distorted results and sometimes to show the robustness of the findings to assumptions around missing data.

Response: We presented a complete case analysis of the original 917 participants' imputed analysis in Supplemental Table 7. However, only 192 participants had complete predictors, covariates, and outcomes at all time points (T1, T2, and T3). In the unadjusted model of the complete case analysis, the results were similar to the imputed results. However, in the fully adjusted model, results were statistically non-significant for total fat mass but not lean mass. The direction of the association was similar in the imputed and complete case analysis.

The description on Lines 193–195 doesn't, for me, explain exactly how MI was done so that a reader would know (at least roughly) what code they would have used in their preferred statistical software. If providing this level of detail becomes cumbersome in the text, providing code in a supplement might help? It's unclear precisely what "especially" (Line 195) means. If variables on both sides of the equation were imputed, which is my presumption, what does this mean? A little more detail about the bootstrapped mediation models (around Line 232) would also be useful (Preacher-Hayes?), along with a reference or two for the reader wishing to understand exactly what was done here (I do appreciate the authors explaining more in the text before this point). Related to this, I'd avoid absolute phrases such as "no mediating effect" when the values listed include non-zero. The definition of what constitutes mediation and suppression (Lines 223–229) could be expanded to distinguish between non, partial, and total (e.g., Lines 307–325 might indicate an implicit rule that <1% = no [which I would avoid, perhaps "minimal" or similar], 1%+ = partial, but if so this should be made explicit in the methods), although there is plenty of debate about this, if you decided to move some of the more complicated methods to a supplement as I suggest.

Response: We have included more details on the handling of missing data in the methods section as copied below.

“Handling of missing covariates and multiple imputations

Eligible sample size varied by covariates and exclusions via listwise deletion of missing values ranged from 27.6 to 50.1 percent for covariates at either 11, 15 or 24 years clinic visit. We restricted study participants to those who had complete outcome variables at all study time points at ages 11, 15, and 24 years. We conducted a Little's missing completely at random (MCAR) test to ascertain data missingness.²² Little's MCAR test: Chi-Square = 1159.60, degree of freedom = 260, p-value <0.0001, made us conclude that the variables were not missing completely at random. Regression-modeled multiple imputations were conducted using SPSS version 27 (IBM Corp, Armonk, NY, USA). The observed minimum and maximum values were constraints for the imputation process and 20 cycles of imputation with 10 iterations resulted in 20 imputed data sets. The multiple imputation module in SPSS pooled the results from these imputed data. In line with previous evidence,¹⁷ the percentage of missing values would be sufficiently addressed with 20 imputations: the variable with the highest missing value (50.1%, maternal socioeconomic status) had an estimate that was 98% efficient after 20 imputations (computed using Rubin's formula).²² The distributions of imputed variables had the same pattern as in the observed data as evidenced in the histogram normality plot. Where multiple imputations have been conducted, presenting imputed results is preferred to presenting non-imputed results.²³”

We have added two references on mediation and included a sentence that *“We considered a statistically significant mediation or suppression of <1% as minimal and ≥1% as partial.”* ...

Preacher KJ, Hayes AF. Asymptotic and resampling strategies for assessing and comparing indirect effects in multiple mediator models. *Behav Res Methods*. 2008;40:879-891. doi:10.3758/BRM.40.3.879

Preacher KJ, Hayes AF. SPSS and SAS procedures for estimating indirect effects in simple mediation models. *Behav Res Methods, Instruments, Comput*. 2004;36:717-731. doi:10.3758/BF03206553

I appreciate the use of Model 1 and Model 2, presumably to reduce the risk of overadjustment in Model 2, but Model 1 has several time-varying variables that could potentially be on the causal pathway between exposure and outcome (e.g., blood glucose and insulin for PA -> body composition). Indeed, this is part of the research question addressed in the manuscript. Is there a reason for not including a Model 0 (with only baseline/unchanging characteristics included or reflecting “Equation 3” from the Figures)? Model 2 is challenging to interpret as, e.g., a decrease in either/both LPA or/and MVPA would presumably automatically increase ST if sleep was not affected (or increase non-wear times)? I also wonder if readers will not perhaps be confused by the table note “For continuous variable predictors (ST, LPA and MVPA), a 1-standard deviation change is associated with a 1-standard deviation change in the outcome” without this being related back to the reported coefficients.

Response: We have included an unadjusted model in the revised Model 1 and other Models 2 to 4 with varying degrees of adjustments as shown in Table 2.

The main models seem to have a single slope for the PA and biochemistry variables over all three ages. Assuming this is the case, are the authors confident that, e.g., the effect of ST on blood glucose might not vary over the three ages, i.e., did they consider and perhaps investigate effect modification by age? I

appreciate that the presence of such interactions would complicate the presentation of the results, but I feel that such interactions would also be of interest and importance.

Response: We have conducted separate analyses using only variables at 15 and 24 years and presented in Supplemental Tables 6 and 11 and the results were consistent with the main analyses of ages 11 through 24 years data. This potentially suggests little or no variation in the effect of PA on biochemistry variables across the 3 ages.

This leads to an important question about sleep for me. Sleep (both quantity and quality) seem especially plausible mediators between the PA and composition variables (as well as potentially independent predictors of the outcomes here). The lack of sleep data seems a major limitation of the present research and I was very surprised to be unable to find any instances of the word “sleep” (or its inflected forms) in the manuscript or the supplement.

Response: Accelerometer sleep data were unavailable and this has been added to the study limitation.

I think readers will appreciate a clearer identification of the family for the multiplicity adjustment (Lines 259–260, and table notes). In theory, at least, the family could be at the regression model level, table level, or any of a number of other levels.

Response: As stated earlier more than 98% of the included children are from separate mothers and families.

I appreciated the authors considering the adequacy of their sample size for their proposed analyses (Lines 260–262) but I’m not sure that these will assist readers, particularly in the context of the present research with its clustering and multiplicity adjustments. Could this be made more specific (including using your actual analytic sample size) and perhaps easier for readers to understand? Ultimately, the power of your analyses is now entirely communicated through your confidence interval limits, so perhaps explaining this, along with calculations prior to performing the analyses, would be useful.

Response: We have cited the power calculation for ALSPAC study prior to performing the analyses. We are not quite clear on this suggestion, whether we are to include another power calculation after the analyses in the method section. We will appreciate further clarification.

Table 3 stratifies analyses by sex, but doesn’t examine evidence for effect modification by sex (which would be “sex-specific” rather than “sex-based”, c.f. title and Lines 204–205 and 303–304). This might be easier to include by rearranging the tables so that males and females are columns. With appropriate interactions, results for the two sexes and evidence for effect modification could be extracted from a single model in each case.

Response: Based on the new results presented in Table 5 in the expanded cohort of 2457 participants, there were no sex differences in the longitudinal relation of MVPA with total fat mass. This is possibly due to the larger sample size among males 334 vs 964 compared with previous Table 3, now Supplemental Table 2.

I appreciate the authors overall care around causality in places, but I feel that “causal effects” (Line 41) and similar overstate their ability to draw causal claims from observational data here. With an underlying causal model, they certainly can investigate consistency between this model and their data, but given the likelihood of unmodelled confounding (I felt Lines 427–430 was rather optimistic), I would carefully remove all causal language (including for me “confirms” on Line 372) aside from where it is clearly about potential mechanisms or speculative (e.g., Lines 81–83 are clearly about mechanisms and Lines 85–89 are perfectly fine to me). This includes the conclusion about associations being “unlikely [to be] causal” (Line 54) and even “may be causally associated” (Line 55) unless stronger arguments (around confounding bias being

sufficiently low to support these) can be added. Lines like 247–248’s “changes in the earlier variables are considered to lead to changes in the later, and vice versa.” in particular stood out to me as claiming much too much for observational data.

Response: We have revised the text accordingly.

I appreciated you providing references to articles that informed your underlying causal model (Lines 258–259) but I wonder if a visual representation (e.g., a directed acyclic graph) here would help readers to understand the minimum adjustment set for each exposure:outcome pairing. This could also go in a supplement, with a more thorough explanation of the variables in the models, but I personally like to see a DAG in any observational study of moderate complexity or above as a matter of course.

Response: A DAG on the relationship of PA and fat mass has been provided in Supplemental Figure 2.

It wasn’t clear to me why the goodness of fit indices described on Lines 252–255 seemed specific to the cross-lagged models and not the mediation models. Personally, I wouldn’t see these as necessary in general given the seeming lack of latent variables. Regression model diagnostics would suffice for me here if this is the case.

Response: The indices have been removed.

I was uncertain whether “cumulative” here (e.g., Lines 145 and 209) meant more than “at each age”, particularly for ST, LPA, and MVPA.

Response: ‘Cumulative’ meant values combined for all ages rather than a specific value at a specific age, using data output from long-format restructured analyses.

.....
Reviewer #2 (Remarks to the Author):

General comments:

This is an interesting paper that uses data from the UK AVON birth cohort study to explore the overall and sex-specific longitudinal associations between accelerometer assessed movement behaviours and adiposity from age 11y to 24y. There is a lack of longitudinal research using quality objective measures, so this paper fills this gap. I do have specific comments below, however, about some of the analyses in terms of timing of data collection and claims about causality. A general comment is there is a lot in this paper, and I wonder for the sake of coherence and the sheer volume of analyses performed if the number of tests could be reduced (eg, the mediation and categorical analyses, and even the sex specific analyses which are not discussed at all or mentioned in conclusions).

Response: We thank the reviewer for the compliment and helpful suggestions. We have conducted additional mediation analyses and added discussion on sex-specific analyses.

Specific comments:

1. Paper Title: While I understand the analyses provide an estimation of “causal effects”, I have concerns about this term (described further in comment #2 below) and wonder if the term is appropriate here?

Response: As suggested by all reviewers we have added 'potential' to the title due to the limitation of observational design regarding residual confounding from unmeasured variables.

2. Abstract, line 32: The authors need to be careful in use of the term cause here as longitudinal data are still observational. The analytic technique applied in the paper can facilitate understanding of causal pathways (potential causality), but not prove them. I recommend toning down claims regarding cause-effect throughout the paper (including the title).

Response: We have included potential causality across the text.

3. Abstract, lines 33-35: Please amend the aims here to include mention of sex specific analyses and also exploring potential 'reverse causality'.

Response: The aims have been revised accordingly.

4. Abstract, lines 39-40: Please mention who's guidelines. The WHO?

Response: The World Health Organization has been mentioned in the abstract.

5. Abstract, lines 49-50: "Cumulative insulin, glucose, lipids, and low-grade inflammation differently mediated the associations of movement behaviours with body composition". The wording in this sentence is awkward, please amend.

Response: The sentence has been revised to "*The inverse associations of MVPA with total fat mass was partially mediated by increased high-density lipoprotein cholesterol (3.4% mediation) and lean mass (7.5% mediation).*"

6. Abstract, lines 51-52: I was surprised by the mention of fat mass predicting MVPA because this wasn't mentioned in the aims (see comment #3).

Response: We have updated the study aims to include examining potential reverse causation.

7. Abstract, lines 53-54: "ST increases and LPA decreases fat mass from childhood through young adulthood, but the relationship is unlikely causal." I don't think this statement can be made based on the study design. Please amend this statement.

Response: We have removed the later part of the sentence.

8. Abstract, lines 54-56: I have concerns about the mediating analysis because of the timing of the collection of bloods. This conclusion may need to change.

Response: We acknowledge the limitation of the non-availability of fasting blood samples at baseline measures of PA and fat mass. However, when we conducted the mediating analyses only with PA, fat mass, and blood samples using only 2-time measures at 15 and 24 years for predictor, mediator, and outcomes the results were substantially similar. Furthermore, we repeated the mediation analyses in the expanded cohort of 2457 participants who had at least one PA valid measure and the results were consistent. Also, we repeated the T2 and T3 specific mediation analyses including only blood samples at 15 and 24 years with PA and fat mass at 15 and 24 years in the expanded cohort and the results were similar to previous findings. These additional mediation analyses have been presented in Supplementary Tables 6 and 11, and Table 5. We observed that the repeated mediation analyses using different population sizes and time points are consistent.

9. Keywords: I don't think Health Promotion is an appropriate key word for this paper.

Response: Health promotion has been removed.

10. Introduction, line 77: The updated WHO PA guidelines have removed the need for this to be every day. The updated version recommends at least 60 minutes/day of moderate-to-vigorous PA (MVPA) on average.

Response: The sentence has been updated accordingly.

11. Introduction, lines 84-90: I understand that word limits in journals make it difficult to provide a comprehensive rationale leading to the study aims, but I think some mention of the biological mediating pathways is important in the Introduction. For example, what are some biological pathways that mediate associations between movement behaviours and adiposity so that there is no direct causal pathway (what is being argued in the conclusions I believe)?

Response: We have included a brief sentence in the introduction, copied below.

“A few biological pathways on the associations between ST and adiposity identified in an experimental animal model and small sample-sized human studies include, elevated inflammation, altered lipid and glucose metabolism, and muscular atrophy.⁴⁻⁷”

The cited references are:

Bey L, Hamilton MT. Suppression of skeletal muscle lipoprotein lipase activity during physical inactivity: a molecular reason to maintain daily low-intensity activity. *J Physiol.* 2003;551(Pt 2):673-682. doi:10.1113/jphysiol.2003.045591

Bunprajun et al. Lifelong Physical Activity Prevents Aging-Associated Insulin Resistance in Human Skeletal Muscle Myotubes via Increased Glucose Transporter Expression. *PLoS One.* 2013;8(6):e66628. doi:10.1371/journal.pone.0066628

Yaribeygi et al. Pathophysiology of Physical Inactivity-Dependent Insulin Resistance: A Theoretical Mechanistic Review Emphasizing Clinical Evidence. *J Diabetes Res.* 2021;2021:7796727. doi:10.1155/2021/7796727

Burini et al. Inflammation, physical activity, and chronic disease: An evolutionary perspective. *Sport Med Heal Sci.* 2020;2(1):1-6. doi:10.1016/j.smhs.2020.03.004

12. Introduction: In relation to the above point, a brief rationale is needed for the sex specific analyses. While many studies report boys/men are more active than girls/women no study is cited to support this aim. Furthermore, what is the rationale for a sex-specific longitudinal association and/or mediating pathway?

Response: We have included a brief rationale as copied below.

“It is known that males are generally more physically active than females, potentially yielding better health among males.^{1,3} However, it is important to clarify whether exposure to movement behaviour typical of a particular sex, significantly influences body composition during growth from childhood through young adulthood.

13. Introduction, line 96: Aim 3 needs to mention also exploring the potential reverse associations between adiposity and PA/ST.

Response: We have included potential reverse causality in the aim #3 sentence.

14. Methods, lines 133-135: It would be helpful to mention consistency in assessing movement behaviours between these two accelerometer models.

Response: We have included the following sentence.

“There is a strong absolute agreement between the Actigraph™ models (intraclass correlation coefficient 0.99 (95% CI = 0.98–0.99) thus making it acceptable to use different models within a study.^{12”}

Robusto KM, Trost SG. Comparison of three generations of ActiGraph™ activity monitors in children and adolescents. *J Sports Sci.* 2012;30(13):1429-35. doi: 10.1080/02640414.2012.710761.

15. Methods, lines 143-144: Please report what published accelerometer algorithm (cutpoints) was used for the 11yr olds and 15yr olds. Reference 8 is suitable for the adult cutpoint, I don't believe the other references (9 and 10) are validity studies?

Response: We have provided additional details as copied below and updated the references.

“Activity counts per minute threshold validated in children and adolescents were used to calculate the amount of time spent; MVPA, >2296 counts per minute (cpm); for LPA, 100 – 2296 cpm; and for ST, 0 – <100 cpm at ages 11 and 15 years using the Evenson cutpoint whereas, at the 24-year assessment, the 2020 cpm Troiano cut point was used.¹³⁻¹⁵ The Evenson cutpoint used in stratifying activity threshold has shown the best overall performance across all intensity levels and was suggested as the most appropriate cut point for youth.^{14,15”}

Troiano RP, Berrigan D, Dodd KW, Mâsse LC, Tilert T, Mcdowell M. Physical activity in the United States measured by accelerometer. *Med Sci Sports Exerc.* 2008;40(1):181-188. doi:10.1249/mss.0b013e31815a51b3

Trost SG, Loprinzi PD, Moore R, Pfeiffer KA. Comparison of accelerometer cut points for predicting activity intensity in youth. *Med Sci Sports Exerc.* 2011;43(7):1360-1368. doi:10.1249/MSS.0b013e318206476e

Migueles JH, Cadenas-Sanchez C, Ekelund U, et al. Accelerometer Data Collection and Processing Criteria to Assess Physical Activity and Other Outcomes: A Systematic Review and Practical Considerations. *Sport Med.* 2017;47(9):1821-1845. doi:10.1007/s40279-017-0716-0

16. Methods, lines 145-146: Please report which timepoint the correlations between intensities are taken from.

Response: The correlation was taken from the cumulative data including 11, 15 and 24 years after data restructure. However, we have included each timepoint correlation as copied below.

“Pearson bi-variate correlations (r^2) between cumulative 11, 15, and 24 years ST and LPA are (-0.70), ST and MVPA (-0.33), LPA and MVPA (0.19), p -value <0.001 for all. For each age clinic visit, at age 11 years, the correlations between ST and LPA are (-0.59), ST and MVPA (-0.27), LPA and MVPA (0.12), p -value <0.001 for all. At age 15 years clinic visit, the correlations between ST and LPA are (-0.47), ST and MVPA (-0.35), LPA and MVPA (0.35), p -value <0.001 for all. At age 24 years clinic visit, the correlations between ST and LPA are (-0.18), p = 0.009, ST and MVPA (-0.22), p = 0.002, LPA and MVPA (0.14), p = 0.043.”

17. Methods, lines 148-149: These categories of PA are not recommended by the WHO, this reference needs to be removed from this sentence.

Response: The reference has been removed.

18. Methods, lines 149-150: I have a few issues with the categories, both in interpreting the analyses (more on that later) and in understanding whether this was completed at each time point? And if tertiles were created at each time point, because of the declining & then increasing PA levels with age, I assume that at T2 and T3 the tertiles may not have aligned with the guidelines so well?

Response: The MVPA tertile categories were selected based on values at the baseline age 11 years clinic visits where participants accumulated the highest minutes of MVPA. These tertile values were maintained for each T2 and T3 visit irrespective of the decline in PA. Thus, the interpretation favours a persistent MVPA of 60mins/day across T1, T2, and T3 in the general population rather than a growth curve tracking of individual patterns.

19. Methods, line 155: The BMI acronym was already defined in the Introduction

Response: Repeated definition has been removed.

20. Methods, lines 162-174, and lines 200-202: The use of data collected at timepoints (13yrs and 17yrs) that differ from the exposure and outcome variables (11yrs, 15yrs, 24yrs) is I think problematic. While family medical history (collected at 17yrs) is probably ok to include as a covariate, I have concerns about including bloods from the data collection at 17yrs when no corresponding physical activity or adiposity data have been collected. This is a 2 yr gap from the 15yr old timepoint and 7 year gap to the 24 yr old timepoint. Given how much PA changes with age, having this time point included as a mediator assumes from a causal perspective steady state of PA/ST between 15yrs and 17yrs? Also smoking was assessed at 13yrs and 17yrs as well as 24yrs. Although this is just a covariate, only 1 of these time points lines up with the exposure and outcome data.

Response: We acknowledge the limitation of the non-availability of fasting blood samples at baseline measures of PA and fat mass. However, when we conducted the mediating analyses only with PA, fat mass, and blood samples using only 2-time measures at 15 and 24 years for predictor, mediator, and outcomes the results were substantially similar. Furthermore, we repeated the mediation analyses in the expanded cohort of 2457 participants who had at least one PA valid measure and the results were consistent. Also, we repeated the T2 and T3 specific mediation analyses including only blood samples at 15 and 24 years with PA and fat mass at 15 and 24 years in the expanded cohort and the results were similar to previous findings. These additional mediation analyses have been presented in Supplementary Tables 6 and 11, and Table 5. We observed that the repeated mediation analyses using different population sizes and time points are consistent.

We corrected the typo error regarding smoking variables which were available at 13, 15, and 24 years.

21. Methods, lines 187-188 and lines 203-204: I am concerned about adjustments for all intensities in Model 2. The high correlation between several intensities (especially ST and LPA) is a concern, especially with a longitudinal study where changes in one intensity will be related to changes in another. It is for this reason that more recently researchers have been using compositional analyses which is better equipped to handle inter-related (in this case time bound) data. Is it possible to provide a reference to support the statement that “The GLMM is robust for handling highly correlated variables such as ST and LPA”?

Response: A reference has been provided.

Schielzeth H, Dingemanse NJ, Nakagawa S, et al. Robustness of linear mixed-effects models to violations of distributional assumptions. Methods Ecol Evol. 2020;11:1141– 1152. doi:10.1111/2041-210X.13434

22. Methods, lines 208-210: As noted in comment #20, the collection of bloods at 17 yrs means there is a misalignment here between the PA/ST exposures and the adiposity outcomes with relation to the mediators. Given the declines in PA in youth (eg, from 15 yrs to 17yrs), does this model assume some stability in PA behaviour given the bloods were collected at 17 yrs? How were the bloods from the 2 time points included in the model? I don't think the mediating analysis is strong given this limitation and suggest it is removed.

Response: Reviewer #3 noted that “Mediation analyses are a major part of the manuscript, which adds to the strengths of the work.”

We acknowledge the limitation of the non-availability of fasting blood samples at baseline measures of PA and fat mass. However, when we conducted the mediating analyses only with PA, fat mass, and blood samples using only 2-time measures at 15 and 24 years for predictor, mediator, and outcomes the results were substantially similar. Furthermore, we repeated the mediation analyses in the expanded cohort of 2457 participants who had at least one PA valid measure and the results were consistent. Also, we repeated the T2 and T3 specific mediation analyses including only blood samples at 15 and 24 years with PA and fat mass at 15 and 24 years in the expanded cohort and the results were similar to previous findings. These additional mediation analyses have been presented in Supplementary Tables 6 and 11, and Table 5. We observed that the repeated mediation analyses using different population sizes and time points are consistent.

23. Methods, line 256: “Collinearity diagnoses were performed and accepted results with a variance inflation factor <5”. Please clarify which models had a VIF greater than 5.

Response: No model had a VIF greater than 5.

24. Results, lines 284-285: What about the number of participants who provided data at 17 yrs? This should also be presented in Supplemental Figure 1.

Response: We have added the number of participants who attended the 17 years clinic visit and provided blood samples to the footnote of Supplemental Figure 1

25. Results, lines 290-291: The J or U-shaped increase in PA from 15 yrs to 24 yrs could be due to changes in cut-points from child to adult. One of the challenges with longitudinal studies from early childhood to childhood or from childhood to adulthood is the change in threshold counts means it is 'easier' for young adults to accumulate MVPA mins due to the lowering of the threshold. Did you also look at CPM? I would be surprised if PA actually increased from 15 yrs to 24 yrs.

Response: The mean (SD) count per mins in the entire cohort is as follows.

Age 11 years: 607.54 (399.01) cpm

Age 15 years: 467.20 (587.50) cpm

Age 24 years: 402.80 (201.29) cpm

26. Discussion: There is no discussion at all about how the findings differed by sex.

Response: While the analyses among 917 participants showed that there are differences by sex, the larger sample size of 2457 participants with less strict inclusion criteria of at least one accelerometer valid measure showed that sex difference does not exist. The difference earlier seen was probably due to a lower sample size among males. We have included this observation in the discussion.

“While the results in 917 participants with at least two time-point valid accelerometer measures suggested that MVPA was associated with lower total fat mass in females but not in males, the analyses using a larger sample size of 2457 participants with less strict inclusion criteria of at least one time-point accelerometer measure indicates that increased MVPA was associated with lower total fat mass in both males and females. This disparity in findings is likely due to a lower sample size, i.e 334 males of 917 participants in comparison with 964 males of 2457 participants.”

27. Discussion, lines 361-364 and lines 401-402: Please re-word these sentences for ease of reading.

Response: The sentences have been reworded.

28. Discussion, lines 413-414: As per comment #25, different count thresholds may also explain this apparent U/J-shaped curve.

Response: We have added this suggestion to the discussion. *“Another explanation for the U or J-shaped curve may be due to different cutpoints used in adolescence (MVPA >2296 cpm) and young adulthood (MVPA >2020 cpm), since the average counts per mins in the entire cohort steadily declined across the 13-year follow-up period.”*

29. Discussion, line 426: The mis-alignment between data collection of bloods at 17y with accelerometer and adiposity data is a major limitation which I think impacts credibility of the mediation analyses.

Response: As stated above in response to comment #22 we have added this limitation to the discussion and how it was potentially addressed.

30. Discussion, line 434 and line 439: I disagree with these conclusions about causality, particularly if this is based on the mediation analysis which used bloods collected at a different time point from the PA/adiposity measures (eg 17yrs).

Response: We have explained the limitation of blood collection at different timepoint and how we partly addressed the problem in our response to comment #22.

31. Table 1: The ‘mean (SD)’ at the top of each column in Table 1 don’t align with all the variables underneath (eg, %s). I suggest removing these and instead put ‘mean (SD)’ alongside relevant rows for continuous variables. Also, the sub-heading ‘Anthropometry’ doesn’t seem to describe the variables sitting under it. Perhaps change to Sociodemographic characteristics?

Response: We have removed the mean (SD) as suggested, and moved ethnicity to the sociodemographic factor sub-heading.

32. Table 1: It seemed from the Methods that sociodemographic information was collected at 17yrs of age, not at 11yrs?

Response: Sociodemographic information were collected in childhood and not at age 17 years.

33. Table 2: “Categorical cumulative predictor variable from ages 11 – 24 years”. I had trouble understanding the categorical analysis and didn’t see a clear explanation of it in the results. Because participants can move from one category to another at each time point, how are the changes in categories (or stability) over time handled & interpreted? Is this analysis needed given how much is already in this paper?

Response: The MVPA tertile categories were selected based on values at the baseline age 11 years clinic visits where participants accumulated the highest minutes of MVPA. These tertile values were maintained for each T2 and T3 visit irrespective of the decline in PA. Thus, the interpretation favours a persistent MVPA of 60mins/day across T1, T2, and T3 in the general population rather than a growth curve tracking of individual patterns. The analysis is important to buttress the current WHO recommendation of an average of 60 mins/day of MVPA.

.....

Reviewer #3 (Remarks to the Author):

This study examined the associations of cumulative sedentary time (ST), light physical activity (LPA), and moderate to vigorous physical activity (MVPA) from age 11 – 24 years with repeated measures of body mass index (BMI), and measures of body composition assessed by dual-energy Xray absorptiometry. The authors found that over a 13-year follow-up, ST increase was associated with increased adiposity while cumulative LPA was associated with lower adiposity. Furthermore, persistent exposure to =60mins/day of MVPA was associated with decreasing total and truncal adiposity. Authors also report that MVPA may causally lower total fat mass via an increase in high-density lipoprotein cholesterol and lean mass. This is an important study providing novel information of significance that can inform physical activity guidelines and interventions. The manuscript is clearly written. However, I do have the following comments.

Response: We thank the reviewer for the compliment and helpful suggestions.

It would be informative to provide estimates of minimally adjusted or crude models in addition to models 1 and 2.

Response: An unadjusted model has been presented as the new Model 1, while model 1 is now model 2 and the former model 2 has been updated as Models 3 and 4 as presented in Table 2.

Model 1 is adjusted for the hypothesized mediators, glucose and insulin, which is not recommended. In contrast, height adjustment should be considered. For example, waist circumference is also determined by height.

Response: If we correctly understand the reviewer is referring to the GLMM longitudinal associations where we adjusted for all possible covariates. The adjustment modality changed in the mediation model analysis. In the mediation model, we did not adjust for both glucose and insulin while either was the mediator. In our study population, we have previously shown a very high correlation >0.95 between height and DEXA-measured lean mass. Hence, we did not include both height and lean mass in the same model. *Hypertens Res.* 2023. doi: 10.1038/s41440-022-01065-1.

Mediation analyses are a major part of the manuscript, which adds to the strengths of the work. However, I do have a few comments that may help to improve the interpretation of the findings. It would be good to follow the consensus-based reporting guideline for studies reporting mediation analyses (A Guideline for Reporting Mediation Analyses; AGRReMA): <https://jamanetwork.com/journals/jama/fullarticle/2784353>

Response: We thank the reviewer for this helpful suggestion. We have aligned our manuscript with the guideline and cited it in the text.

In particular, it would be helpful to provide a directed acyclic graph (DAG) to represent the causal relationship between variables. Also, the theory that underpins the proposed mechanisms of interest in the mediation analyses should be described and evidenced.

Response: We have provided a directed acyclic graph in the supplemental appendix as supplementary figure 2. A briefly proposed mechanism has been added to the introduction and method section.

In Figure 1, it would be preferable to report 95% CI in addition to (or instead of) of p-values.

Response: For the temporal analysis the software generated standard errors, standardized and unstandardized regression coefficients, and p-values but not confidence intervals.

Discussion, line 361, it is not clear what the authors meant with “confounding longitudinal analyses”.

Response: This refers to the GLMM model which is a confounding-based analyses. The other two analyses were mediation and temporal analyses. We have added confounding-based to the sentence.

Interpretation of “suppressive roles” is unclear and not common in a mediation context. Could the authors add a brief explanation.

Response: We have added a brief explanation to the previous information provided in the method as copied below.

“This means that suppression occurs when the mediational path has an opposite effect, i.e. instead of a decrease in the point estimate of the direct effect between an exposure and an outcome in relation to the total effect, there is rather an increase in the direct effect above the total effect’s point estimate.”

MacKinnon DP, Krull JL, Lockwood CM. Equivalence of the mediation, confounding and suppression effect. *Prev Sci.* 2000;1(4):173-181. doi:10.1023/a:1026595011371

How do the authors explain the “suppressive role” of ST through insulin with total fat mass? One would expect a positive association between ST and insulin levels.

Response: While this mechanism is not completely understood, we have provided a brief explanation and references in the discussion as copied below.

“The mediation analyses showed that a higher insulin concentration which could reflect hyperinsulinemia was associated with higher total body fat mass. However, higher ST was associated with decreased insulin concentration. Evidence suggests that a 6-day period of physical inactivity reduces insulin action, through the reduction in Glut-4 levels in endurance-trained runners’ skeletal muscle.^{18,28} ST has been associated with increased endoplasmic reticulum oxidative stress, inflammation, and mitochondrial dysfunction, which could result in beta cell insufficiency and decreased insulin secretion.^{18,29} It is however plausible that optimal insulin levels in relatively healthy young populations may offer minimal protection against the deleterious effect of ST on body fat since the magnitude of the suppression effect of insulin on the relationship between ST and fat mass was 7.5%.”

Bunprajun et al. Lifelong physical activity prevents aging-associated insulin resistance in human skeletal muscle myotubes via increased glucose transporter expression. *PLoS One.* 2013;8(6, article e66628) doi: 10.1371/journal.pone.0066628.

Slentz et al. Effects of exercise training intensity on pancreatic beta-cell function. *Diabetes Care.* 2009 Oct;32(10):1807-11. doi: 10.2337/dc09-0032.

Could the authors explain a bit more how ST would increase fat mass via increased muscle mass?

Response: We have provided a brief explanation as copied below.

“We observed that increased ST was associated with increased fat mass via an increase in muscle mass. While the pathophysiological mechanism may not be fully understood, it is known that skeletal muscle mass regulation of lipoprotein lipase activity is impaired during physical inactivity.^{18,30} Lipoprotein lipase influences cholesterol metabolism, the partitioning of triglyceride-derived fatty acid uptake between different tissues, downstream intracellular effects related to lipid availability, alterations in muscle glucose, and fatty acid metabolism.^{30”}

Yaribeygi et al. Pathophysiology of Physical Inactivity-Dependent Insulin Resistance: A Theoretical Mechanistic Review Emphasizing Clinical Evidence. *J Diabetes Res.* 2021 Oct 7;2021:7796727. doi: 10.1155/2021/7796727.

Bey L, Hamilton MT. Suppression of skeletal muscle lipoprotein lipase activity during physical inactivity: a molecular reason to maintain daily low-intensity activity. *J Physiol*. 2003 Sep 1;551(Pt 2):673-82. doi: 10.1113/jphysiol.2003.045591.

Could the authors add a brief explanation in line 433-434 as to why the relationships are "...unlikely causal".

Response: We have added a brief explanation as follows.

ST increase was associated with increased adiposity while cumulative LPA was associated with lower adiposity in the confounding-based and mediation analyses, but there was no evidence of temporal and potential reverse causation which may suggest a slight insufficient likelihood of potential causal relationships.

Despite the longitudinal study design, causality in observed associations is uncertain, because unmeasured confounding or other biases may distort the observed associations. This should probably be emphasized more in the limitations and use of causal language tempered in the conclusions (abstract and main text). Furthermore, it is unclear how authors judge which of the relationships were deemed causal while others were not.

Response: Results that showed consistency across the confounding, mediation, and temporal analyses were deemed potentially causal. We have added the limitation of observational designs as it relates to residual biases due to unmeasured confounders in the discussion. We have also added "potential" to the conclusion, abstract and main text.

In the abstract, please provide the unit in the reported "effect estimate".

Response: The reported estimates are standardized (Z-scores).

Reviewers' Comments:

Reviewer #1:

Remarks to the Author:

Thank you for your responses to my comments and the revised version of your manuscript. As with the initial version, I enjoyed reading your work, and thinking about your findings and their implications. I found many of your revisions to be thoughtful and constructive and have fewer comments on this new version of your work. However, there are still aspects of the statistical analyses and the data used that remain unclear to me. These comments are again from my perspective as a biostatistician whose work involves childhood and adult obesity in a public health context.

Overall, I still feel that the details of the statistical analyses need to be increased in places (I would not expect to be able to reproduce all analyses in the situation where, and I appreciate that data cannot be shared, I had the data, at least not without trial and error) and I still cannot understand why the primary analyses do not use all participants with exposure data at at least one time and outcome data at at least one time. Based on Figure S1, it seems that at least 6000 participants would meet the first criterion and over 7000 the second. All of these participants would contribute at least some information using MI or FIML, with some lower number contributing through ML alone. Finally, I remain unconvinced that family as a source of clustering shouldn't be incorporated and I think that not considering potentially informative data is a limitation.

At the same time, I also still feel that this is a valuable data set and that the questions asked here are important (and provide some answers that are needed in the literature), so I hope that my comments will be useful to the authors in thinking about any potential revisions.

I appreciate the challenges with data availability. It's a shame that SPSS Amos doesn't provide code (as a clear and unambiguous description of the analyses, something that can be difficult to achieve in prose) but that's not something in the authors' control. This does mean, however, that the text needs to be especially clear, and I will raise specific points where I think that clarity could be enhanced in comments below.

While compositional and non-compositional analyses can sometimes produce similar results, as the authors point out, I wouldn't take that as evidence that the same would apply for a specific analysis. I appreciate the approach of looking at a single PA measure, then adding one of the other PA measures, and then the last of the PA measures. This still makes Models 3 and 4 complicated by the compositional nature of the data. I wasn't sure that I followed the logic of Models 3 and 4 here. From the note under Table 2, Model 3 for ST would have to involve adding LPA (ST being already in the model), and Model 4 would have to involve adding MVPA. However, for MVPA, it's not clear to me from the note for Model 3 which (of ST and LPA) would be added, except that the note for Model 4 would require LPA to be added there, making ST the only possible choice for Model 3. It's not quite Sudoku, but perhaps represents more of a puzzle than might be intended. Ultimately, it's not clear to me what Model 3 offers to the reader as an intermediate step to Model 4. I feel I understand Models 1 and 2; and while I still have significant concerns with it, I feel I also understand Model 4; but what does Model 3 add in the authors' view?

When it comes to Model 4, and this reflects my biases as a biostatistician, when faced with two alternative approaches where one is "more correct" in terms of the structure of the data and the other is an approach that often gives very similar findings, I still have a very strong preference for the analysis that better matches the data generating process/mechanism behind the data. If this non-compositional approach is used, I think it warrants some consideration in the discussion at the very least, but I would still find some of the coefficients in the tables to be challenging to interpret. For example, in Table 2, for ST in Model 4, having 100 minutes more ST per day would be associated with BMI being higher by 2.1 points, but this is conditional on no change in LPA or MVPA (or any of the other variables in the model). For small amounts of time (a few minutes, say), this "holding everything else constant" doesn't seem unreasonable or too difficult to imagine; but I would struggle to increase my ST by nearly 2 hours in a counterfactual without decreasing my LPA/MVPA. The same would apply for increasing my MVPA by 100 minutes per day in a counterfactual. Model 4 would predict my BMI to be 1.0 points lower in that case compared to

myself without that increase in MVPA, but this also seems difficult to achieve without me changing some of my ST and/or LPA into MVPA time. These are not small changes in my PA behaviour, but they are certainly not impossible differences between people (some people could do 100 minutes more MVPA than others and others could do 100 minutes more ST than others, the lower levels of PA seem especially plausible to me for someone with family commitments, an injury, or a long commute, for example).

Tables 3 and 5 still appears to include the other two PA variables for each analysis (this is my reading of the table notes). The note under Table 4 seemed ambiguous to me ("and additional adjustment for either sedentary time, light physical activity, or moderate to vigorous physical activity depending on the predictor.") as "either" suggests only one other variable, but I wonder if you might mean both other variables?

I can't say that the results would change using a compositional approach, but I don't think I can assume that Models 3 and 4 would be unchanged with such an approach either. To be unambiguous, I think that compositional data analysis would be best practice here if more than one PA variable was to be included at the same time.

For the Model 1 versus Model 2 distinction, this certainly helps with avoiding over-adjustment through mediation, but some of the variables in Model 2 (sex, age, family history), cannot be mediators (unless family history was updated). SES, if it was updated at each age, but Lines 206–207 don't suggest that for me, could in theory be a mediator, but this would be a surprising mechanism. At least some if not all of these could always be safely include in Model 1, which should improve precision there (particular the additions of age and sex, I would expect). Lines 246–250 suggest to me that all of the listed variables are time-varying, but sex seems unlikely to be so and I can't see it clearly stated that family history and SES varied over time (neither of these are listed after age 11 in Table 1 and so I am assuming they do not vary over time). Similar would apply to the table notes.

I appreciate that family-level clustering would be unlikely to affect results beyond a very slight widening of the CIs (from your comment, it sounds as if 2% of the children were from families with multiple study members). But it's not a matter of this being "applicable". It is part of the data structure. If this is not modelled, that might be a pragmatic decision to avoid making the models more complex (were there issues with convergence?), but that would add a brief limitation to the discussion for me (along with the number of such children in this study). Again, I think best practice would be to include the family-level clustering on the basis that it will either leave the results effectively unchanged or improve their validity.

I wonder if my point about multiple imputation was confusing. MI is usually used to impute both dependent and independent variables (imputing the latter generally adds information, imputing the former can be useful for longitudinal data). Participants with no outcome/dependent variable data will not provide useful information, but those with any outcome data (at at least one of T1, T2, or T3) can have their missing outcome data imputed based on the available outcome data. If 6362 participants had at least some exposure data, I'm unable to see why this was reduced to the 2457 with all three clinic visits when any one clinic visit would make them informative. Am I misunderstanding here? Lines 40–41 seem to make it clear to me that more participants were excluded than necessary and I cannot see why the analyses on the largest set of participants wouldn't be the primary analyses (Line 37 onward) rather than included in repeated analysis (Line 40).

For complete case (CC) analyses, given the use of mixed models, it doesn't seem necessary here for participants to have data for all three outcome times (one timepoint with both independent and dependent variables would be sufficient for that participant to be included in the analyses), so there were presumably more than the 192 participants in Table S7 with complete predictors, covariates, and outcome data at at least one of T1, T2, and/or T3? A complete case analysis here would, for me, include all participants who could be included in analyses without imputation or FIML, i.e., those who could be included using ML. I'm afraid that I still cannot understand why you restricted analyses to participants with data at all three times. I also couldn't understand the statement on Line 382 that the CC results were "significantly attenuated ... due to low sample

size". Sample size will affect the precision, but I'm not following the apparent connection between sample size and point estimates that you seem to be making here.

If no informative missing data analyses are included, and I can understand the desire not to further complicate these analyses, this should still be noted as a limitation in the discussion, I feel.

My comment about the family for multiplicity adjustment ("I think readers will appreciate a clearer identification of the family for the multiplicity adjustment (Lines 259–260, and table notes). In theory, at least, the family could be at the regression model level, table level, or any of a number of other levels.") was referring to the groupings of analyses that were considered together (i.e. what was considered a family of hypothesis tests) when performing adjustments for multiplicity, not the families of the study participants.

My question about the power for this particular study was whether there had been a specific power calculation (or other sample size adequacy checks) prior to this particular study. This is often required by funders and ethics committees in my experience, and sometimes is done just to make sure that the available data (not that this is amenable to change) will likely be enough before undertaking the considerable analysis work. From your response, it sounds as if this was not the case? If so, this should not be done now (there would be no stop/go decision to be made at this point). Rather than readers wondering if this detail was not included, I wonder if it would be prudent to state it outright (i.e., that no formal power calculations were conducted prior to this study), perhaps along with my suggestion that the effective power of this study is communicated through the precision of estimates (such as the CI widths).

I appreciated the additional information about the multiple imputation, but I would still like to know about the variables in the imputation model (Line 217 doesn't make this clear). Were these the same as in your analyses or were auxiliary variables also included?

As a very, very minor point, Line 233 should be "value" not "valve".

Lines 233–234: I still think some clarity is needed here since skew in variables isn't problematic for statistical modelling in itself. Skew in model residuals might indicate skew in the population error term, and that could motivate transformations (as long as these match the understanding of the mechanisms/data generating process).

I'm not sure what you mean in your response by "non-linear data". At the risk of pedantry, data cannot be linear or non-linear; associations can be these things. If you used generalised linear mixed models (Line 239), rather than general linear mixed models, you will need to state the link used and the error family. Given that these models appear to be for BMI, etc. as the dependent variables (Lines 237–238), I remain confused by this terminology but providing these details should remove my confusion.

While I don't disagree with Ref 24 about the dangers of collinearity often being overstated (and this applies in general to regression modelling), I don't see any justification for saying "The GLMM is robust for handling highly correlated variables such as ST and LPA" (Lines 239–240). In my view, this statement would be equally true for a fixed effects only model as for a mixed effects model. In other words, can the authors explain why they appear to suggest that a GLMM is more robust to collinearity in its fixed effects than a GLM?

Lines 242–243: A GLMM (generalised or general linear) cannot handle "missing at random predictor and covariate data" unless full information maximum likelihood is used. I don't use SPSS/Amos so I'm not aware of its defaults, but I think sufficient detail needs to be included here so that a reader (who will not have the data in hand) could feel that if they did have the data in hand, they would be able to reproduce the analyses using whatever software they know (within reason). At the moment, I lack this confidence myself.

For Table 3, I would still like to see p-values for effect modification by sex. The text on this table (Lines 303–304) would be usefully extended by evaluating the evidence for such effect modification.

Reviewer #2:

Remarks to the Author:

The authors have done substantial work to address my comments and those of the other reviewers. The paper is much stronger and I think will make an important contribution to the longitudinal physical activity literature. I have no additional comments to make.

Reviewer #3:

Remarks to the Author:

The authors made great efforts to improve the manuscript. Most of my comments have been satisfactorily addressed in the revised version.

However, the reporting of the mediation analysis remained rather vague and the AGRReMA guideline was only partially followed.

It would be recommended to specify in the methods part whether controlled direct effect, natural direct and indirect effects, or interventional direct and indirect effects are estimated and presented.

Table 2 reports results of 4 different models, of which models 3 and 4 are little informative. Instead, and as suggested in my previous report, results without adjustment of hypothesized mediators should be reported. This would allow, together with the crude model 1, judgement of the degree of confounding. As it is now reported, one is left wondering whether the huge drop in the beta-estimate (comparing Model 1 to Model 2) is due to substantial confounding or due to strong mediation. For example, the beta-estimate for sedentary time associated with total fat mass dropped from 0.216 to 0.023 (Table 2). However, given that the % mediated association hardly exceeds 7.5%, one is left to assume that there is substantial residual confounding in presented associations.

It is unclear which estimates Table 2 is aiming to present; given the adjustments, one can assume it shows the direct effects, but this should be explicitly stated.

Similarly, Table 4 shows results from a crude model, which is appreciated, but then only the results of a model with adjustment for hypothesized mediators. The authors may therefore state that the estimates refer to some direct effect.

The footnotes of the Figures (2-4) should include information about adjustments made and what types of associations were estimated with the different equations (total effect, exposure-mediator effect, mediator-outcome effect, controlled direct effect, natural direct and indirect effects, or interventional direct and indirect effects are estimated and presented).

Response to reviewers' comments

We are grateful for the compliments and helpful suggestions. Kindly find below the point-by-point response.

Reviewer #1 (Remarks to the Author):

Thank you for your responses to my comments and the revised version of your manuscript. As with the initial version, I enjoyed reading your work, and thinking about your findings and their implications. I found many of your revisions to be thoughtful and constructive and have fewer comments on this new version of your work. However, there are still aspects of the statistical analyses and the data used that remain unclear to me. These comments are again from my perspective as a biostatistician whose work involves childhood and adult obesity in a public health context.

Overall, I still feel that the details of the statistical analyses need to be increased in places (I would not expect to be able to reproduce all analyses in the situation where, and I appreciate that data cannot be shared, I had the data, at least not without trial and error) and I still cannot understand why the primary analyses do not use all participants with exposure data at at least one time and outcome data at at least one time. Based on Figure S1, it seems that at least 6000 participants would meet the first criterion and over 7000 the second. All of these participants would contribute at least some information using MI or FIML, with some lower number contributing through ML alone. Finally, I remain unconvinced that family as a source of clustering shouldn't be incorporated and I think that not considering potentially informative data is a limitation.

Response 1

We have provided additional analyses which included 6059 participants that had at least one movement behaviour measure and fat mass measure and presented in Supplemental Table 12.

There were missing data (predictor, covariates, and outcomes) up to a maximum of 88% missing variables at a particular time point. Hence, we could not consider the result of the 6059 participants as our primary analysis since readers who are mostly not biostatisticians are likely to question conclusions based on such a large amount of imputed data.

Supplemental Table 12 Longitudinal associations of cumulative sedentary time and physical activity with body composition in 6059 participants who had at least one time point measure of movement behaviour and fat mass during ages 11 through 24 years

N=6059	Body mass index (kg/m ²)		Total fat mass (kg)		Lean mass (kg)	
	β (95% CI)	p-value	β (95% CI)	p-value	β (95% CI)	p-value
Continuous cumulative predictor variables from ages 11 – 24 years						
Sedentary Time (mins/day)						
Model 1	0.147 (0.140 – 0.155)	<0.0001	0.122 (0.115 – 0.130)	<0.0001	0.226 (0.218 – 0.234)	<0.0001
Model 2	0.024 (0.019 – 0.029)	<0.0001	0.020 (0.015 – 0.026)	<0.001	0.034 (0.030 – 0.039)	<0.0001
Light Physical Activity (mins/day)						
Model 1	-0.163 (-0.170 – -0.155)	<0.0001	-0.136 (-0.144 – -0.129)	<0.0001	-0.240 (-0.248 – -0.232)	<0.0001
Model 2	-0.028 (-0.033 – -0.023)	<0.0001	-0.026 (-0.032 – -0.020)	<0.0001	-0.037 (-0.042 – -0.032)	<0.0001
Moderate to Vigorous Physical Activity (mins/day)						
Model 1	-0.045 (-0.054 – -0.036)	<0.0001	-0.055 (-0.065 – -0.046)	<0.0001	-0.057 (-0.068 – -0.046)	<0.0001
Model 2	-0.005 (-0.011 – 0.000)	0.057	-0.024 (-0.031 – -0.017)	<0.001	0.006 (0.001 – 0.011)	0.030

Model 1 was unadjusted. Model 2 was adjusted for sex, and other time-varying covariates measured at both baseline and follow-up such as age, low-density lipoprotein cholesterol, triglyceride, high sensitivity C-reactive protein, high-density lipoprotein cholesterol, heart rate, systolic blood pressure, glucose, insulin, smoking status, family history of hypertension/diabetes/high cholesterol/vascular disease, socioeconomic status, and fat mass or lean mass, depending on the outcome and additional adjustment for both sedentary time and light physical activity or moderate to vigorous physical activity depending on the predictor. Body mass index model was not adjusted for fat mass and lean mass. Skewed covariates were logarithmically transformed. Standardized regression coefficients (β) were computed from the generalized linear mixed-effect model for repeated measures, direct effect estimates are presented; CI, confidence interval. A 2-sided P-value <0.05 is considered statistically significant. Multiple testing was corrected with Sidak correction. Multiple imputations were used to account for missing variables. For continuous variable predictors (ST, LPA and MVPA), a 1-standard deviation change is associated with a 1-standard deviation change in the outcome.

Also provided below is the analysis in 2457 participants that included family clustering.

N=2457	Total fat mass (kg)	
	β (95% CI)	p -value
Continuous cumulative predictor variables from ages 11 – 24 years		
Sedentary Time (mins/day)		
Family cluster	0.021 (0.014 – 0.029)	<0.001
Without family cluster	0.021 (0.013 – 0.029)	<0.001
Light Physical Activity (mins/day)		
Family cluster	-0.015 (-0.023 – -0.007)	<0.001
Without family cluster	-0.013 (-0.021 – -0.005)	<0.001
Moderate to Vigorous Physical Activity (mins/day)		
Family cluster	-0.042 (-0.049 – -0.035)	<0.0001
Without family cluster	-0.021 (-0.031 – -0.011)	<0.001

At the same time, I also still feel that this is a valuable data set and that the questions asked here are important (and provide some answers that are needed in the literature), so I hope that my comments will be useful to the authors in thinking about any potential revisions.

I appreciate the challenges with data availability. It's a shame that SPSS Amos doesn't provide code (as a clear and unambiguous description of the analyses, something that can be difficult to achieve in prose) but that's not something in the authors' control. This does mean, however, that the text needs to be especially clear, and I will raise specific points where I think that clarity could be enhanced in comments below.

While compositional and non-compositional analyses can sometimes produce similar results, as the authors point out, I wouldn't take that as evidence that the same would apply for a specific analysis. I appreciate the approach of looking at a single PA measure, then adding one of the other PA measures, and then the last of the PA measures. This still makes Models 3 and 4 complicated by the compositional nature of the data. I wasn't sure that I followed the logic of Models 3 and 4 here. From the note under Table 2, Model 3 for ST would have to involve adding LPA (ST being already in the model), and Model 4 would have to involve adding MVPA. However, for MVPA, it's not clear to me from the note for Model 3 which (of ST and LPA) would be added, except that the note for Model 4 would require LPA to be added there, making ST the only possible choice for Model 3. It's not quite Sudoku, but perhaps represents more of a puzzle than might be intended. Ultimately, it's not clear to me what Model 3 offers to the reader as an intermediate step to Model 4. I feel I understand Models 1 and 2; and while I still have significant concerns with it, I feel I also understand Model 4; but what does Model 3 add in the authors' view?

Response 2:

Thank you for the detailed explanations. Model 3 was necessary to show the distinct contributions of each of the activity levels. As the reviewer noted, for analysis of MVPA as the predictor, Model 3 was adjusted for ST, while Model 4 was additionally adjusted for LPA. ST and LPA are inversely related and combining them as Model 4 only would not provide information on each of the activity level contributions. We have added more clarification to the table footnotes.

We consider that this approach addresses the "specific research recommendations for children and adolescents (5–17 years)" by the WHO physical activity and sedentary behavior guidelines development group which states that "Conduct adequately-powered observational studies to examine the independent

and joint effects of physical activity and sedentary behavior on health outcomes in children and adolescents.”

DiPietro et al. Advancing the global physical activity agenda: recommendations for future research by the 2020 WHO physical activity and sedentary behavior guidelines development group. Int J Behav Nutr Phys Act. 2020 Nov 26;17(1):143. doi: 10.1186/s12966-020-01042-2.

When it comes to Model 4, and this reflects my biases as a biostatistician, when faced with two alternative approaches where one is “more correct” in terms of the structure of the data and the other is an approach that often gives very similar findings, I still have a very strong preference for the analysis that better matches the data generating process/mechanism behind the data. If this non-compositional approach is used, I think it warrants some consideration in the discussion at the very least, but I would still find some of the coefficients in the tables to be challenging to interpret. For example, in Table 2, for ST in Model 4, having 100 minutes more ST per day would be associated with BMI being higher by 2.1 points, but this is conditional on no change in LPA or MVPA (or any of the other variables in the model). For small amounts of time (a few minutes, say), this “holding everything else constant” doesn’t seem unreasonable or too difficult to imagine; but I would struggle to increase my ST by nearly 2 hours in a counterfactual without decreasing my LPA/MVPA. The same would apply for increasing my MVPA by 100 minutes per day in a counterfactual. Model 4 would predict my BMI to be 1.0 points lower in that case compared to myself without that increase in MVPA, but this also seems difficult to achieve without me changing some of my ST and/or LPA into MVPA time. These are not small changes in my PA behaviour, but they are certainly not impossible differences between people (some people could do 100 minutes more MVPA than others and others could do 100 minutes more ST than others, the lower levels of PA seem especially plausible to me for someone with family commitments, an injury, or a long commute, for example).

Tables 3 and 5 still appears to include the other two PA variables for each analysis (this is my reading of the table notes). The note under Table 4 seemed ambiguous to me (“and additional adjustment for either sedentary time, light physical activity, or moderate to vigorous physical activity depending on the predictor.”) as “either” suggests only one other variable, but I wonder if you might mean both other variables?

I can’t say that the results would change using a compositional approach, but I don’t think I can assume that Models 3 and 4 would be unchanged with such an approach either. To be unambiguous, I think that compositional data analysis would be best practice here if more than one PA variable was to be included at the same time.

Response 3

Thank you for the explanation.

The note under Table 3, 5 “and additional adjustment for either sedentary time, light physical activity, or moderate to vigorous physical activity depending on the predictor” mean both other variables. This implies that if MVPA is the predictor, there were adjustments for both sedentary time and light physical activity. Where sedentary time is the predictor, there were adjustments for both light physical activity and moderate to vigorous physical activity. We have clarified this in the footnote.

We did not aim to compare standard analytical strategy with compositional data analysis. Gupta et al 2018 compared these strategies and the results of both approaches were similar with compositional data yielding a 0.004 to 0.007 unit higher effect size, and sometimes equal effect size when compared to standard regression analysis effect size.

Last week we published a paper in *JCEM* on longitudinal movement behaviour and inflammation using standard regression analysis and concluded that “increased LPA had a two-fold inflammatory lowering effect and was more resistant to the attenuating effect of fat mass compared with MVPA.”

Two days later a different group from Spain (*Segura-Jiménez et al.*) published a similar paper in *IJBNPA* using compositional data analysis on longitudinal movement behaviours which included sleep time and inflammation in children and adolescents and concluded that “reallocating time away from LPA appears to be most consistently unfavourably associated with inflammatory markers.”

Unfortunately, we do not have data on sleep time so we are unable to examine how changes in other activities within a 24-hour period will affect other activity patterns. Thus, compositional data using the available variables will assume that sleep time remains constant while sedentary time, Light PA, or MVPA changes. The lack of sleep time data limits compositional data analysis as 100% daily activity would be based on available activity data rather than on a 24-hour daily activity. We, therefore, consider that the current standard regression approach is sufficient.

Gupta et al. A comparison of standard and compositional data analysis in studies addressing group differences in sedentary behavior and physical activity. Int J Behav Nutr Phys Act. 2018 Jun 15;15(1):53. doi: 10.1186/s12966-018-0685-1.

Agbaje AO. Longitudinal Mediating effect of Fatmass and Lipids on Sedentary Time, Light PA, and MVPA with Inflammation in Youth. J Clin Endocrinol Metab. 2023 Jun 13;dgad354. doi: 10.1210/clinem/dgad354. Epub ahead of print.

Segura-Jiménez et al. Longitudinal reallocations of time between 24-h movement behaviours and their associations with inflammation in children and adolescents: the UP&DOWN study. Int J Behav Nutr Phys Act. 2023 Jun 15;20(1):72.

For the Model 1 versus Model 2 distinction, this certainly helps with avoiding over-adjustment through mediation, but some of the variables in Model 2 (sex, age, family history), cannot be mediators (unless family history was updated). SES, if it was updated at each age, but Lines 206–207 don't suggest that for me, could in theory be a mediator, but this would be a surprising mechanism. At least some if not all of these could always be safely include in Model 1, which should improve precision there (particular the additions of age and sex, I would expect). Lines 246–250 suggest to me that all of the listed variables are time-varying, but sex seems unlikely to be so and I can't see it clearly stated that family history and SES varied over time (neither of these are listed after age 11 in Table 1 and so I am assuming they do not vary over time). Similar would apply to the table notes.

Response 4.

SES and family history did not vary over time. We have clarified this in the results footnote. In previous comments, other reviewers' specifically requested to provide results for unadjusted model.

I appreciate that family-level clustering would be unlikely to affect results beyond a very slight widening of the Cis (from your comment, it sounds as if 2% of the children were from families with multiple study members). But it's not a matter of this being “applicable”. It is part of the data structure. If this is not modelled, that might be a pragmatic decision to avoid making the models more complex (were there issues with convergence?), but that would add a brief limitation to the discussion for me (along with the number of such children in this study). Again, I think best practice would be to include the family-level clustering on the basis that it will either leave the results effectively unchanged or improve their validity.

Response 5:

Analysis with and without family clustering is presented below.

N=2457	Total fat mass (kg)	
	β (95% CI)	p-value
Continuous cumulative predictor variables from ages 11 – 24 years		
Sedentary Time (mins/day)		
Family cluster	0.021 (0.014 – 0.029)	<0.001
Without family cluster	0.021 (0.013 – 0.029)	<0.001
Light Physical Activity (mins/day)		
Family cluster	-0.015 (-0.023 – -0.007)	<0.001
Without family cluster	-0.013 (-0.021 – -0.005)	<0.001
Moderate to Vigorous Physical Activity (mins/day)		
Family cluster	-0.042 (-0.049 – -0.035)	<0.0001
Without family cluster	-0.021 (-0.031 – -0.011)	<0.001

I wonder if my point about multiple imputation was confusing. MI is usually used to impute both dependent and independent variables (imputing the latter generally adds information, imputing the former can be useful for longitudinal data). Participants with no outcome/dependent variable data will not provide useful information, but those with any outcome data (at at least one of T1, T2, or T3) can have their missing outcome data imputed based on the available outcome data. If 6362 participants had at least some exposure data, I'm unable to see why this was reduced to the 2457 with all three clinic visits when any one clinic visit would make them informative. Am I misunderstanding here? Lines 40–41 seem to make it clear to me that more participants were excluded than necessary and I cannot see why the analyses on the largest set of participants wouldn't be the primary analyses (Line 37 onward) rather than included in repeated analysis (Line 40).

Response 6:

We have repeated the analysis in 6059 participants who had at least 1 measure of movement behaviour and one time-point measure of fat mass. The result is now presented in Supplemental Table 12 as shown in Response 1 above.

For complete case (CC) analyses, given the use of mixed models, it doesn't seem necessary here for participants to have data for all three outcome times (one time-point with both independent and dependent variables would be sufficient for that participant to be included in the analyses), so there were presumably more than the 192 participants in Table S7 with complete predictors, covariates, and outcome data at at least one of T1, T2, and/or T3? A complete case analysis here would, for me, include all participants who could be included in analyses without imputation or FIML, i.e., those who could be included using ML. I'm afraid that I still cannot understand why you restricted analyses to participants with data at all three times. I also couldn't understand the statement on Line 382 that the CC results were "significantly attenuated ... due to low sample size". Sample size will affect the precision, but I'm not following the apparent connection between sample size and point estimates that you seem to be making here.

If no informative missing data analyses are included, and I can understand the desire not to further complicate these analyses, this should still be noted as a limitation in the discussion, I feel.

Response 7:

Unfortunately, by default the SPSS software estimates complete case GLMM analysis by listwise deletion rather than pairwise deletion, and there is no option to change the setting, hence the significantly reduced participants.

My comment about the family for multiplicity adjustment (“I think readers will appreciate a clearer identification of the family for the multiplicity adjustment (Lines 259–260, and table notes). In theory, at least, the family could be at the regression model level, table level, or any of a number of other levels.”) was referring to the groupings of analyses that were considered together (i.e. what was considered a family of hypothesis tests) when performing adjustments for multiplicity, not the families of the study participants.

My question about the power for this particular study was whether there had been a specific power calculation (or other sample size adequacy checks) prior to this particular study. This is often required by funders and ethics committees in my experience, and sometimes is done just to make sure that the available data (not that this is amenable to change) will likely be enough before undertaking the considerable analysis work. From your response, it sounds as if this was not the case? If so, this should not be done now (there would be no stop/go decision to be made at this point). Rather than readers wondering if this detail was not included, I wonder if it would be prudent to state it outright (i.e., that no formal power calculations were conducted prior to this study), perhaps along with my suggestion that the effective power of this study is communicated through the precision of estimates (such as the CI widths).

Response 8:

With the SPSS default GLMM model option dialog box, multiplicity adjustment was applied at the regression model level. We utilized the power calculations already published for the ALSPAC data 20 years ago, which covers the scope of the current study. The power calculation was conducted for different ALSPAC population sizes and whether the outcome is continuous or categorical. This was already cited in the methods. We have submitted this reference to funders and have not been requested to conduct additional power calculations for every new analysis involving the same cohort.

Golding G, Pembrey P, Jones J. ALSPAC - The Avon Longitudinal Study of Parents and Children I. Study methodology. Paediatr Perinat Epidemiol. 2001;15(1):74-87. doi:10.1046/j.1365-3016.2001.00325.x

I appreciated the additional information about the multiple imputation, but I would still like to know about the variables in the imputation model (Line 217 doesn't make this clear). Were these the same as in your analyses or were auxiliary variables also included?

Response 9:

The same variables in the analyses were used in the imputation model including measured auxiliary variables like diastolic blood pressure, height, and weight.

As a very, very minor point, Line 233 should be “value” not “valve”.

Response 10:

This has been corrected, thanks.

Lines 233–234: I still think some clarity is needed here since skew in variables isn't problematic for statistical modelling in itself. Skew in model residuals might indicate skew in the population error term, and that could motivate transformations (as long as these match the understanding of the mechanisms/data generating process).

Response 11:

We have clarified in the text that skewness referred to model residuals.

I'm not sure what you mean in your response by "non-linear data". At the risk of pedantry, data cannot be linear or non-linear; associations can be these things. If you used generalised linear mixed models (Line 239), rather than general linear mixed models, you will need to state the link used and the error family. Given that these models appear to be for BMI, etc. as the dependent variables (Lines 237–238), I remain confused by this terminology but providing these details should remove my confusion.

Response 12:

A generalised linear mixed model with an identity link was used. We have added this to the method.

While I don't disagree with Ref 24 about the dangers of collinearity often being overstated (and this applies in general to regression modelling), I don't see any justification for saying "The GLMM is robust for handling highly correlated variables such as ST and LPA" (Lines 239–240). In my view, this statement would be equally true for a fixed effects only model as for a mixed effects model. In other words, can the authors explain why they appear to suggest that a GLMM is more robust to collinearity in its fixed effects than a GLM?

Response 13:

The SPSS manual states "Generalized linear mixed models extend the linear model so that: The target is linearly related to the factors and covariates via a specified link function, The target can have a non-normal distribution, The observations can be correlated". We did not write or suggest that a "GLMM is more robust to collinearity when compared to GLM."

https://www.ibm.com/docs/en/SSLVMB_27.0.0/pdf/en/IBM_SPSS_Advanced_Statistics.pdf

<https://www.ibm.com/docs/en/spss-modeler/18.2.2?topic=node-generalized-linear-mixed-models>

Lines 242–243: A GLMM (generalised or general linear) cannot handle "missing at random predictor and covariate data" unless full information maximum likelihood is used. I don't use SPSS/Amos so I'm not aware of its defaults, but I think sufficient detail needs to be included here so that a reader (who will not have the data in hand) could feel that if they did have the data in hand, they would be able to reproduce the analyses using whatever software they know (within reason). At the moment, I lack this confidence myself.

Response 14:

The sentence has been modified as "GLMM using full information maximum likelihood is robust for handling missing at random predictor and covariate data"

For Table 3, I would still like to see p-values for effect modification by sex. The text on this table (Lines 303–304) would be usefully extended by evaluating the evidence for such effect modification.

Response 15:

Effect modification by sex for sedentary time as predictor is ($p = 0.881$) and moderate to vigorous physical activity as predictor is ($p = 0.414$) considering fat mass as outcome.

.....

Reviewer #2 (Remarks to the Author):

The authors have done substantial work to address my comments and those of the other reviewers. The paper is much stronger and I think will make an important contribution to the longitudinal physical activity literature. I have no additional comments to make.

Response 16:

We thank the reviewer for the helpful suggestions and compliments.

.....

Reviewer #3 (Remarks to the Author):

The authors made great efforts to improve the manuscript. Most of my comments have been satisfactorily addressed in the revised version.

However, the reporting of the mediation analysis remained rather vague and the AGReMA guideline was only partially followed.

It would be recommended to specify in the methods part whether controlled direct effect, natural direct and indirect effects, or interventional direct and indirect effects are estimated and presented.

Response 17:

Thank you for the compliments and helpful suggestions. We have specified that natural direct and indirect effects were estimated and presented and cited the following references.

VanderWeele TJ. Controlled direct and mediated effects: definition, identification and bounds. Scand Stat Theory Appl. 2011 Sep;38(3):551-563. doi: 10.1111/j.1467-9469.2010.00722.x.

Richiardi et al. Mediation analysis in epidemiology: methods, interpretation and bias. Int J Epidemiol. 2013 Oct;42(5):1511-9. doi: 10.1093/ije/dyt127.

Table 2 reports results of 4 different models, of which models 3 and 4 are little informative. Instead, and as suggested in my previous report, results without adjustment of hypothesized mediators should be reported. This would allow, together with the crude model 1, judgement of the degree of confounding. As it is now reported, one is left wondering whether the huge drop in the beta-estimate (comparing Model 1 to Model 2) is due to substantial confounding or due to strong mediation. For example, the beta-estimate for sedentary time associated with total fat mass dropped from 0.216 to 0.023 (Table 2). However, given that the % mediated association hardly exceeds 7.5%, one is left to assume that there is substantial residual confounding in presented associations.

Response 18:

The current analysis for model 3 (adjustment for sedentary time when LPA is the predictor) and model 4 (adjustment for MVPA when is the predictor) helps clarify the independent and combined contribution of

different activity levels as stated in the “specific research recommendations for children and adolescents (5–17 years)” by the WHO physical activity and sedentary behavior guidelines development group. It was recommended to “Conduct adequately-powered observational studies to examine the independent and joint effects of physical activity and sedentary behavior on health outcomes in children and adolescents.”

DiPietro et al. Advancing the global physical activity agenda: recommendations for future research by the 2020 WHO physical activity and sedentary behavior guidelines development group. Int J Behav Nutr Phys Act. 2020 Nov 26;17(1):143. doi: 10.1186/s12966-020-01042-2.

Kindly see below Table 2 (model 2) with the results without adjustments for hypothesized mediators. Slight differences in point estimates compared with the result in the main article are due to the peculiarity of repeated multiple imputations yielding different imputed values.

Model 1 is unadjusted.

Model 2 adjustments for confounders (age, sex, socioeconomic status, smoking status, heart rate, systolic blood pressure, family history of hypertension/diabetes/high cholesterol/vascular disease, light physical activity, and additional adjustment for either sedentary time, light physical activity or moderate to vigorous physical activity depending on the predictor).

Model 3 is an additional adjustment of model 2 for hypothesized mediators (lean mass, low-density lipoprotein cholesterol, triglyceride, high sensitivity C-reactive protein, high-density lipoprotein cholesterol, glucose, insulin, lean mass).

N=917	Total fat mass (kg)	p -value
	β (95% CI)	
Continuous cumulative predictor variables from ages 11 – 24 years		
Sedentary Time (mins/day)		
Model 1	0.216 (0.195 – 0.236)	<0.0001
Model 2	0.010 (-0.005 – 0.010)	0.189
Model 3	0.016 (0.001 – 0.030)	0.031
Light Physical Activity (mins/day)		
Model 1	-0.270 (-0.289 – -0.251)	<0.0001
Model 2	-0.013 (-0.027 – -0.009)	0.096
Model 3	-0.018 (-0.032 – -0.004)	0.011
Moderate to Vigorous Physical Activity (mins/day)		
Model 1	-0.050 (-0.076 – -0.023)	<0.001
Model 2	-0.012 (-0.032 – 0.008)	0.249
Model 3	-0.014 (-0.034 – 0.005)	0.158

Another analysis with initial adjustments for hypothesized mediators (model 2) before adjustments for confounders.

Model 1 is unadjusted.

Model 2 is additional adjustment of model 2 for hypothesized mediators (lean mass, low-density lipoprotein cholesterol, triglyceride, high sensitivity C-reactive protein, high-density lipoprotein cholesterol, glucose, insulin, lean mass).

Model 3 is an adjustment of model 2 **for confounders** (*age, sex, socioeconomic status, smoking status, heart rate, systolic blood pressure, family history of hypertension/diabetes/high cholesterol/vascular disease, light physical activity, and additional adjustment for either sedentary time, light physical activity or moderate to vigorous physical activity depending on the predictor*).

N=917	Total fat mass (kg)	
	β (95% CI)	p-value
Continuous cumulative predictor variables from ages 11 – 24 years		
Sedentary Time (mins/day)		
Model 1	0.216 (0.195 – 0.236)	<0.0001
Model 2	0.020 (0.002 – 0.039)	0.034
Model 3	0.016 (0.001 – 0.030)	0.031
Light Physical Activity (mins/day)		
Model 1	-0.270 (-0.289 – -0.251)	<0.0001
Model 2	-0.061 (-0.079 – -0.043)	<0.001
Model 3	-0.018 (-0.032 – -0.004)	0.011
Moderate to Vigorous Physical Activity (mins/day)		
Model 1	-0.050 (-0.076 – -0.023)	<0.001
Model 2	-0.006 (-0.016 – 0.027)	0.618
Model 3	-0.014 (-0.034 – 0.005)	0.158

It is unclear which estimates Table 2 is aiming to present; given the adjustments, one can assume it shows the direct effects, but this should be explicitly stated.

Similarly, Table 4 shows results from a crude model, which is appreciated, but then only the results of a model with adjustment for hypothesized mediators. The authors may therefore state that the estimates refer to some direct effect.

Response 19:

We have included in Tables 2 and 4 footnotes that the estimates refer to direct effect.

The footnotes of the Figures (2-4) should include information about adjustments made and what types of associations were estimated with the different equations (total effect, exposure-mediator effect, mediator-outcome effect, controlled direct effect, natural direct and indirect effects, or interventional direct and indirect effects are estimated and presented).

Response 20:

We have included additional information in Figures 2-4 footnotes.

Reviewers' Comments:

Reviewer #1:

Remarks to the Author:

Thank you for your rebuttal and revisions. I greatly appreciated the additional explanations and results.

In particular, thank you for the additional analyses on 2457 participants that incorporated family clustering (Page 2 of your rebuttal). For sedentary activity, there was no meaningful change (coefficients were identical to 2 significant figures); and for LPA, the change in estimate looked to be around 15% in relative terms (-0.015/-0.013, so somewhere between 7% and 24% depending on further decimal places); however, for MVPA, the change in estimate was around 100% (-0.042/-0.021). While this doesn't affect statistical significance, the effect sizes seem key to me here and these do appear to vary considerably. Surely a literal doubling of the amount of fat mass per unit of MVPA is of clinical significance? Since the clustered analyses are mathematically "more correct" and seem to change the results, this makes me even more certain that all analyses ought to be adjusting for this aspect of the study design. Showing that incorporating clustering doesn't change statistical significance doesn't mean that incorporating clustering doesn't change the results in important ways.

I'm also still unconvinced as to why the 6059 participants with at least some activity and body composition data could not be used for the primary analyses. In a way, I understand your argument that "since readers who are mostly not biostatisticians are likely to question conclusions based on such a large amount of imputed data", but it is standard practice in (bio)statistics to consider the analyses using imputed data as the primary (since they ought to be closest to the "truth" on average) and to present complete-case analyses in a supplement (in part as a sensitivity analysis but also to facilitate exploring the missingness mechanism itself). As you note on Lines 203–204, "Where multiple imputations have been conducted, presenting imputed results is preferred to presenting non-imputed results." and Lines 127–129 and 297–298 sound very much like a sensitivity analysis to me. I appreciated the additional results on Lines 408–415, but was disappointed that the differences did not seem addressed in the discussion. The coefficients in Supplementary Table 12 (page 1 of your rebuttal) change to a meaningful extent in places. Setting aside the effect on statistical significance, it is again the effect sizes (the practical or clinical significance of the findings) that are key here for me. What is the reader to make of these differences?

The responses to point 2 (about compositional analyses), in particular the importance of considering the order of adding activity variables, strengthens my conviction that compositional analyses should be used throughout. I take your point around sleep time, but in any case, I struggle with the interpretation of a coefficient, say comparing myself against my counterfactual who is identical except that they have a substantially higher MVPA, say 60 minutes more, when the coefficients presented assume that my and their Sedentary and LPA times are identical. Including other variables in the model here means that to interpret one exposure/predictor variable's coefficient, we need to imagine holding all of the other variables' coefficients constant. Even if the results were similar (or even equivalent), with a compositional data analysis approach, the "more correct" strategy is still my preference.

There are three aspects of the analyses here that I still think could/need to be improved (clustering within families, multiple imputation, and compositional analyses). I am most sympathetic to your points around the compositional aspects. While I do appreciate your responses and additional analyses, the relative changes in coefficients from MI and adding family clustering, alongside the challenges of interpreting differences in one PA component while holding the others constant, leaves me still wanting these aspects incorporated, or for stronger justifications for not doing so.

I appreciate that I run the risk of becoming, if I have not already done so, dogmatic here, but I think that there are strong arguments for the "most correct" statistical analyses being included as the primary analyses here. As noted above, I feel less strongly about the compositional analyses than the other two aspects.

Reviewer #3:

Remarks to the Author:

I thank the authors for addressing my remaining concerns. Congratulations to this important work.
No further comments.

Response to reviewers' comments

We are grateful to the reviewers for their time and suggestions. We have conducted additional analyses among 6059 participants with family clustering as the primary analyses, kindly see Tables 1 - 3. Moreover, we presented compositional data analysis results in Supplementary Table 14.

Reviewer #1 (Remarks to the Author):

Thank you for your rebuttal and revisions. I greatly appreciated the additional explanations and results.

In particular, thank you for the additional analyses on 2457 participants that incorporated family clustering (Page 2 of your rebuttal). For sedentary activity, there was no meaningful change (coefficient were identical to 2 significant figures); and for LPA, the change in estimate looked to be around 15% in relative terms (-0.015/-0.013, so somewhere between 7% and 24% depending on further decimal places); however, for MVPA, the change in estimate was around 100% (-0.042/-0.021). While this doesn't affect statistical significance, the effect sizes seem key to me here and these do appear to vary considerably. Surely a literal doubling of the amount of fat mass per unit of MVPA is of clinical significance? Since the clustered analyses are mathematically "more correct" and seem to change the results, this makes me even more certain that all analyses ought to be adjusting for this aspect of the study design. Showing that incorporating clustering doesn't change statistical significance doesn't mean that incorporating clustering doesn't change the results in important ways.

Response 1:

The updated primary analyses involving 6059 participants incorporated family clustering as suggested. Kindly see the updated Table 2 below.

Table 2 Longitudinal associations of cumulative sedentary time and physical activity with body composition from ages 11 through 24 years.

N=6059	Body mass index		Waist circumference		Trunk fat mass	
	β (95% CI)	p -value	β (95% CI)	p -value	β (95% CI)	p -value
Continuous cumulative predictor variables from ages 11 – 24 years						
Sedentary Time (mins/day)						
Model 1	0.142 (0.135 – 0.150)	<0.0001	0.144 (0.134 – 0.153)	<0.0001	0.131 (0.123 – 0.138)	<0.0001
Model 2	0.027 (0.021 – 0.032)	<0.0001	-0.011 (-0.017 – -0.006)	<0.001	0.026 (0.020 – 0.032)	<0.0001
Model 3	0.020 (0.015 – 0.025)	<0.001	-0.019 (-0.024 – -0.014)	<0.001	0.020 (0.014 – 0.025)	<0.001
Model 4	0.019 (0.014 – 0.024)	<0.001	-0.017 (-0.022 – -0.012)	<0.001	0.017 (0.012 – 0.023)	<0.001
Light Physical Activity (mins/day)						
Model 1	-0.162 (-0.170 – -0.155)	<0.0001	-0.215 (-0.225 – -0.204)	<0.0001	-0.149 (-0.156 – -0.141)	<0.0001
Model 2	-0.035 (-0.040 – -0.030)	<0.0001	-0.031 (-0.037 – -0.026)	<0.0001	-0.033 (-0.039 – -0.028)	<0.0001
Model 3	-0.031 (-0.036 – -0.025)	<0.0001	-0.035 (-0.041 – -0.030)	<0.0001	-0.029 (-0.035 – -0.024)	<0.0001
Model 4	-0.030 (-0.035 – -0.025)	<0.0001	-0.036 (-0.042 – -0.031)	<0.0001	-0.028 (-0.034 – -0.023)	<0.0001

Moderate to Vigorous Physical Activity (mins/day)						
Model 1	-0.048 (-0.057 – -0.039)	<0.0001	-0.009 (-0.019 – 0.001)	0.076	-0.061 (-0.070 – -0.051)	<0.0001
Model 2	-0.014 (-0.020 – -0.008)	<0.001	0.019 (0.014 – 0.024)	<0.001	-0.027 (-0.034 – -0.020)	<0.001
Model 3	-0.011 (-0.017 – -0.005)	<0.001	0.018 (0.013 – 0.023)	<0.001	-0.025 (-0.031 – -0.018)	<0.001
Model 4	-0.010 (-0.015 – -0.004)	<0.001	0.019 (0.014 – 0.024)	<0.001	-0.024 (-0.030 – -0.017)	<0.001
Categorical cumulative predictor variable from ages 11 – 24 years						
Moderate to Vigorous Physical Activity (<40mins/day as reference)						
40 – <60mins/day	-0.011 (-0.026 – 0.003)	0.117	-0.007 (-0.019 – 0.006)	0.288	-0.053 (-0.068 – -0.038)	<0.001
≥60mins/day	-0.020 (-0.036 – -0.005)	0.011	0.012 (-0.002 – 0.026)	0.081	-0.097 (-0.114 – -0.080)	<0.0001

N=6059	Total fat mass		Lean mass		Lean mass/fat mass ratio	
	β (95% CI)	p-value	β (95% CI)	p-value	β (95% CI)	p-value
Continuous cumulative predictor variables from ages 11 – 24 years						
Sedentary Time (mins/day)						
Model 1	0.120 (0.113 – 0.128)	<0.0001	0.225 (0.217 – 0.233)	<0.0001	-0.064 (-0.593 – 0.465)	0.813
Model 2	0.029 (0.023 – 0.035)	<0.0001	0.040 (0.035 – 0.045)	<0.0001	-0.071 (-0.735 – 0.593)	0.833
Model 3	0.022 (0.016 – 0.027)	<0.001	0.032 (0.027 – 0.037)	<0.0001	-0.189 (-0.814 – 0.437)	0.554
Model 4	0.019 (0.013 – 0.024)	<0.001	0.033 (0.028 – 0.038)	<0.0001	-0.174 (-0.776 – 0.429)	0.572
Light Physical Activity (mins/day)						
Model 1	-0.140 (-0.148 – -0.133)	<0.0001	-0.237 (-0.245 – -0.229)	<0.0001	-0.330 (-0.933 – 0.273)	0.283
Model 2	-0.038 (-0.044 – -0.032)	<0.0001	-0.042 (-0.048 – -0.037)	<0.0001	-0.490 (-1.083 – 0.103)	0.105
Model 3	-0.033 (-0.039 – -0.028)	<0.0001	-0.035 (-0.040 – -0.031)	<0.0001	-0.531 (-1.078 – 0.015)	0.057
Model 4	-0.032 (-0.038 – -0.027)	<0.0001	-0.036 (-0.041 – -0.031)	<0.0001	-0.537 (-1.072 – -0.001)	0.050
Moderate to Vigorous Physical Activity (mins/day)						
Model 1	-0.056 (-0.065 – -0.046)	<0.0001	-0.055 (-0.067 – -0.044)	<0.0001	0.081 (-0.439 – 0.602)	0.760
Model 2	-0.031 (-0.038 – -0.024)	<0.001	0.000 (-0.006 – 0.005)	0.902	0.125 (-0.428 – 0.677)	0.659
Model 3	-0.028 (-0.035 – -0.021)	<0.001	0.004 (-0.001 – 0.010)	0.110	0.118 (-0.396 – 0.623)	0.653
Model 4	-0.027 (-0.034 – -0.020)	<0.001	0.006 (0.000 – 0.011)	0.038	0.138 (-0.365 – 0.641)	0.591
Categorical cumulative predictor variable from ages 11 – 24 years						
Moderate to Vigorous Physical Activity (<40mins/day as reference)						
40 – <60mins/day	-0.048 (-0.063 – -0.033)	<0.001	0.012 (-0.001 – 0.025)	0.061	0.809 (-1.009 – 2.627)	0.383
≥60mins/day	-0.090 (-0.107 – -0.073)	<0.0001	0.017 (0.002 – 0.032)	0.025	1.366 (-0.218 – 2.951)	0.091

For continuous variable analyses, model 1 was unadjusted. Model 2 was adjusted for sex, family history of hypertension/diabetes/high cholesterol/vascular disease, socioeconomic status, and other time-varying covariates measured at both baseline and follow-up such as age, low-density lipoprotein cholesterol, triglyceride, high sensitivity C-reactive protein, high-density lipoprotein cholesterol, heart rate, systolic blood pressure, glucose, insulin, smoking status, and fat mass or lean mass, depending on the outcome.

Model 3 was an additional adjustment for sedentary time (LPA and MVPA model) or light physical activity (Sedentary time model). Model 4 was an additional adjustment for light physical activity (MVPA model) or moderate to vigorous physical activity (Sedentary time and LPA model). For categorical predictor variable analyses, all the above-listed covariates were adjusted for in one model. Skewed covariates were logarithmically transformed. Standardized regression coefficients (β) were computed from the generalized linear mixed-effect model for repeated measures, direct effect estimates are presented; CI, confidence interval. A 2-sided P-value <0.05 is considered statistically significant. Multiple testing was corrected with Sidak correction. Multiple imputations were used to account for missing variables. For continuous variable predictors (ST, LPA and MVPA), a 1-standard deviation change is associated with a 1-standard deviation change in the outcome. For categorical variable predictor (MVPA), time spent in a category in relation to the reference is associated with a 1-standard deviation change in the outcome.

I'm also still unconvinced as to why the 6059 participants with at least some activity and body composition data could not be used for the primary analyses. In a way, I understand your argument that "since readers who are mostly not biostatisticians are likely to question conclusions based on such a large amount of imputed data", but it is standard practice in (bio)statistics to consider the analyses using imputed data as the primary (since they ought to be closest to the "truth" on average) and to present complete-case analyses in a supplement (in part as a sensitivity analysis but also to facilitate exploring the missingness mechanism itself). As you note on Lines 203–204, "Where multiple imputations have been conducted, presenting imputed results is preferred to presenting non-imputed results." and Lines 127–129 and 297–298 sound very much like a sensitivity analysis to me. I appreciated the additional results on Lines 408–415, but was disappointed that the differences did not seem addressed in the discussion. The coefficients in Supplementary Table 12 (page 1 of your rebuttal) change to a meaningful extent in places. Setting aside the effect on statistical significance, it is again the effect sizes (the practical or clinical significance of the findings) that are key here for me. What is the reader to make of these differences?

Response 2:

As suggested, analyses involving 6059 participants have been used for primary analyses. Descriptive characteristics, whole cohort, and sex-based analyses are presented in Tables 1, 2, and 3, respectively.

The responses to point 2 (about compositional analyses), in particular the importance of considering the order of adding activity variables, strengthens my conviction that compositional analyses should be used throughout. I take your point around sleep time, but in any case, I struggle with the interpretation of a coefficient, say comparing myself against my counterfactual who is identical except that they have a substantially higher MVPA, say 60 minutes more, when the coefficients presented assume that my and their Sedentary and LPA times are identical. Including other variables in the model here means that to interpret one exposure/predictor variable's coefficient, we need to imagine holding all of the other variables' coefficients constant. Even if the results were similar (or even equivalent), with a compositional data analysis approach, the "more correct" strategy is still my preference.

Response 3:

Compositional data analysis results have been presented in Supplementary Table 14 as copied below. We observed that compared with the standard analytical strategy, the effect estimates from compositional data were 95% lower for sedentary time (0.019 vs. 0.002) and light physical activity (-0.032 vs. -0.001) in relation to total fat mass.

However, the effect estimate for MVPA in relation to total fat mass after full adjustment was slightly higher by 22% (-0.033 vs. -0.027) for compositional data analysis in comparison with the standard analytic strategy. Nonetheless, the non-adjusted effect estimate for MVPA in relation to total fat mass was similar in compositional and standard analysis (-0.055 vs. -0.056).

Supplemental Table 14 Compositional data analysis of the longitudinal associations of sedentary time and physical activity with body composition from ages 11 through 24 years in 6059 participants

N=6059	Total fat mass		Lean mass		Trunk fat mass	
	β (95% CI)	p -value	β (95% CI)	p -value	β (95% CI)	p -value
ST relative to LPA and MVPA						
Model 1	0.006 (0.005 – 0.007)	<0.0001	0.008 (0.007 – 0.009)	<0.0001	0.006 (0.005 – 0.007)	<0.0001
Model 2	0.002 (0.001 – 0.003)	<0.001	0.001 (0.000 – 0.001)	0.200	0.001 (0.001 – 0.002)	0.002
LPA relative to ST and MVPA						
Model 1	-0.004 (-0.005 – -0.003)	<0.001	-0.006 (-0.007 – -0.005)	<0.0001	-0.004 (-0.005 – -0.003)	<0.001
Model 2	-0.001 (-0.002 – 0.000)	0.026	-0.001 (-0.001 – 0.000)	0.115	-0.001 (-0.001 – 0.000)	0.244
MVPA relative to ST and LPA						
Model 1	-0.055 (-0.065 – -0.045)	<0.0001	-0.061 (-0.073 – -0.049)	<0.0001	-0.061 (-0.072 – -0.051)	<0.0001
Model 2	-0.033 (-0.041 – -0.025)	<0.001	0.003 (-0.004 – 0.010)	0.393	-0.026 (-0.034 – -0.018)	<0.001
LPA relative to ST						
Model 1	-0.004 (-0.005 – -0.003)	<0.0001	-0.006 (-0.007 – -0.005)	<0.0001	-0.004 (-0.005 – -0.003)	<0.0001
Model 2	-0.001 (-0.002 – -0.001)	<0.001	-0.001 (-0.001 – 0.000)	0.149	-0.001 (-0.002 – 0.000)	0.028
MVPA relative to ST						
Model 1	-0.012 (-0.014 – -0.010)	<0.0001	-0.016 (-0.018 – -0.014)	<0.0001	-0.012 (-0.014 – -0.011)	<0.0001
Model 2	-0.005 (-0.007 – -0.004)	<0.001	-0.001 (-0.002 – 0.001)	0.356	-0.004 (-0.005 – -0.002)	<0.001

Model 1 was unadjusted, Model 2 was adjusted for sex, family history of hypertension/diabetes/high cholesterol/vascular disease, socioeconomic status, and time-varying covariates measured at both baseline and follow-up such as age, low-density lipoprotein cholesterol, triglyceride, high sensitivity C-reactive protein, high-density lipoprotein cholesterol, heart rate, systolic blood pressure, glucose, insulin, smoking status, in addition to lean mass or total fat mass depending on the outcome. Skewed covariates were logarithmically transformed. Standardized regression coefficients (β) were computed from generalized linear mixed-effect model for repeated measures; CI, confidence interval; LPA, light physical activity; MVPA, moderate to vigorous physical activity; ST, sedentary time. A 2-sided P-value <0.05 is considered statistically significant. Multiple testing was corrected with Sidak correction. Isometric logarithmic transformation of movement behaviours was used in the compositional data analysis.

There are three aspects of the analyses here that I still think could/need to be improved (clustering within families, multiple imputation, and compositional analyses). I am most sympathetic to your points around the compositional aspects. While I do appreciate your responses and additional analyses, the relative changes in coefficients from MI and adding family clustering, alongside the challenges of interpreting

differences in one PA component while holding the others constant, leaves me still wanting these aspects incorporated, or for stronger justifications for not doing so.

Response 4:

As stated above, we have provided the requested additional analyses incorporating “(clustering within families, multiple imputation, and compositional analyses)” in Tables 1-3 (primary analyses) and Supplementary Table 14.

I appreciate that I run the risk of becoming, if I have not already done so, dogmatic here, but I think that there are strong arguments for the “most correct” statistical analyses being included as the primary analyses here. As noted above, I feel less strongly about the compositional analyses than the other two aspects.

Response 5:

The compositional analyses have been presented in Supplementary Table 14. However, the effect estimates especially for sedentary time and light physical activity were significantly smaller than standard analysis results (Table 2 and Supplemental Table 14).

.....

Reviewer #3 (Remarks to the Author):

I thank the authors for addressing my remaining concerns. Congratulations to this important work. No further comments.

Response 6:

We thank the reviewer for the helpful suggestions and compliments.

Reviewers' Comments:

Reviewer #1:

Remarks to the Author:

Thank you for your extremely constructive revisions. I appreciate that these were not small undertakings. I am satisfied with all of your revisions and have no additional requests or comments.